# Revisit last-iterate convergence of mSGD under milder requirement on step size

**Ruinan Jin**[1]    **Xingkang He**[2]*    **Lang Chen**[3]    **Difei Cheng**[4]    **Vijay Gupta**[2,5]

[1]The Chinese University of Hong Kong, Shenzhen, China
[2]Department of Electrical Engineering, University of Notre Dame, IN, USA
[3]College of Information Engineering, Shenyang University of Chemical Technology, Shenyang, China
[4]Institute of Automation, Chinese Academy of Sciences, Beijing, China
[5]Elmore Family School of Electrical and Computer Engineering, Purdue University, IN, USA

## Abstract

Understanding convergence of stochastic gradient descent (SGD) based optimization algorithms can help deal with enormous machine learning problems. To ensure last-iterate convergence of SGD and momentum-based SGD (mSGD), the existing studies usually constrain the step size $\epsilon_n$ to decay as $\sum_{n=1}^{+\infty} \epsilon_n^2 < +\infty$, which however is rather conservative and may lead to slow convergence in the early stage of the iteration. In this paper, we relax this requirement by studying an alternate step size for the mSGD. First, we relax the requirement of the decay on step size to $\sum_{n=1}^{+\infty} \epsilon_n^{2+\eta_0} < +\infty$ ($0 \leq \eta_0 < 1/2$). This implies that a larger step size, such as $\epsilon_n = \frac{1}{\sqrt{n}}$ can be utilized for accelerating the mSGD in the early stage. Under this new step size and some common conditions, we prove that the gradient norm of mSGD for a class of non-convex loss functions asymptotically decays to zero. In addition, we show that this step size can indeed help make the iterates of mSGD converge into a neighborhood of the stationary points quicker in the early stage. Finally, we establish the convergence of mSGD under a constant step size $\epsilon_n \equiv \epsilon > 0$ by removing a common requirement in the literature on strong convexity of the loss functions. Some experiments are given to illustrate the developed results.

## 1 Introduction

The booming development of machine learning over the past decade relies on the employment of effective optimization algorithms for training parameterized machine learning models (e.g., neural networks). A large number of such optimization algorithms are based on gradient descent (GD). The optimization problem in machine learning can be cast as minimizing a loss function $g(\theta) \in \mathbb{R}$ over the choice of an $N$-dimensional real-valued parameter vector $\theta$, i.e., by solving the problem

$$\theta^* = \arg \min_{\theta \in \mathbb{R}^N} g(\theta). \tag{1}$$

This problem can be solved with a typical GD algorithm through an iteration of the form

$$\theta_{n+1} = \theta_n - \epsilon_n \nabla_{\theta_n} g(\theta_n), \tag{2}$$

where $\theta_n$ is the estimate of $\theta^*$ at step $n$, $\epsilon_n$ is a positive step size (learning rate) to be designed, and $\nabla_{\theta_n} g(\theta_n)$ stands for the gradient of $g(\theta_n)$ at step $n$. Under certain technical conditions, $\theta_n$ in (2) can asymptotically (as $n \to \infty$) converge to the optimal solution $\theta^*$. However, (2) is not efficient in machine leaning applications with enormous training data. To accelerate (2), one attempt is stochastic

---

*Corresponding author: xhe9@nd.edu

36th Conference on Neural Information Processing Systems (NeurIPS 2022).

gradient descent (SGD) originating from [1]. Instead of calculating $\nabla_{\theta_n} g(\theta_n)$ over all the training data, SGD computes gradient estimate $\nabla_{\theta_n} g(\theta_n, \xi_n)$ by sampling a subset of data (following the distribution of sampling noise $\{\xi_n\}$ and then updates the iterate as follows

$$\theta_{n+1} = \theta_n - \epsilon_n \nabla_{\theta_n} g(\theta_n, \xi_n). \tag{3}$$

In addition to sampling noise, $\xi_n$ can be used to model any external noises arising from the gradient computation.

However the price to pay for this data efficiency is that the employment of stochastic gradients (3) displays a relatively slow convergence rate during the learning process. In order to improve the convergence rate of SGD, momentum-based stochastic gradient descend (mSGD), which reduces the update variance by averaging the past gradients ([2]), has been proposed. A typical iteration form of mSGD is as follows ([3, 4]):

$$v_n = \alpha v_{n-1} + \epsilon_n \nabla_{\theta_n} g(\theta_n, \xi_n), \quad \theta_{n+1} = \theta_n - v_n, \tag{4}$$

where $\alpha \in [0, 1)$ and $\epsilon_n > 0$ are momentum coefficient and step size (learning rate), respectively. An alternative formulation of mSGD, named stochastic heavy ball (SHB) [5, 6], has also been proposed as follows

$$v_n = \beta_n v_{n-1} + (1 - \beta_n) \nabla_{\theta_n} g(\theta_n, \xi_n), \quad \theta_{n+1} = \theta_n - \gamma_n v_n, \tag{5}$$

where $\beta_n \in (0, 1)$ and $\gamma_n$ are momentum coefficient and step size. It has been shown that mSGD and SHB are essentially equivalent [7]. In recent years, mSGD has been widely employed in many applications of deep learning such as image classification [8], fault diagnosis [9], statistical image reconstruction [10], among others. Moreover, a number of variants on momentum have been emerging, see, e.g., synthesized Nesterov variants [11], robust momentum [12], and PID-control based methods [13]. The importance of momentum in deep learning has been illustrated in [4] through experiments.

There has been a long line of literature analyzing the convergence of SGD and mSGD algorithms. Regarding SGD, authors in [14] studied the last-iterate convergence when the step size is chosen to decay as $1/n$. Authors in [15, 16] established the last-iterate convergence of SGD for strongly convex loss functions without the bounded gradients assumption. For the normalized mSGD (SHB), Polyak [2, 17] and Kaniovski [18] established the convergence (subsequence convergence and convergence of time averages) properties for convex loss functions. Igor Gitman [5] provided convergence results of mSGD (SHB) for non-convex loss functions under a somewhat restrictive requirement on uniform boundedness of a noise term $\mathbb{E}(\|\nabla_{\theta_n} g(\theta_n, \xi_n) - \nabla_{\theta_n} g(\theta_n)\|) \leq \delta$. It has been pointed out [4, 19] that the designs of momentum coefficients in [2, 5, 17, 18] may not be consistent with the requirements of some practical applications. The last-iterate convergence of mSGD for non-convex loss functions has recently been established in [7, 20] with the step size condition $\sum_{n=1}^{+\infty} \epsilon_n^2 < +\infty$, which has been widely utilized in stochastic optimization [1, 15, 16, 21]. However, this condition requires the step size to decay relatively fast, so it may lead to slow convergence especially in the early stage of the iteration. With the requirement that the loss function is strongly convex, [22] showed that the last-iterate of mSGD can converge to a neighbor of the stationary point when the step size is a constant.

## 2  Problem of interest and contributions

In this paper, we consider the problem of last-iterate convergence of mSGD under relaxed requirements on step size. Specifically, we seek to relax the condition $\sum_{n=1}^{+\infty} \epsilon_n^2 < +\infty$ [1, 7, 15, 16, 21] that has been required in the literature for proving last-iterate convergence of mSGD. We consider two possible step sizes: step size that decay as $\epsilon_n = \frac{1}{\sqrt{n}}$ and constant step size $\epsilon_n \equiv \epsilon > 0$. Using such larger step size is expected to lead to the last iterate of mSGD (i.e., $\theta_n$) converging faster. We note that since the SGD (3) is a special case of mSGD (4), the developed results in this paper for mSGD also work for SGD.

The contributions of this paper are summarized as follows:

1) Under the setting of step size decaying as $\epsilon_n = \frac{1}{\sqrt{n}}$, we prove that the gradient norm of mSGD for a class of non-convex loss functions asymptotically decays to zero. This result is more general than mean-square gradient convergence established in [20]. In fact, this result holds for any step sizes of

the form $\sum_{n=1}^{+\infty} \epsilon_n^{2+\eta_0} < +\infty$ $(0 \leq \eta_0 < 1/2)$, which is a superset of the widely required condition $\sum_{n=1}^{+\infty} \epsilon_n^2 < +\infty$ [1, 7, 15, 16, 21]. Given this relaxed step size condition, one can run mSGD by employing a larger step size like $\epsilon_n = \frac{1}{\sqrt{n}}$ for last-iterate convergence.

2) Under some mild conditions, we provide an estimate of the convergence rate of mSGD. Furthermore, under a probability-based metric, we show that the new step size can help to improve the convergence speed in the early stage of the iteration.

3) Under the setting of constant step size, first we prove that given any small neighborhood of stationary points, it is feasible to design a step size, such that there is a convergent iterate subsequence of mSGD staying within the neighborhood almost surely. In addition, we prove that the mean-square gradient can be arbitrary small by tunning the step size. Comparing with [22], we remove the requirement of strong convexity on the loss functions.

Regarding the definitions on sequence convergence, the following ones from literature are typical. For a stochastic variable sequence $\{\xi_n\} \in \mathbb{R}^N$ with 0 as the unique limit point, the sequence is said to satisfy last-iterate convergence if $\lim_{n \to +\infty} \|\xi_n\| = 0, a.s.$; and time-average (mean-square convergence) if $\lim_{n \to +\infty} \frac{1}{n} \sum_{k=1}^{n} \mathbb{E}(\|\xi_k\|^2) = 0$. It can be proved that last-iterate convergence implies time-average convergence under the general setting of the considered problem.

**Paper outline.** The rest of the paper is organized as follows. In Section 3, we provide the main results of the paper on last-iterate convergence of mSGD under decaying step size and constant step size, respectively. In Section 4, two simulation experiments are given. Conclusion is made in Section 5. The main proofs are given in Appendix.

## 3 Main results

In this section, we provide the main results of this paper on the last-iterate convergence for mSGD (4) under two possible step sizes: step size that decays as $\epsilon_n = \frac{1}{\sqrt{n}}$ and constant step size $\epsilon_n \equiv \epsilon > 0$.

### 3.1 Convergence of mSGD under decaying step size

The following assumptions are needed in this paper.

**Assumption 3.1.** *Loss function $g(\theta)$ in (1) satisfies the following conditions:*

1. *Noise sequence $\{\xi_n\}$ are mutually independent and independent of $\theta_1$ and $v_0$, such that $g(\theta) = \mathbb{E}_{\xi_n}\big(g(\theta, \xi_n)\big)$ for any $\theta \in \mathbb{R}^N$.*

2. *$g(\theta)$ is a non-negative and continuously differentiable function. The set of its stationary points $J = \{\theta \mid \|\nabla_\theta g(\theta)\| = 0\}$ is a bounded set which has only finite connected components $J_1, ..., J_n$. In addition, there is $\tilde{\epsilon}_1 > 0$, such that for any $i$ and $0 < d(\theta, J_i) < \tilde{\epsilon}_1$, it holds that $\big|g(\theta) - g_i\big| \neq 0$, where $g_i = \{g(\theta) \mid \theta \in J_i\}$ is a constant.*

3. *There are two constants $M' > 0$ and $a' > 0$ such that for any $\theta \in \mathbb{R}^N$ and $n \in \mathbb{N}_+$,*

$$E_{\xi_n}\Big(\big\|\nabla_\theta g(\theta, \xi_n)\big\|^2\Big) \leq M'\big\|\nabla_\theta g(\theta)\big\|^2 + a'. \tag{6}$$

4. *The gradient $\nabla_\theta g(\theta)$ satisfies the Lipschitz condition, i.e., there is a constant $c > 0$, such that for any $x, y \in \mathbb{R}^N$,*

$$\big\|\nabla_x g(x) - \nabla_y g(y)\big\| \leq c\|x - y\|.$$

Conditions 1 and 4 in Assumption 3.1 are common in the literature [1, 15, 16, 21]. [7] required a condition on strong growth, i.e., $\mathbb{E}_{\xi_n} \|\nabla_{\xi_n} g(\theta, \xi_n)\|^2 \leq M_0 \|\nabla_\theta g(\theta)\|^2$, which however makes it close to the deterministic case. In contrast, condition 3 allows more randomness of data sampling. Condition 2 is a mild condition for the reasons as follows. In some works, the non-negative condition may be replaced by a lower bound condition $g(\theta) > \hat{l}_0 > -\infty$. These two conditions are essentially equivalent, since one can construct a new loss function $\bar{g} = g - \hat{l}_0$ under the lower bound condition, such that the new loss function is non-negative. The rest of condition 2 is quite general, since it allows the loss function to have multiple stationary points and to be non-convex.

**Assumption 3.2.** *In the mSGD* (4)*, momentum coefficient $\alpha \in [0, 1)$ and the sequence of step size $\epsilon_n$ is positive and monotonically decreasing to zero, such that $\sum_{n=1}^{+\infty} \epsilon_n = +\infty$ and $\sum_{n=1}^{+\infty} \epsilon_n^{2+\eta_0} < +\infty$, where $0 \leq \eta_0 < 1/2$ is a constant.*

In Assumption 3.2, the momentum coefficient $\alpha \in [0, 1)$ is a constant. Comparing with the setting in [5, 6, 17, 18], where $\alpha^{(n)}$ tends to 1 or 0, constant momentum coefficient is more common in practice [4, 7, 19]. Regarding the step size condition, it is more general than the one in many existing works on last-iterate convergence of SGD and mSGD [1, 7, 15, 13, 16, 21], i.e., $\sum_{n=1}^{+\infty} \epsilon_n^2 < +\infty$, which is obtained from Assumption 3.2 if $\eta_0 = 0$. Under the step size condition in Assumption 3.2, one can choose larger step size like $\epsilon_n = \frac{1}{\sqrt{n}}$, which however is not feasible in the commonly required condition $\sum_{n=1}^{+\infty} \epsilon_n^2 < +\infty$. Since the new step size can decay more slowly than the existing one, it provides more space for step size fine tuning, such that the algorithm can quickly converge to a neighborhood of the stationary point [23, 24, 22].

Then we attain the first main result in this paper on last-iterate convergence of mSGD under the step size condition in Assumption 3.2.

**Theorem 3.1.** *Consider the mSGD in* (4) *with any $v_0 \in \mathbb{R}^N$ and $\theta_1 \in \mathbb{R}^N$. Under Assumptions 3.1 and 3.2, the gradient norm tends to 0 almost surely, i.e.*

$$\lim_{n \to +\infty} \|\nabla_{\theta_n} g(\theta_n)\| = 0 \; a.s..$$

Due to page constraint, the complete proof is given Appendix. Here, we provide a proof outline to present the main proof ideas.

**Proof Sketch of Theorem 3.1**: We aim to prove that $\nabla_{\theta_n} g(\theta_n) \to 0$ *a.s.* via the following key steps.

Step 1: In this step we aim to prove $\mathbb{E}\left(\epsilon_n^{\eta_0} g(\theta_n)\right)$ is uniformly upper bounded. We present this result as Lemma B.8 in Appendix.

Step 2: We prove there exists a subsequence of $\nabla_{\theta_n} g(\theta_n)$ which is convergent to 0 a.s.. It is attained by proving $\sum_{n=1}^{+\infty} \epsilon_n^{1+2\eta_0} \|\nabla_{\theta_n} g(\theta_n)\|^2 < +\infty$ *a.s.*. We present this result as Lemma B.10 in Appendix.

Step 3: We aim to extend the subsequence convergence in Step 2 to asymptotic convergence. The basic idea is to prove that the adjacent terms $g(\theta_n)$ and $g(\theta_{n+1})$ are "close" enough, such that

$$g(\theta_{n+1}) - g(\theta_n) \leq \hat{k} \epsilon_t^{1+2\eta_0} + Q_n,$$

where $\hat{k} > 0$ is a constant and $\{Q_n\}$ is a sequence, such that $\sum_{n=1}^{+\infty} Q_n$ converges a.s.. This result is presented in Lemma B.11.

Step 4: Finally, we use some techniques to attain the result of asymptotic convergence $\|\nabla_{\theta_n} g(\theta_n)\| \overset{n \to \infty}{\to} 0$ *a.s.*.

In the last-iterate convergence analysis of mSGD, there are several technical challenges when we try to extend the condition $\sum_{n=1}^{+\infty} \epsilon_n^2 < +\infty$ to $\sum_{n=1}^{+\infty} \epsilon_n^{2+\delta_0} < +\infty$ with $0 \leq \eta_0 < 1/2$. One of the challenges is: When we try to prove $\sum_{n=1}^{+\infty} \zeta_n = \sum_{n=1}^{+\infty} \epsilon_n \nabla_{\theta_n} g(\theta_n)^T (\nabla_{\theta_n} g(\theta_n, \xi_n) - \nabla_{\theta_n} g(\theta_n))$ is convergent a.s., which is the vital step to get the last-iterate convergence, we usually use the *Martingale convergence theorem*, through which we just need to prove $\sum_{n=1}^{+\infty} \mathbb{E}(\zeta_n^2)$ is convergent. If the condition $\sum_{n=1}^{+\infty} \epsilon_n^2 < +\infty$ holds, we can get $\sum_{n=1}^{+\infty} \mathbb{E}(\zeta_n^2) < \sum_{n=1}^{+\infty} \hat{K} \epsilon_n^2 < +\infty$ ($\hat{K}$ is a constant) directly. However, under the new condition $\sum_{n=1}^{+\infty} \epsilon_n^{2+\delta_0}$, we can not use this approach. Instead, we introduce a more technical approach to handle this problem.

Comparing with the corresponding result in [7] (Theorem 1 in [7]), we relax the condition on the decay of step size. Under this new condition, one can utilize step size with more slow decaying speed in the iteration of mSGD. Let $\alpha = 0$ in mSGD, then we can get a similar result for SGD

**Corollary 3.1.** *Consider the SGD in* (3) *with any $\theta_1 \in \mathbb{R}^N$. Under Assumptions 3.1 and 3.2 with $\alpha = 0$, the gradient norm tends to 0 almost surely, i.e.*

$$\lim_{n \to +\infty} \|\nabla_{\theta_n} g(\theta_n)\| = 0 \; a.s..$$

In the following, we aim to estimate the last-iterate convergence rate of $\nabla_{\theta_n} g(\theta_n)$. Instead, the convergence rate of $\mathbb{E}\left(\|\nabla_\theta g(\theta_n)\|^2\right)$ is quite meaningful. In general, if we need to estimate the convergence rate of the last iterate, we need a $probability - retention\ assumption$, i.e., $\forall k_0 > 0$, $\exists a > 0$, making $P(\|\theta_n\| > a) \sim P(\|\theta_1\| > k_0\epsilon_1^2, ..., \|\theta_n\| > k_0\epsilon_n^2)$, and we usually need some extra assumptions to establish a quantitative relationship between $g$ and $\nabla g$. Existing works were established usually under the strongly-convex assumption [25, 26, 13], local P-L condition [7], or convex loss function satisfying some stability conditions, e.g., $\mathbb{E}\|\nabla_{\theta_n} g(\theta_n)\|^2 < \delta < +\infty$. However, these assumptions are rather relatively strong, especially the stability condition which is very difficult to verify. In our paper, we need a milder assumption than the above conditions as follow.

**Assumption 3.3.** *Regarding the loss function $g(\theta)$ defined in* (1)*, there exists $\delta_0 > 0$, $T_0 > 0$, such that for any $\theta \in \{\theta | \|\nabla_\theta g(\theta)\|^2/(g(\theta) - g^*) < \delta_0\}$, it holds that $g(\theta) < T_0$, where $g^* = \inf_{\theta \in \mathbb{R}^N} g(\theta)$.*

This assumption requires the value of the loss function $g(\theta)$ to be bounded in the set $S = \{\theta | \|\nabla_\theta g(\theta)\|^2/(g(\theta) - g^*) < \delta_0\}$. The motivation of this assumption is: For a stochastic algorithm, there is always a probability at which the iterates are far away from the true value. Since second-derivative information has a more positive effect on convergence (such as strongly convex loss functions), this condition restricts iterates which are far away due to extreme noise, so that it does not have a significant effect on global updates. This requirement is milder than the strong convexity condition, which is commonly used in existing works [25, 26, 13] for the analysis of the last-iterate convergence. In addition, this assumption allows the loss function to be non-convex, and does not need the local P-L condition used in [7] on the second derivative information near stationary points. Note that Assumption 3.3 can be satisfied by many loss functions commonly encountered in machine learning. First of all, many common loss functions are bounded [27–30], and thus satisfy Assumption 3.3. For unbounded loss functions, there are a number of instances satisfying Assumption 3.3. For linear regression functions, since they are strongly convex, it is easy to verify the assumption. In the Appendix, we explain (not in a strict but an illustrating manner) this assumption can also be satisfied by a two layer neural network with Relu active function under a square loss function.

In the following theorem, we provide an estimate of the convergence rate for the mSGD in (4).

**Theorem 3.2.** *Consider the mSGD in* (4) *with a unique stationary point $\theta^*$. Then for any $v_0 \in \mathbb{R}^N$ under Assumptions 3.1–3.3 and probability-retention assumption. Then exists $a > 0$, for any $\|\nabla_{\theta_1} g(\theta_1)\|^2$, there is*

$$\mathbb{E}\left(\|\nabla_\theta g(\theta_n)\|^2\right) = O\left(\left(e^{-\frac{2\hat{c}}{(1-\alpha)^2}\sum_{i=1}^n \epsilon_i}\right)\left(\sum_{i=1}^n \epsilon_i^2 e^{\frac{2\hat{c}}{(1-\alpha)^2}\sum_{k=1}^i \epsilon_k}\right)\right), \quad (7)$$

*where $\hat{c} = \min\{a/2T_0, 2\delta_0\}$.*

Comparing with [7], we remove the strong growth condition, i.e., $\mathbb{E}_{\xi_n}\|\nabla_\theta g(\theta, \xi_n)\|^2 \le M\|\nabla_\theta g(\theta)\|^2$, and relax the step size condition to be $\sum_{n=1}^{+\infty} \epsilon_n^{2+\delta_0}$ ($0 < \delta_0 \le 1/2$). In the comparison with [20], we do not require the convexity of loss function or $\mathbb{E}\|\nabla_{\theta_n} g(\theta_n)\|^2 < G$. In addition, we relax the step size condition in [20]. There are quite a few results on the average iterates in the literature, such as [13, 31, 32]. In [13, 31, 32], the convergence rate $O(\frac{1}{\sqrt{n}})$ of mSGD is established with the step size $\frac{1}{\sqrt{n}}$. According to our result, let $\epsilon_n = \frac{1}{\sqrt{n}}$, it holds that

$$\mathbb{E}(\|\nabla_{\theta_n} g(\theta_n)\|^2) = O\left(e^{-\sqrt{n}} \sum_{k=1}^n \frac{e^{\sqrt{k}}}{k}\right) = O\left(\frac{1}{\sqrt{n}}\right).$$

It can be proved that the convergence rate of average iterates is also $O\left(\frac{1}{\sqrt{n}}\right)$.

From (7), we see that larger step size may not lead to larger convergence rate. This is reasonable, since the convergence rate reflects how fast the iterate $\theta_n$ converges when $n \to +\infty$ and a larger step size usually makes the iterate converges to a neighborhood of stationary point quicker. We have the following theorem on this point.

**Theorem 3.3.** *Consider the mSGD in* (4) *with any $v_0 \in \mathbb{R}^N$. Under Assumptions 3.1–3.3, given any $\hat{a} > 0$, if $\|\nabla_{\theta_1} g(\theta_1)\|^2 > \hat{a}$, then it holds that*

$$P(\tau^{(\hat{a})} \ge n) = O\left(e^{-\frac{2\hat{c}}{(1-\alpha)^2}\sum_{i=1}^n \epsilon_i}\right),$$

*where* $\tau^{(\hat{a})} = \min_{n>0}\{\|\nabla_{\theta_n}g(\theta_n)\|^2 < \hat{a}\}$ *and* $\hat{c} = \min\{\hat{a}/2T_0, 2\delta_0\}$.

Theorem 3.3 provides a probability description of how fast the iterates of mSGD converge into the preset neighborhood of stationary points. It can be illustrated as follows. It follows from Theorem 3.3 that

$$P(\tau^{(a)} < n) > 1 - \hat{k}_1 e^{-\frac{2\hat{c}}{(1-\alpha)^2}\sum_{i=1}^{n}\epsilon_i},$$

where $\hat{k}_1 > 0$ is a constant. From the inequality, as $n$ increases, the event $\|\nabla_{\theta_n}g(\theta_n)\|^2 < a$ occurs with a higher and higher probability tending to probability 1. The influence of momentum coefficient $\alpha$ and step size $\epsilon_n$ is also quantified. We note that with the measure in Theorem 3.3, the convergence to a neighborhood of the stationary point is faster than under the traditional condition $\sum_{n=1}^{+\infty}\epsilon_n^2 < +\infty$ by setting the step size with a relatively slow decay speed as $\sum_{n=1}^{+\infty}\epsilon_n^{2+\delta_0}$ $(0 < \delta_0 \leq 1/2)$, such as $\epsilon_n = \frac{1}{\sqrt{n}}$ and $\eta_0 = 1/4$. Similar results (the large step size can make the iteration quickly converge to a neighborhood of stationary point, but may converge slowly in the late stage of the algorithm) appear in [23, 24, 22] through experiments.

## 3.2 Convergence of mSGD under constant step size

In the previous subsection, we establish the last-iterate convergence of mSGD under a relaxed decay condition on step size. In this subsection, we study the convergence of mSGD under a constant step size $\epsilon_n \equiv \epsilon > 0$. We start with the following result on subsequence convergence.

**Theorem 3.4.** *Consider the mSGD in* (4) *with any* $v_0 \in \mathbb{R}^N$ *and* $\theta_1 \in \mathbb{R}^N$. *Under Assumption 3.1, for any* $\varphi > 0$, *there exists* $\mu_0^{(\varphi)} > 0$, *such that for any* $0 < \epsilon_n \equiv \epsilon < \mu_0^{(\varphi)}$, *it holds that*

$$\left\|\nabla_{\theta_{k_n}^{(\varphi)}}g(\theta_{k_n}^{(\varphi)})\right\|^2 \leq \varphi, \ a.s.$$

*where* $\{\theta_{k_n}^{(\varphi)}\}$ *is a subsequence* $\{\theta_n\}$.

*Proof.* Due to page constraint, the complete proof is given Appendix. Here, we provide a proof outline to present the main proof ideas.

To study the mSGD with constant step size, we notice that the impact of the noise cannot be totally eliminated through the step size. But when $\|\nabla_{\theta_n}g(\theta_n)\|$ is relatively large (meaning the distance between $\theta_n$ and the stationary point $\theta^*$ is relatively large), it holds that $\mathbb{E}_{\xi_n}\left(\|\nabla_{\theta_n}g(\theta_n, \xi_n)\|^2\right) \leq M'\|\nabla_{\theta_n}g(\theta_n)\|^2 + a' \leq (M' + \hat{k})\|\nabla_{\theta_n}g(\theta_n)\|^2$. This implies that with a proper step size $\epsilon$, the noise can be controlled. As a result, the algorithm can be stabilized around a stationary point. In the following, several key steps in the proof are provided.

Step 1: We define an event by $A_n^{(\varphi)} = \left\{\|\nabla_{\theta_1}g(\theta_1)\|^2 > \varphi, \|\nabla_{\theta_2}g(\theta_2)\|^2 > \varphi, ..., \|\nabla_{\theta_n}g(\theta_n)\|^2 > \varphi\right\}$ and a characteristic function of this event by $I_n^{(\varphi)}$.

Step 2: We prove that the probability of $A_n^{(\varphi)}$ is tending to 0 as $n \to \infty$. First, we attain a recursion formula:

$$\mathbb{E}\left(I_{t+1}^{(\varphi)}g(\theta_{t+1})\right) - \mathbb{E}\left(I_t^{(\varphi)}g(\theta_t)\right)$$
$$\leq -\alpha^{t-1}I_1^{(\varphi)}\mathbb{E}\left(\nabla_{\theta_1}g(\theta_1)^\mathsf{T}v_1\right) - \sum_{i=2}^{t}\alpha^{t-i}(\epsilon - \hat{k}M'\epsilon^2)\mathbb{E}\left(I_{i-1}^{(\varphi)}\|\nabla_{\theta_i}g(\theta_i)\|^2\right).$$

By taking the sum of multiple inequalities as above, we have

$$\sum_{i=2}^{t}\mathbb{E}\left(I_{i-1}^{(\varphi)}\|\nabla_{\theta_i}g(\theta_i)\|^2\right) < +\infty.$$

This leads to

$$\mathbb{E}\left(I_{t-1}^{(\varphi)}\|\nabla_{\theta_t}g(\theta_t)\|^2\right) \to 0.$$

From the *Chebyshev inequality*, we attain $P(A_n^{(\varphi)}) \to 0$.

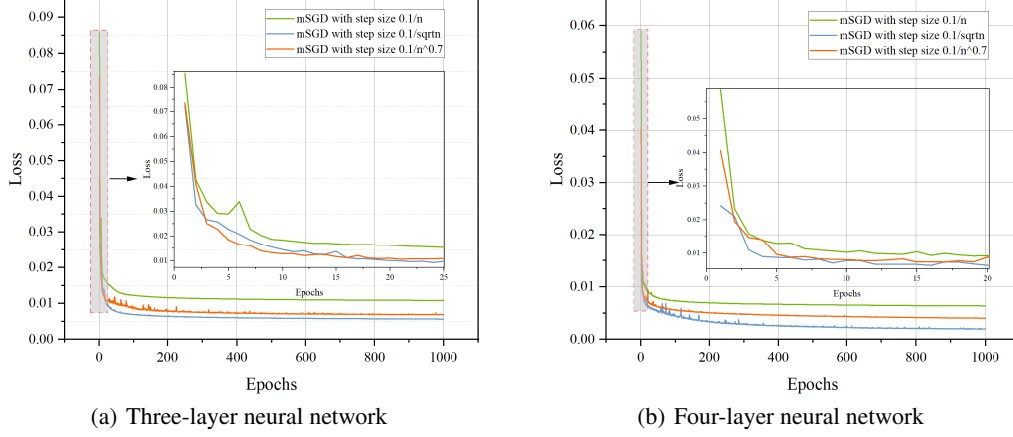

(a) Three-layer neural network

(b) Four-layer neural network

Figure 1: Training loss-iteration chart on the Boston house-price data.

Step 3: Step 2 implies that with probability 1 there exists at least one instant $n$ at which $\|\nabla_{\theta_n} g(\theta_n)\|^2 \leq \varphi$. We note that for the time instant $m$ at which $\|\nabla_{\theta_m} g(\theta_m)\|^2 > \varphi$, we can use a new start point. By repeating the process of Step 2, we attain the conclusion of this theorem.

$\square$

Theorem 3.4 shows that under the common conditions in the literature, the constant step size can guarantee the subsequence convergence of mSGD (4) in any small neighborhood of stationary point. In addition, we have a similar result on $\mathbb{E}\left(\|\nabla_{\theta_n} g(\theta_n)\|^2\right)$ as follows.

**Proposition 3.1.** *Suppose $\{\theta_n\}$ is a sequence generated by* (4) *with any $\theta_1 \in \mathbb{R}^N$ and $v_0 \in \mathbb{R}^N$. Under Assumptions 3.1 and 3.3, if* (1) *has a unique solution $\theta^*$ and $\inf_{d(\theta,\theta^*)>\delta_0} \|\nabla_\theta g(\theta)\|^2 > 0 \ (\forall \delta_0 > 0)$, then for any $\varphi_0 > 0$, there exist scalars $\mu_0^{(\varphi_0)} > 0$ and $n_0 > 0$, such that for any $\epsilon_n \equiv \epsilon < \mu_0^{(\varphi_0)}$ and $n > n_0$, it holds that*

$$\mathbb{E}\left(\|\nabla_{\theta_n} g(\theta_n)\|^2\right) \leq \varphi_0.$$

This result is a substantial extension of the existing results on SGD with constant step size to mSGD. Comparing with the results in [22], we remove the requirement of $\mu$-strong convexity on loss function.

## 4 Experiments

Since this paper focuses on convergence analysis on the well-known mSGD, two relatively simple experiments on regression and classification tasks are given respectively to show the effectiveness of the results.

### 4.1 Regression task

In this subsection, we study a house price prediction problem with neural networks trained by mSGD.

**Network Architectures**. We respectively employ a 3-layer and a 4-layer fully-connected neural network with ReLu active function and squared loss function.

**Implementation**. We implement two neural networks to train on this dataset using Keras. The first neural network consists of three fully connected layers with 13, 8, and 1 neurons. The second neural network consists of four fully connected layers with 13, 32, 16, 1 neurons, respectively. We initialize the weights use glorot uniform algorithm. We use mSGD with a mini-batch size of 16, and use mean square error loss function to train the model. The momentum term coefficient is 0.9 and the model is trained for up to 1000 epochs. The training takes about two hours at a time in 3080GPU. We do not use dropout.

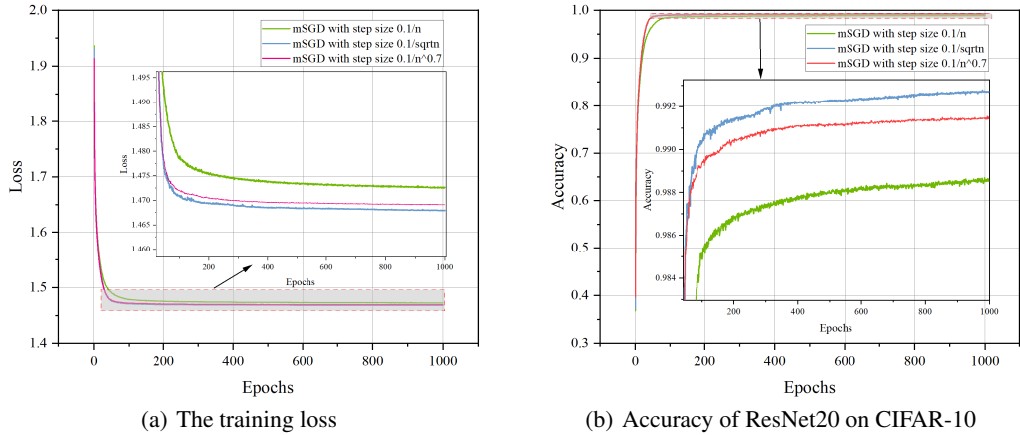

(a) The training loss          (b) Accuracy of ResNet20 on CIFAR-10

Figure 2: Training and prediction performance on CIFAR-10.

**Dataset**. The training dataset we use is from Boston House Price Dataset[2]. This dataset is a regression prediction dataset, which consists of 506 sets of data with dimension 13. During the training process, we randomly divide $30\%$ as the test set and normalize the data before training.

**Results**. We use the mSGD in (4) under three different step sizes respectively: $\epsilon_n = \frac{1}{n}$, $\epsilon_n = \frac{1}{n^{0.7}}$, and $\epsilon_n = \frac{1}{n^{0.5}}$. The experiment results are given in Figure 1. The figure shows that 1) the loss decays to zero under the three step size settings; 2) $\epsilon_n = \frac{1}{n^{0.5}}$ can make the loss tend to a small neighborhood of zero fastest among the three step size settings. This conforms to the theoretical analysis in Theorems 3.1–3.3.

## 4.2 Classification task

In this subsection, we study an image classification problem with neural networks trained by mSGD.

**Network Architectures**. We employ a 20-layer ResNet network. The convolutional layers mostly have 3×3 filters and follow two simple design rules: (i) for the same output feature map size, the layers have the same number of filters; (ii) if the feature map size is halved, the number of filters is doubled so as to preserve the time complexity per layer. We perform downsampling directly by convolutional layers that have a stride of 2 and padding with valid. The network ends with an average pooling layer and a 10-way fully-connected layer with softmax.

**Implementation**. We implement the ResNet20 network using Keras. We initialize the weights use glorot uniform algorithm. We use mSGD with a mini-batch size of 64, momentum coefficient 0.9 and use categorical crossentropy loss function to train the model. The models are trained for up to 1000 epochs, taking about two hours at a time in 3080GPU. We do not use dropout.

**Dataset**. We use two different datasets CIFAR-10 and CIFAR-100[3]. CIFAR-10 consists of 50k training images and 10k testing images in 10 classes. CIFAR-10 is a color image dataset close to ubiquitous objects, and the image size is 32x32x3. CIFAR-100 consists of 50k training images and 10k testing images in 100 classes. Each category contains 500 training images and 100 testing images. CIFAR-100 is a color image dataset closer to ubiquitous objects, and the image size is 32x32x3. Normalize the dataset between 0-1 before training.

**Results**. We use the mSGD in (4) under three different step sizes: $\epsilon_n = \frac{1}{n}$, $\epsilon_n = \frac{1}{n^{0.7}}$, and $\epsilon_n = \frac{1}{n^{0.5}}$. The experiment results are given in Figure 2 and Figure 3. The figures show that 1) the loss decays to zero and the accuracy goes up to 1 under the three step size settings; 2) $\epsilon_n = \frac{1}{n^{0.5}}$ can make the loss tend to a small neighborhood of zero and make the accuracy tend to a small neighborhood of 1 fastest among the three step size settings, followed by the setting of $\epsilon_n = \frac{1}{n^{0.7}}$. These results conform to the theoretical analysis in Theorems 3.1–3.3.

---

[2] https://www.cs.toronto.edu/~delve/data/boston/bostonDetail.html
[3] https://www.cs.toronto.edu/~kriz/cifar.html

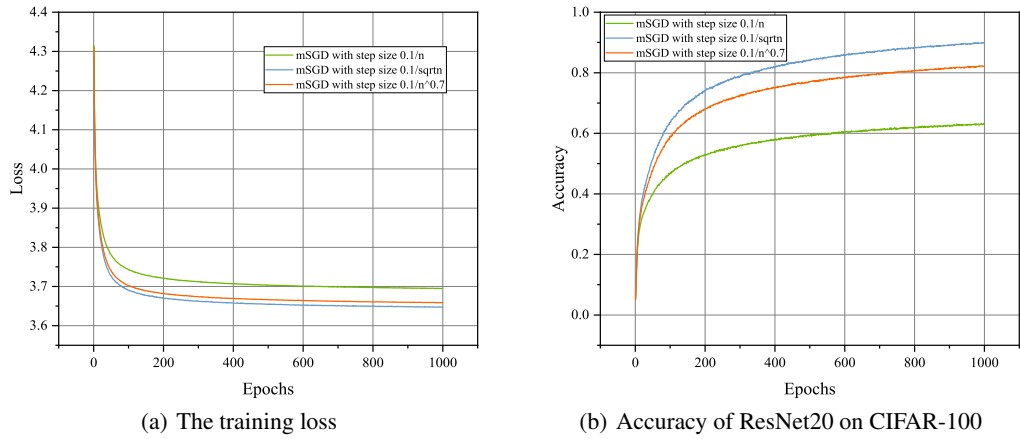

(a) The training loss        (b) Accuracy of ResNet20 on CIFAR-100

Figure 3: Training and prediction performance on CIFAR-100.

# 5 Conclusion

In this paper, we studied the last-iterate convergence of momentum-based stochastic gradient descent (mSGD) under relaxed conditions on step sizes. First, under the relaxed condition $\sum_{n=1}^{+\infty} \epsilon_n^{2+\eta_0} < +\infty$ $(0 \leq \eta_0 < 1/2)$, we proved the last-iterate convergence of mSGD for a class of non-convex loss functions. In addition, we showed that this step size can indeed help to improve the convergence speed in the early stage of the algorithm by quantifying the influence of the step size and momentum coefficient. This implies that a larger step size, such as $\epsilon_n = \frac{1}{\sqrt{n}}$ can be utilized with guaranteed convergence. We also proved that the algorithm with a constant step size (i.e., $\epsilon_n \equiv \epsilon > 0$) can ensure the last-iterate convergence of mSGD without requiring the strong convexity assumption on loss functions.

# 6 Acknowledgments and Disclosure of Funding

Xingkang He was supported in part by ARO Grant W911NF1910483 and AFOSR Grant FA9550-21-1-0231. Vijay Gupta were supported in part by NSF Grants 2020246, 2225978, and 2208794. We thank the anonymous reviewers for helpful comments and suggestions.

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
