# Appendix

## A An illustration example for Assumption 3.3

In the following, we show that Assumption 3.3 can be met in simple neural networks by using a two layer neural network with $Relu$ active function under a square loss function. First we have that

$$g(\theta, W) = \frac{1}{N} \sum_{i=1}^{N} \left| y_i - \theta^T \sigma(W x_i) \right|^2,$$

then we can calculate its partial derivatives with respect to $\theta$ and $W$

$$\frac{\partial g(\theta, W)}{\partial \theta} = \frac{2}{N} \sum_{i=1}^{N} \left( y_i - \theta^T \sigma(W x_i) \right) \sigma(W x_i),$$

$$\frac{\partial g(\theta, W)}{\partial W} = \frac{2}{N} \sum_{i=1}^{N} \left( y_i - \theta^T \sigma(W x_i) \right) (\theta \odot \mathbf{1}_{W x_i \geq 0}) x_i^T.$$

For illustration, we just consider the situation which only has one date, i.e., $N = 1$. We derive that

$$\left\| \frac{\partial g(\theta, W)}{\partial (\theta, W)} \right\|^2 = \left\| \frac{\partial g(\theta, W)}{\partial \theta} \right\|^2 + \left\| \frac{\partial g(\theta, W)}{\partial W} \right\|^2 = 4 \left| y - \theta^T \sigma(W x) \right|^2 \|\sigma(W x)\|^2$$

$$+ 4 \left| y - \theta^T \sigma(W x) \right|^2 \|\theta \odot \mathbf{1}_{W x_i \geq 0}\|^2 \|x\|^2 = 4 \left( \|\sigma(W x)\|^2 + \|\theta \odot \mathbf{1}_{W x_i \geq 0}\|^2 \|x\|^2 \right) \left| y - \theta^T \sigma(W x) \right|^2.$$

As a result, $\|\nabla_\theta g(\theta)\|^2 / g(\theta) = 4 \left( \|\sigma(W x)\|^2 + \|\theta \odot \mathbf{1}_{W x_i \geq 0}\|^2 \|x\|^2 \right)$. When $4 \left( \|\sigma(W x)\|^2 + \|\theta \odot \mathbf{1}_{W x_i \geq 0}\|^2 \|x\|^2 \right) < \delta_0$, we obtain $\|\theta^T \sigma(W x)\|^2 \leq 1/2 \left( \|\sigma(W x)\|^2 + \|\theta \odot \mathbf{1}_{W x_i \geq 0}\|^2 \|x\|^2 \right) \leq 2\delta_0$. This ensures that $g(\theta, W)$ bounded.

## B Useful lemmas

To prove the main results, we need the following lemmas, where Lemmas B.1–B.5 are from the literature and Lemmas B.6–B.11 are first introduced in this paper with proofs given in the next section.

**Lemma B.1.** *(Lemma 1.2.3, [33]) Suppose $f(x) \in C^1$ $(x \in \mathbb{R}^N)$ with gradient satisfying the following Lipschitz condition $\|\nabla f(x) - \nabla f(y)\| \leq c\|x - y\|$, where $c > 0$, then for any $x, y \in \mathbb{R}^N$, it holds that*

$$f(y) + \nabla f(y)^T (x - y) - \frac{c}{2} \|x - y\|^2 \leq f(x) \leq f(y) + \nabla f(y)^T (x - y) + \frac{c}{2} \|x - y\|^2.$$

**Lemma B.2.** *(Lemma 10, [7]) Suppose $f(x) \in C^1$ $(x \in \mathbb{R}^N)$ with $f(x) > -\infty$ and its gradient satisfying the following Lipschitz condition $\|\nabla f(x) - \nabla f(y)\| \leq c\|x - y\|$, where $c > 0$, then $\forall x_0 \in \mathbb{R}^N$, there is*

$$\left\| \nabla f(x_0) \right\|^2 \leq 2c \left( f(x_0) - f^* \right),$$

*where $f^* = \inf_{x \in \mathbb{R}^N} f(x)$.*

**Lemma B.3.** *(Lemma 4, [7]) Suppose $f(x) \in C^1$ $(x \in \mathbb{R}^N)$ with gradient satisfying the following Lipschitz condition $\|\nabla f(x) - \nabla f(y)\| \leq c\|x - y\|$, where $c > 0$, and the set $S = \{x | \nabla f(x) = 0\}$ is bounded and only has finite connected components $\{S_1, S_2, \ldots, S_m\}$. Furthermore, assume there exists $\epsilon'_1 > 0$, such that for any $i = 1, 2, ..., m$ and $x \in \{x | 0 < d(x, S_i) < \epsilon'_1\}$, it holds that $\left| f(x) - f_i \right| \neq 0$, where $f_i = f(x)$ for $x \in S_i$. Then for any $i = 1, 2, ..., m$, if there is $\epsilon'_0 > 0$ satisfying $d(x, S_i) < \epsilon'_0$, it follows that*

$$\left\| \nabla f(x) \right\|^2 \leq 2c \left| f(x) - f_i \right|.$$

**Lemma B.4.** *[34] Suppose that $\{x_n\} \in \mathbb{R}^N$ is an $\mathcal{L}_2$ martingale difference sequence, and $(x_n, \mathcal{F}_n)$ is an adaptive process. Then it holds that $\sum_{k=0}^{\infty} x_k < +\infty$ a.s., if*

$$\sum_{n=1}^{\infty} \mathbb{E}(\|x_n\|^2) < +\infty, \quad or \quad \sum_{n=1}^{\infty} \mathbb{E} \left( \|x_n\|^2 \big| \mathcal{F}_{n-1} \right) < +\infty. \quad a.s.$$

**Lemma B.5.** *(Lemma 6, [7]) Suppose that $\{x_n\} \in \mathbb{R}^N$ is a non-negative sequence of random variables, then it holds that $\sum_{n=0}^{\infty} x_n < +\infty$ a.s., if $\sum_{n=0}^{\infty} \mathbb{E} \left( x_n \right) < +\infty$.*

**Lemma B.6.** *If $0 < \mu < 1$ and $0 < \sigma < 1$ $(\sigma \neq \mu)$ are two constants, then for any positive sequence $\{\psi_n\}$, there is*

$$\sum_{k=1}^{n} \mu^{n-k} \sum_{i=1}^{k} \sigma^{k-i} \psi_i = \hat{R} \sum_{i=1}^{n} \kappa^{n-i} \psi_i,$$

*where $1 < \hat{R} < 1/\left( 1 - \kappa^{(\omega_0 - 1)} \right)$, and $\kappa = \max\{\mu, \sigma\}$ and $\omega_0 = \log_\kappa \min\{\mu, \sigma\}$.*

**Lemma B.7.** *Let $0 < \tau < 1$ be a constant, then for any sequence $\{\hat{\epsilon}_n\}$ satisfying $\hat{\epsilon}_n \to 0$, it holds that $\sum_{n=1}^{+\infty} c_n^{(\hat{\epsilon})} = 0$, where $c_n^{(\hat{\epsilon})} = \sum_{i=1}^{n} \tau^{n-i}\hat{\epsilon}_i - (1/(1-\tau))\hat{\epsilon}_n$.*

**Lemma B.8.** *Consider the mSGD in* (4). *If Assumptions 3.1–3.2 hold, then for any $\theta_1 \in \mathbb{R}^N$, $v_0 \in \mathbb{R}^N$, there is a scalar $T(\theta_1, v_0)$, such that $\mathbb{E}\left(\epsilon_n^{\eta_0} g(\theta_n)\right) < T(\theta_1, v_0)$ for any $n \geq 1$.*

**Lemma B.9.** *Suppose $\{v_n\}$ is a sequence generated by mSGD in* (4). *Under Assumptions 3.1–3.2, it holds that $\sum_{n=0}^{+\infty} \mathbb{E}\left(\epsilon_n^{2\eta_0} \|v_n\|^2\right) < c(v_0, \theta_1)$, and $\sum_{n=0}^{+\infty} \epsilon_n^{2\eta_0} \|v_n\|^2 < +\infty$ a.s., where $c(v_0, \theta_1)$ is a constant only related to $v_0$ and $\theta_1$.*

**Lemma B.10.** *Suppose $\{\theta_n\}$ is a sequence generated by mSGD in* (4). *Under Assumptions 3.1–3.2, it holds that: for $n \geq 1$,*

$$\sum_{t=1}^{n} \epsilon_t^{1+2\eta_0} \mathbb{E}\left(\|\nabla_{\theta_t} g(\theta_t)\|^2\right) < B(v_0, \theta_1) < +\infty, \qquad \sum_{t=1}^{n} \epsilon_t^{1+2\eta_0} \|\nabla_{\theta_t} g(\theta_t)\|^2 < +\infty,$$

*where $B(v_0, \theta_1) > 0$ is a constant only related to $v_0$ and $\theta_1$.*

**Lemma B.11.** *Suppose $\{\theta_n\}$ is a sequence generated by mSGD in* (4). *Under Assumptions 3.1–3.2, it holds that: for $n \geq 1$,*

$$g(\theta_{n+1}) - g(\theta_n) \leq \hat{k}\epsilon_t^{1+2\eta_0} + Q_n, \tag{8}$$

*where $\hat{k} > 0$ is a constant and $\{Q_n\}$ is a sequence, such that $\sum_{n=1}^{+\infty} Q_n$ exists and is finite a.s..*

## B.1  Proof of Lemma B.6

The proof of this lemma needs some identical transformations. We assume $\mu > \sigma$ (the case $\mu < \sigma$ is the same), and let $\omega_0 = \log_\mu \sigma > 1$. Then we derive

$$\sum_{k=1}^{n} \mu^{n-k} \sum_{i=1}^{k} \sigma^{k-i}\psi_i = \sum_{k=1}^{n} \sum_{i=1}^{k} \mu^{n-k}\sigma^{k-i}\psi_i = \sum_{i=1}^{n} \sum_{k=i}^{n} \mu^{n-k}\sigma^{k-i}\psi_i = \sum_{i=1}^{n} \sum_{k=i}^{n} \mu^{n-k+\omega_0(k-i)}\psi_i$$

$$= \sum_{i=1}^{n} \mu^{n-\omega_0 i} \sum_{k=i}^{n} \mu^{(\omega_0-1)k}\psi_i = \sum_{i=1}^{n} \mu^{n-\omega_0 i} \frac{1 - \mu^{(\omega_0-1)(n-i+1)}}{1 - \mu^{(\omega_0-1)}} \mu^{(\omega_0-1)i}\psi_i := \hat{R}\sum_{i=1}^{n} \mu^{n-i}\psi_i,$$

where $1 < \hat{R} < 1/\left(1 - \kappa^{(\omega_0-1)}\right)$, and $\kappa = \max\{\mu, \sigma\}$ and $\omega_0 = \log_\kappa \min\{\mu, \sigma\}$.

## B.2  Proof of Lemma B.7

We calculate $\sum_{t=1}^{n} c_t^{(\hat{\epsilon})}$ as follows

$$\sum_{t=1}^{n} c_t^{(\hat{\epsilon})} = \sum_{t=1}^{n} \left(\sum_{i=1}^{t} \tau^{t-i}\hat{\epsilon}_i\right) - \sum_{t=1}^{n} \frac{1}{1-\tau}\hat{\epsilon}_t = \sum_{i=1}^{n} \left(\sum_{t=i}^{n} \tau^{t-i}\right)\hat{\epsilon}_i - \sum_{t=1}^{n} \frac{1}{1-\tau}\hat{\epsilon}_t$$

$$= \sum_{t=1}^{n} \left(\sum_{i=t}^{n} \tau^{i-t}\right)\hat{\epsilon}_t - \sum_{t=1}^{n} \frac{1}{1-\tau}\hat{\epsilon}_t = \sum_{t=1}^{n} \left(\sum_{i=0}^{n-t} \tau^{i}\right)\hat{\epsilon}_t - \sum_{t=1}^{n} \frac{1}{1-\tau}\hat{\epsilon}_t$$

$$= \sum_{t=1}^{n} \left(\frac{1 - \tau^{n-t+1}}{1-\tau}\right)\hat{\epsilon}_t - \sum_{t=1}^{n} \frac{1}{1-\tau}\hat{\epsilon}_t = -\frac{\tau}{1-\tau}\sum_{t=1}^{n} \tau^{n-t}\hat{\epsilon}_t.$$

So we just need to prove $\lim_{n\to+\infty} \left|\sum_{t=1}^{n} \tau^{n-t}\hat{\epsilon}_t\right| = 0$. It holds that

$$\left|\sum_{t=1}^{n} \tau^{n-t}\hat{\epsilon}_t\right| = \left|\left(\sum_{t=1}^{n-[\sqrt{n}]} + \sum_{t=[n-\sqrt{n}]+1}^{n}\right)\tau^{n-t}\hat{\epsilon}_t\right| \leq \left|\sum_{t=1}^{n-[\sqrt{n}]} \tau^{n-t}\hat{\epsilon}_t\right| + \left|\sum_{t=[n-\sqrt{n}]+1}^{n} \tau^{n-t}\hat{\epsilon}_t\right|.$$

Due to $n - \sqrt{n} \to +\infty$, $\forall \sigma_0 > 0$, $\exists N_0 > 0$, when $t > N_0 - [\sqrt{N_0}]$, it holds that $|\hat{\epsilon}_t| < \sigma_0$. So when $n > N_0$, it holds that

$$\left|\sum_{t=n-[\sqrt{n}]+1}^{n} \tau^{n-t}\hat{\epsilon}_t\right| < \sum_{t=n-[\sqrt{n}]+1}^{n} \tau^{n-t}\sigma_0 < \sigma_0 \sum_{t=0}^{+\infty} \tau^t = \frac{\sigma_0}{1-\tau}.$$

On the other hand

$$\left|\sum_{t=1}^{n-[\sqrt{n}]} \tau^{n-t}\hat{\epsilon}_t\right| < \tilde{k}_0 \sum_{t=1}^{n-[\sqrt{n}]} \tau^{n-t} = \tilde{k}_0 \sum_{t=[\sqrt{n}]}^{n-1} \tau^t \to 0,$$

where $\tilde{k}_0 > 0$ is a constant. As a result, $\forall \, \sigma_0 > 0$, $\exists \, N_1 > 1$, when $n > N_1$, it holds that $\left| \sum_{t=1}^{n-[\sqrt{n}]} \tau^{n-t} \hat{\epsilon}_t \right| < \sigma_0$. Then when $n > \max\{N_0, N_1\}$, it holds that

$$\left| \sum_{t=1}^{n} \tau^{n-t} \hat{\epsilon}_t \right| \le \left| \sum_{t=1}^{n-[\sqrt{n}]} \tau^{n-t} \hat{\epsilon}_t \right| + \left| \sum_{t=[n-\sqrt{n}]+1}^{n} \tau^{n-t} \hat{\epsilon}_t \right| < \frac{2-\tau}{1-\tau} \sigma_0.$$

Due to the arbitrariness of $\sigma_0$, we know

$$\sum_{t=1}^{n} \tau^{n-t} \hat{\epsilon}_t \to 0.$$

Thus, it holds that $\sum_{t=1}^{n} c_t^{(\hat{\epsilon})} = 0$.

## B.3 Proof of Lemma B.8

Recall the mSGD algorithm in (4)

$$v_n = \alpha v_{n-1} + \epsilon_n \nabla_{\theta_n} g(\theta_n, \xi_n) \tag{9}$$
$$\theta_{n+1} = \theta_n - v_n. \tag{10}$$

Equation (9) is equivalent to

$$v_n = \alpha v_{n-1} + \epsilon_n \nabla_{\theta_n} g(\theta_n) + \epsilon_n \big( \nabla_{\theta_n} g(\theta_n, \xi_n) - \nabla_{\theta_n} g(\theta_n) \big).$$

Under Assumption 3.1 4), it follows from Lemma B.1 that

$$- \nabla_{\theta_t} g(\theta_t)^\mathsf{T} v_t - \frac{c}{2} \|v_t\|^2 \le g(\theta_{t+1}) - g(\theta_t) \le - \nabla_{\theta_t} g(\theta_t)^\mathsf{T} v_t + \frac{c}{2} \|v_t\|^2. \tag{11}$$

In this subsection, we just use the right side of (11). The left side will be used in the next subsection. Consider $\nabla_{\theta_t} g(\theta_t)^\mathsf{T} v_t$ in the following

$$
\begin{aligned}
\nabla_{\theta_t} g(\theta_t)^\mathsf{T} v_t &= (\nabla_{\theta_t} g(\theta_t))^\mathsf{T} (\alpha v_{t-1} + \epsilon_t \nabla_{\theta_n} g(\theta_t, \xi_t)) \\
&= \alpha (\nabla_{\theta_{t-1}} g(\theta_{t-1}) + \nabla_{\theta_t} g(\theta_t) - \nabla_{\theta_{t-1}} g(\theta_{t-1}))^\mathsf{T} v_{t-1} + \epsilon_t \nabla_{\theta_t} g(\theta_t)^\mathsf{T} \nabla_{\theta_t} g(\theta_t, \xi_t) \\
&= \alpha \nabla_{\theta_{t-1}} g(\theta_{t-1})^\mathsf{T} v_{t-1} + \alpha (\nabla_{\theta_t} g(\theta_t) - \nabla_{\theta_{t-1}} g(\theta_{t-1}))^T v_{t-1} + \epsilon_t \nabla_{\theta_t} g(\theta_t)^\mathsf{T} \nabla_{\theta_t} g(\theta_t, \xi_t).
\end{aligned}
$$

Recursively applying the above equation yields

$$
\begin{aligned}
\nabla_{\theta_t} g(\theta_t)^\mathsf{T} v_t &= \alpha^{t-1} \nabla_{\theta_1} g(\theta_1)^\mathsf{T} v_1 + \sum_{i=1}^{t-1} \alpha^{t-i} (\nabla_{\theta_{i+1}} g(\theta_{i+1}) - \nabla_{\theta_i} g(\theta_i))^T v_i \\
&\quad + \sum_{i=2}^{t} \alpha^{t-i} \epsilon_i \nabla_{\theta_i} g(\theta_i)^\mathsf{T} \nabla_{\theta_i} g(\theta_i, \xi_i).
\end{aligned} \tag{12}
$$

Substitute (12) into (11), then we have

$$
\begin{aligned}
&\mathbb{E} \big( g(\theta_{t+1}) - g(\theta_t) \big) \\
&- \le \alpha^{t-1} \mathbb{E} \big( \nabla_{\theta_1} g(\theta_1)^\mathsf{T} v_1 \big) + c \sum_{i=1}^{t} \alpha^{t-i} \mathbb{E} \big( \|v_i\|^2 \big) - \sum_{i=2}^{t} \alpha^{t-i} \epsilon_i \, \mathbb{E} \left\| \nabla_{\theta_i} g(\theta_i) \right\|^2 \\
&< -\alpha^{t-1} \mathbb{E} \big( \nabla_{\theta_1} g(\theta_1)^\mathsf{T} v_1 \big) + c \sum_{i=1}^{t} \alpha^{t-i} \mathbb{E} \big( \|v_i\|^2 \big).
\end{aligned} \tag{13}
$$

From Lemma B.3, it follows that $\mathbb{E}_{\xi_n} \big( \|\nabla_{\theta_n} g(\theta_n, \xi_n)\|^2 \big) \le M' \|\nabla_{\theta_n} g(\theta_n)\|^2 + a' \le 2cM' \big( g(\theta_n) - g^* \big) + a' \le M(g(\theta_n) + 1)$ *(M always exists)*. Given $\forall \, \eta_0 > 0$, we denote $Z(t) := \prod_{k=t}^{+\infty} (1 + M_0 \epsilon_k^{2+\eta_0})$, where

$$M_0 = \frac{cM}{\alpha^{1-\delta}(1-\alpha^\delta)(1-\alpha)^2},$$

in which $\delta > 0$ is a constant and $M$ is defined above. Here we define $M_0$, $Z(t)$ and $\delta$ to ease the proof. From Assumption 3.2, it holds that $\sum_{t=1}^{+\infty} \epsilon_t^{2+\eta_0} < +\infty$, which ensures $\sum_{t=1}^{+\infty} M_0 \epsilon_t^{2+\eta_0} < +\infty$. From a general inequality $\ln(1+\theta) \le \theta$ for $\theta > -1$, we get

$$Z(t) \le \prod_{k=1}^{+\infty} (1 + M_0 \epsilon_k^{2+\eta_0}) = \exp \left\{ \sum_{k=1}^{\infty} \ln(1 + M_0 \epsilon_k^{2+\eta_0}) \right\} \le \exp \left\{ \sum_{k=1}^{\infty} M_0 \epsilon_k^{2+\eta_0} \right\} < +\infty,$$

which means that $Z(t)$ is uniformly upper bounded. Then multiplying $Z(t+1)$ on both sides of (13), taking the mathematical expectation and noting $\epsilon_t \geq \epsilon_{t+1}$ yield

$$Z(t+1)\,\mathbb{E}\left(\epsilon_{t+1}^{\eta_0}g(\theta_{t+1}) - \epsilon_t^{\eta_0}g(\theta_t)\right)$$

$$< -Z(t+1)\alpha^{t-1}\epsilon_t^{\eta_0}\,\mathbb{E}\left(\nabla_{\theta_1}g(\theta_1)^\mathsf{T}v_1\right) + c\sum_{i=1}^{t}\alpha^{t-i}Z(i+1)\,\mathbb{E}\left(\epsilon_i^{\eta_0}\|v_i\|^2\right), \tag{14}$$

where the inequality is due to $\epsilon_{t+1} < \epsilon_t$ and $Z(i+1) < Z(i)$.

Next, we aim to analyze $c\sum_{i=1}^{t}\alpha^{t-i}Z(i+1)\,\mathbb{E}\left(\epsilon_t^{\eta_0}\|v_i\|^2\right)$ in (14). It is proved in Appendix B.4 that

$$Z(i+1)\,\mathbb{E}\left(\epsilon_t^{\eta_0}\|v_i\|^2\right)$$

$$\leq \alpha^{(1+\delta)i}Z(1)\epsilon_t^{\eta_0}\,\mathbb{E}\|v_0\|^2 + \frac{1}{\alpha^{1-\delta}}\sum_{k=1}^{i}\alpha^{(1+\delta)(i-k)}\epsilon_k^{2+\eta_0}Z(k+1)\,\mathbb{E}\left(\|\nabla_{\theta_k}g(\theta_k) - \nabla_{\theta_k}g(\theta_k,\xi_k)\|^2\right)$$

$$-\frac{2}{\alpha^{1-\delta}}\sum_{k=1}^{i}\alpha^{(1+\delta)(i-k)}Z(k+1)\left(\mathbb{E}\left(\epsilon_{k+1}^{1+\eta_0}g(\theta_{k+1})\right) - \mathbb{E}\left(\epsilon_k^{1+\eta_0}g(\theta_k)\right)\right). \tag{15}$$

Taking a weighted sum of (15) yields

$$\sum_{i=1}^{t}\alpha^{t-i}Z(i+1)\,\mathbb{E}\left(\epsilon_i^{\eta_0}\|v_i\|^2\right)$$

$$\leq \sum_{i=1}^{t}\alpha^{t-i}\alpha^{(1+\delta)i}Z(1)\,\mathbb{E}\left(\epsilon_0^{\eta_0}\|v_0\|^2\right)$$

$$+\sum_{i=1}^{t}\alpha^{t-i}\left(\frac{1}{\alpha^{1-\delta}}\sum_{k=1}^{i}\alpha^{(1+\delta)(i-k)}\epsilon_k^{2+\eta_0}Z(k+1)\,\mathbb{E}\left(\|\nabla_{\theta_k}g(\theta_k) - \nabla_{\theta_k}g(\theta_k,\xi_k)\|^2\right)\right) \tag{16}$$

$$-\sum_{i=1}^{t}\alpha^{t-i}\left(\frac{2}{\alpha^{1-\delta}}\sum_{k=1}^{i}\alpha^{(1+\delta)(i-k)}Z(k+1)\left(\mathbb{E}\left(\epsilon_{k+1}^{1+\eta_0}g(\theta_{k+1})\right) - \mathbb{E}\left(\epsilon_k^{1+\eta_0}g(\theta_k)\right)\right)\right)$$

$$:= A + B + C.$$

Derive that

$$A = \left(\sum_{i=1}^{t}\alpha^{\delta i}\right)\alpha^t Z(1)\,\mathbb{E}\left(\epsilon_0^{\eta_0}\|v_0\|^2\right) \leq \frac{\alpha^t\alpha^\delta}{1-\alpha^\delta}Z(1)\,\mathbb{E}\left(\|v_0\|^2\right), \tag{17}$$

$$B = \frac{1}{\alpha^{1-\delta}}\sum_{i=1}^{t}\sum_{k=1}^{i}\alpha^{t-k+\delta(i-k)}\left(\epsilon_k^{2+\eta_0}Z(k+1)\,\mathbb{E}\left(\|\nabla_{\theta_k}g(\theta_k) - \nabla_{\theta_k}g(\theta_k,\xi_k)\|^2\right)\right)$$

$$\leq \frac{1}{\alpha^{1-\delta}(1-\alpha^\delta)}\sum_{k=1}^{t}\alpha^{t-k}Z(k+1)\epsilon_k^{2+\eta_0}\,\mathbb{E}\left(\|\nabla_{\theta_k}g(\theta_k) - \nabla_{\theta_k}g(\theta_k,\xi_k)\|^2\right), \tag{18}$$

$$C = -\frac{2}{\alpha^{1-\delta}}\sum_{k=1}^{t}\sum_{i=k}^{t}\alpha^{t-k+\delta(i-k)}Z(k+1)\left(\mathbb{E}\left(\epsilon_{k+1}^{1+\eta_0}g(\theta_{k+1})\right) - \mathbb{E}\left(\epsilon_k^{1+\eta_0}g(\theta_k)\right)\right)$$

$$= -\frac{2}{\alpha^{1-\delta}}\sum_{k=1}^{t}\left(\sum_{i=0}^{t-k}\alpha^{\delta i}\right)\alpha^{t-k}Z(k+1)\left(\mathbb{E}\left(\epsilon_{k+1}^{1+\eta_0}g(\theta_{k+1})\right) - \mathbb{E}\left(\epsilon_k^{1+\eta_0}g(\theta_k)\right)\right). \tag{19}$$

Substituting (17)–(19) into (14) yields

$$Z(t+1)\,\mathbb{E}\left(\left(\epsilon_{t+1}^{1+\eta_0}g(\theta_{t+1}) - \epsilon_t^{1+\eta_0}g(\theta_t)\right)\right)$$

$$\leq -Z(t+1)\alpha^{t-1}\epsilon_t^{1+\eta_0}\,\mathbb{E}\left(\nabla_{\theta_1}g(\theta_1)^\mathsf{T}v_1\right) + \frac{\alpha^{t-1}c\alpha^\delta}{1-\alpha^\delta}Z(1)\,\mathbb{E}\left(\epsilon_0^{1+\eta_0}\|v_0\|^2\right)$$

$$+\frac{c}{\alpha^{1-\delta}(1-\alpha^\delta)}\sum_{i=1}^{t}\alpha^{t-i}Z(i+1)\epsilon_i^{2+\eta_0}\,\mathbb{E}\left(\|\nabla_{\theta_i}g(\theta_i) - \nabla_{\theta_i}g(\theta_i,\xi_i)\|^2\right) \tag{20}$$

$$-\frac{2c}{\alpha^{1-\delta}}\sum_{i=1}^{t}\left(\sum_{k=0}^{t-i}\alpha^{\delta k}\right)\alpha^{t-i}Z(i+1)\left(\mathbb{E}\left(\epsilon_{i+1}^{1+\eta_0}g(\theta_{i+1})\right) - \mathbb{E}\left(\epsilon_i^{1+\eta_0}g(\theta_i)\right)\right).$$

Construct a sequence $\{V_n\}$ as follows

$$V_n = \sum_{t=1}^{n} \left(\frac{1}{2-\alpha}\right)^{n-t} Z(t+1) \, \mathbb{E}\left(\left(\epsilon_{t+1}^{1+\eta_0} g(\theta_{t+1}) - \epsilon_t^{1+\eta_0} g(\theta_t)\right)\right). \tag{21}$$

By substituting (20) into (21) following the way of (17)–(19), we have

$$\begin{aligned}
v_n \leq & \frac{\alpha^n (2-\alpha)}{1-\alpha} Z(1) \epsilon_0^{1+\eta_0} \left| \mathbb{E}\left(\nabla_{\theta_1} g(\theta_1)^{\mathsf{T}} v_1\right) \right| + \frac{c \alpha^n \alpha^\delta (2-\alpha)}{(1-\alpha)(1-\alpha^\delta)} Z(1) \, \mathbb{E}\left(\epsilon_0^{1+\eta_0} \|v_0\|^2\right) \\
& + \frac{c}{\alpha^{1-\delta}(1-\alpha^\delta)(1-\alpha)^2} \sum_{t=1}^{n} \left(\frac{1}{2-\alpha}\right)^{n-i} Z(t+1) \epsilon_t^{2+\eta_0} \, \mathbb{E}\left(\|\nabla_{\theta_t} g(\theta_t) - \nabla_{\theta_t} g(\theta_t, \xi_t)\|^2\right) \\
& - \frac{2c}{\alpha^{1-\delta}(1-\alpha^\delta)} \sum_{t=1}^{n} f(n-t) \left(\frac{1}{2-\alpha}\right)^{n-t} Z(t+1) \left(\mathbb{E}\left(\epsilon_{t+1}^{1+\eta_0} g(\theta_{t+1})\right) - \mathbb{E}\left(\epsilon_t^{1+\eta_0} g(\theta_t)\right)\right),
\end{aligned}$$

where $f(n-t)$ is defined as follows,

$$f(n-t) = \sum_{k=1}^{n-t} \left(\alpha(2-\alpha)\right)^k - \alpha^\delta \sum_{k=1}^{n-t} \left(\alpha^{1+\delta}(2-\alpha)\right)^k.$$

Move the last term to the left-hand side of the above inequality, then we have

$$\begin{aligned}
& \sum_{t=1}^{n} \left(\frac{1}{2-\alpha}\right)^{n-t} Z(t+1) \, \mathbb{E}\left(e_{t+1}^{(n)} g(\theta_{t+1}) - e_t^{(n-1)} g(\theta_t)\right) \\
& \leq \frac{\alpha^n (2-\alpha)}{1-\alpha} Z(1) \left| \mathbb{E}\left(\epsilon_0^{\eta_0} \nabla_{\theta_1} g(\theta_1)^{\mathsf{T}} v_1\right) \right| + \frac{c \alpha^n \alpha^\delta (2-\alpha)}{(1-\alpha)(1-\alpha^\delta)} Z(1) \, \mathbb{E}\left(\epsilon_0^{\eta_0} \|v_0\|^2\right) \\
& + \frac{c}{\alpha^{1-\delta}(1-\alpha^\delta)(1-\alpha)^2} \sum_{t=1}^{n} \left(\frac{1}{2-\alpha}\right)^{n-t} Z(t+1) \epsilon_t^{2+\eta_0} \, \mathbb{E}\left(\|\nabla_{\theta_t} g(\theta_t) - \nabla_{\theta_t} g(\theta_t, \xi_t)\|^2\right),
\end{aligned} \tag{22}$$

where

$$e_t^{(n-1)} = \left(\epsilon_t^{\eta_0} + \frac{2c \epsilon_t^{1+\eta_0} f(n-t)}{\alpha^{1-\delta}(1-\alpha^\delta)}\right). \tag{23}$$

Because of $\alpha < 1$, it holds that $f(n-t) > 0$ and $e_t^{(n-1)} > 1$. It follows from Assumption 3.2 that

$$\begin{aligned}
& \frac{c}{\alpha^{1-\delta}(1-\alpha^\delta)(1-\alpha)^2} \sum_{t=1}^{n} \left(\frac{1}{2-\alpha}\right)^{n-t} Z(t+1) \epsilon_t^{2+\eta_0} \, \mathbb{E}\left(\|\nabla_{\theta_t} g(\theta_t) - \nabla_{\theta_t} g(\theta_t, \xi_t)\|^2\right) \\
& \leq \sum_{t=1}^{n} \left(\frac{1}{2-\alpha}\right)^{n-t} Z(t+1) M_0 \epsilon_t^{2+\eta_0} e_t^{(n-1)} \left(1 + \mathbb{E}(g(\theta_t))\right),
\end{aligned} \tag{24}$$

where

$$M_0 = \frac{cM}{\alpha^{1-\delta}(1-\alpha^\delta)(1-\alpha)^2}.$$

Calculate $f(n-t)$, then we obtain

$$f(n-t) = \sum_{k=1}^{n-t} \left(\alpha(2-\alpha)\right)^k - \alpha^\delta \sum_{k=1}^{n-t} \left(\alpha^{1+\delta}(2-\alpha)\right)^k > 0.$$

It holds that

$$\sum_{t=1}^{n}\left(\frac{1}{2-\alpha}\right)^{n-t}Z(t+1)\,\mathbb{E}\left(e_{t+1}^{(n)}g(\theta_{t+1})-e_t^{(n-1)}g(\theta_t))\right)$$

$$-\sum_{t=1}^{n}\left(\frac{1}{2-\alpha}\right)^{n-t}Z(t+1)M_0\epsilon_t^{2+\eta_0}e_t^{(n-1)}\big(1+\mathbb{E}(g(\theta_t))\big)$$

$$=\sum_{t=1}^{n}\left(\frac{1}{2-\alpha}\right)^{n-t}Z(t+1)\,\mathbb{E}\left(e_{t+1}^{(n)}\big(1+g(\theta_{t+1})\big)\right)$$

$$-\sum_{t=1}^{n}\left(\frac{1}{2-\alpha}\right)^{n-t}Z(t+1)(1+M_0\epsilon_t^{2+\eta_0})\,\mathbb{E}\left(e_t^{(n-1)}\big(1+g(\theta_t)\big)\right) \tag{25}$$

$$-\frac{c}{\alpha^{1-\delta}(1-\alpha^\delta)}\sum_{t=1}^{n}f(n-t)\left(\frac{1}{2-\alpha}\right)^{n-t}(\epsilon_{t+1}^{1+\eta_0}-\epsilon_t^{1+\eta_0})$$

$$>\sum_{t=1}^{n}\left(\frac{1}{2-\alpha}\right)^{n-t}Z(t+1)\,\mathbb{E}\left(e_{t+1}^{(n)}\big(1+g(\theta_{t+1})\big)\right)$$

$$-\sum_{t=1}^{n}\left(\frac{1}{2-\alpha}\right)^{n-t}Z(t+1)(1+M_0\epsilon_t^{2+\eta_0})\,\mathbb{E}\left(e_t^{(n-1)}\big(1+g(\theta_t)\big)\right).$$

Substituting (24) and (25) into (22) yields

$$\sum_{t=1}^{n}\left(\frac{1}{2-\alpha}\right)^{n-t}Z(t+1)\,\mathbb{E}\left(e_{t+1}^{(n)}\big(1+g(\theta_{t+1})\big)\right)$$

$$-\sum_{t=1}^{n}\left(\frac{1}{2-\alpha}\right)^{n-t}Z(t+1)(1+M_0\epsilon_t^{2+\eta_0})\,\mathbb{E}\left(e_t^{(n-1)}\big(1+g(\theta_t)\big)\right) \tag{26}$$

$$\leq\frac{\alpha^n(2-\alpha)}{1-\alpha}Z(1)\epsilon_0^{1+\eta_0}\left|\mathbb{E}\left(\nabla_{\theta_1}g(\theta_1)^{\mathsf{T}}v_1\right)\right|+\frac{c\alpha^n\alpha^\delta(2-\alpha)}{2(1-\alpha)(1-\alpha^\delta)}Z(1)\,\mathbb{E}\left(\epsilon_0^{1+\eta_0}\|v_0\|^2\right).$$

According to the definition of $Z(t)$, we have

$$Z(t)=\prod_{k=t}^{+\infty}(1+M_0\epsilon_k^2)=(1+M_0\epsilon_t^{2+\eta_0})\prod_{k=t+1}^{+\infty}(1+M_0\epsilon_k^{2+\eta_0})=(1+M_0\epsilon_t^{2+\eta_0})Z(t+1).$$

Let

$$F_n:=\sum_{t=1}^{n}\left(\frac{1}{2-\alpha}\right)^{n-t}Z(t+1)\,\mathbb{E}\left(e_{t+1}^{(n)}\big(1+g(\theta_{t+1})\big)\right),$$

and it follows from (26) that

$$F_n-F_{n-1}\leq\widetilde{k}_0(\theta_1,v_0)\left(\left(\frac{1}{2-\alpha}\right)^{n-1}+\alpha^n\right), \tag{27}$$

where $\widetilde{k}_0(\theta_1,v_0)$. Denote $p=\mathbb{E}\left\{\sum_{k=1}^{\infty}M_0\epsilon_k^{2+\eta_0}\right\}$. By taking the summation of (27), we obtain

$$F_n\leq F_1+\widetilde{k}_0\sum_{t=1}^{+\infty}\left(\left(\frac{1}{2-\alpha}\right)^{n-1}+\alpha^n\right).$$

Thus, we derive that

$$\epsilon_{n+1}^{\eta_0}\,\mathbb{E}\left(g(\theta_{n+1})\right)<e_{n+1}^n\,\mathbb{E}\left(1+g(\theta_{n+1})\right)<p\left(\frac{1}{2-\alpha}\right)^{n-n}Z(n+1)\,\mathbb{E}\left(e_{n+1}^{(n)}\big(1+g(\theta_{n+1})\big)\right)$$

$$<p\sum_{t=1}^{n}\left(\frac{1}{2-\alpha}\right)^{n-t}Z(t+1)\,\mathbb{E}\left(e_{t+1}^{(n)}\big(1+g(\theta_{t+1})\big)\right)$$

$$<pF_n<T(\theta_1,v_0),$$

where $T(\theta_1,v_0)$ is a constant determined by $v_0$ and $\theta_1$.

## B.4 Proof of (15)

We consider

$$
\begin{aligned}
&\epsilon_{i+1}^{1+\eta_0} g(\theta_{i+1}) - \epsilon_i^{1+\eta_0} g(\theta_i) \\
&= \epsilon_i^{1+\eta_0}\big(g(\theta_{i+1}) - g(\theta_i)\big) + (\epsilon_{i+1}^{1+\eta_0} - \epsilon_i^{1+\eta_0})g(\theta_i) \\
&\leq \epsilon_i^{1+\eta_0}\big(g(\theta_{i+1}) - g(\theta_i)\big) \\
&\leq -\epsilon_i^{1+\eta_0}\nabla_{\theta_i} g(\theta_i)^{\mathsf{T}} v_i + \frac{c}{2}\epsilon_i^{1+\eta_0}\|v_i\|^2 \\
&= (-\epsilon_i^{1+\eta_0}\nabla_{\theta_i} g(\theta_i,\xi_i)^{\mathsf{T}} v_i - (\epsilon_i^{1+\eta_0}\nabla_{\theta_i} g(\theta_i) - \epsilon_i^{1+\eta_0}\nabla_{\theta_i} g(\theta_i,\xi_i))^{\mathsf{T}} v_i + \frac{c}{2}\epsilon_i^{1+\eta_0}\|v_i\|^2 \\
&= \alpha\epsilon_i^{1+\eta_0} v_i^{\mathsf{T}} v_{i-1} - \epsilon_i^{1+\eta_0}\|v_i\|^2 + \epsilon_i^{2+\eta_0}\big\|\nabla_{\theta_i} g(\theta_i) - \nabla_{\theta_i} g(\theta_i,\xi_i)\big\|^2 \\
&\quad - \big(\epsilon_i^{1+\eta_0}\alpha v_{i-1} + \epsilon_i^{2+\eta_0}\nabla_{\theta_i} g(\theta_i)\big)^{\mathsf{T}}\big(\nabla_{\theta_i} g(\theta_i) - \nabla_{\theta_i} g(\theta_i,\xi_i)\big) + \frac{c}{2}\epsilon_i^{1+\eta_0}\|v_i\|^2,
\end{aligned}
\tag{28}
$$

where the first inequality is due to $\epsilon_i \leq \epsilon_{i-1}$ in Assumption 3.2, and the last equality is from (10). Since $\xi_i$ and $\theta_i$ are independent, taking the mathematical expectation of (28) and noting that

$$
\mathbb{E}\left(\big(\epsilon_i^{1+\eta_0}\alpha v_{i-1} + \epsilon_i^{2+\eta_0}\nabla_{\theta_i} g(\theta_i)\big)^{\mathsf{T}}\big(\nabla_{\theta_i} g(\theta_i) - \nabla_{\theta_i} g(\theta_i,\xi_i))\big)\right) = 0,
$$

yield

$$
\begin{aligned}
&\mathbb{E}\left(\epsilon_{i+1}^{1+\eta_0} g(\theta_{i+1})\right) - \mathbb{E}\left(\epsilon_i^{1+\eta_0} g(\theta_i)\right) \\
&\leq \alpha\epsilon_i^{\eta_0}\mathbb{E}(v_i^{\mathsf{T}} v_{i-1}) - \epsilon_i^{\eta_0}\mathbb{E}\left(\|v_i\|^2\right) + \epsilon_i^{2+\eta_0}\mathbb{E}\left(\big\|\nabla_{\theta_i} g(\theta_i) - \nabla_{\theta_i} g(\theta_i,\xi_i)\big\|^2\right) + \frac{c}{2}\epsilon_i^{1+\eta_0}\mathbb{E}\left(\|v_i\|^2\right).
\end{aligned}
\tag{29}
$$

Moreover, it holds that

$$
\begin{aligned}
&\mathbb{E}\left(\big\|\nabla_{\theta_i} g(\theta_i) - \nabla_{\theta_i} g(\theta_i,\xi_i)\big\|^2\right) \\
&= \mathbb{E}\left(\big\|\nabla_{\theta_i} g(\theta_i,\xi_i)\big\|^2\right) - \mathbb{E}\left(\big\|\nabla_{\theta_i} g(\theta_i)\big\|^2\right) \\
&= \frac{1}{\epsilon_i}\mathbb{E}\left(\|v_i - \alpha v_{i-1}\|^2\right) - \mathbb{E}\left(\big\|\nabla_{\theta_i} g(\theta_i)\big\|^2\right) \\
&= \frac{1}{\epsilon_i}\left(\mathbb{E}\left(\|v_i\|^2\right) + \alpha^2\mathbb{E}\left(\|v_{i-1}\|^2\right) - 2\alpha\mathbb{E}(v_i^{\mathsf{T}} v_{i-1})\right) - \mathbb{E}\left(\big\|\nabla_{\theta_i} g(\theta_i)\big\|^2\right).
\end{aligned}
\tag{30}
$$

Combining (29) and (30), we get

$$
\begin{aligned}
&\mathbb{E}\left(\epsilon_{i+1}^{1+\eta_0} g(\theta_{i+1})\right) - \mathbb{E}\left(\epsilon_i^{1+\eta_0} g(\theta_i)\right) \\
&\leq -\frac{1}{2}\left(\mathbb{E}\left(\epsilon_i^{1+\eta_0}\|v_i\|^2\right) - \alpha^2\mathbb{E}\left(\epsilon_{i-1}^{1+\eta_0}\|v_{i-1}\|^2\right)\right) - \frac{\epsilon_i^{2+\eta_0}}{2}\mathbb{E}\left(\big\|\nabla_{\theta_i} g(\theta_i)\big\|^2\right) + \frac{c}{2}\epsilon_i^{1+\eta_0}\mathbb{E}\left(\|v_i\|^2\right) \\
&\quad + \frac{\epsilon_i^{2+\eta_0}}{2}\mathbb{E}\left(\big\|\nabla_{\theta_i} g(\theta_i) - \nabla_{\theta_i} g(\theta_i,\xi_i)\big\|^2\right).
\end{aligned}
\tag{31}
$$

Since $\epsilon_i \to 0$, given any $\delta > 0$, there is an integer $i_0 \geq 0$, such that for $i \geq i_0$, $1 - c\epsilon_i > \alpha^{1-\delta}(\delta > 0)$. Since $i_0$ is finite, without loss of generality, we assume $i_0 = 0$ for convenience, i.e., $1 - c\epsilon_i > \alpha^{1-\delta}(\delta > 0)(i \geq 1)$. Thus, we have

$$
\begin{aligned}
&\mathbb{E}\left(\epsilon_{i+1}^{1+\eta_0} g(\theta_{i+1})\right) - \mathbb{E}\left(\epsilon_i^{1+\eta_0} g(\theta_i)\right) \\
&\leq -\frac{\alpha^{1-\delta}}{2}\left(\mathbb{E}\left(\epsilon_i^{1+\eta_0}\|v_i\|^2\right) - \alpha^{1+\delta}\mathbb{E}\left(\epsilon_{i-1}^{1+\eta_0}\|v_{i-1}\|^2\right)\right) + \frac{\epsilon_i^{2+\eta_0}}{2}\mathbb{E}\left(\big\|\nabla_{\theta_i} g(\theta_i) - \nabla_{\theta_i} g(\theta_i,\xi_i)\big\|^2\right).
\end{aligned}
\tag{32}
$$

Multiplying both sides of (32) by $Z(i+1)$, and noticing that $Z(i) > Z(i+1)$, we have

$$
\begin{aligned}
&\left(Z(i+1)\mathbb{E}\left(\epsilon_i^{\eta_0}\|v_i\|^2\right) - \alpha^{1+\delta} Z(i)\mathbb{E}\left(\epsilon_{i-1}^{\eta_0}\|v_{i-1}\|^2\right)\right) \\
&\leq -\frac{2}{\alpha^{1-\delta}} Z(i+1)\left(\mathbb{E}\left(\epsilon_{i+1}^{1+\eta_0} g(\theta_{i+1})\right) - \mathbb{E}\left(\epsilon_i^{1+\eta_0} g(\theta_i)\right)\right) + \frac{\epsilon_i^{2+\eta_0}}{\alpha^{1-\delta}} Z(i+1)\mathbb{E}\left(\big\|\nabla_{\theta_i} g(\theta_i) - \nabla_{\theta_i} g(\theta_i,\xi_i)\big\|^2\right).
\end{aligned}
$$

Then (15) is obtained by recursively applying the above inequality.

## B.5 Proof of Lemma B.9

From (31), we have

$$
\mathbb{E}\left(\epsilon_{n+1}^{1+2\eta_0} g(\theta_{n+1})\right) - \mathbb{E}\left(\epsilon_1^{1+2\eta_0} g(\theta_1)\right)
$$
$$
< -\frac{1}{2}\sum_{t=1}^n \mathbb{E}\left(\epsilon_t^{2\eta_0}\|v_t\|^2\right) + \frac{\alpha^2}{2}\sum_{t=1}^n \mathbb{E}\left(\epsilon_{t-1}^{2\eta_0}\|v_{t-1}\|^2\right) - \sum_{t=1}^n \frac{\epsilon_t^{2+2\eta_0}}{2}\mathbb{E}\left(\|\nabla_{\theta_t}g(\theta_t)\|^2\right) + \frac{c}{2}\sum_{t=1}^n \epsilon_t^{1+2\eta_0}\mathbb{E}\left(\|v_t\|^2\right)
$$
$$
+ \sum_{t=1}^n \frac{\epsilon_t^{2+2\eta_0}}{2}\mathbb{E}\left(\|\nabla_{\theta_t}g(\theta_t) - \nabla_{\theta_t}g(\theta_t,\xi_t)\|^2\right).
$$

(33)

It follows from Assumption 3.1 3) and Lemma B.8 that

$$
\epsilon_t^{\eta_0}\,\mathbb{E}\left(\|\nabla_{\theta_t}g(\theta_t) - \nabla_{\theta_t}g(\theta_t,\xi_t)\|^2\right) < \epsilon_t^{\eta_0}\,\mathbb{E}\left(\|\nabla_{\theta_t}g(\theta_t) - \nabla_{\theta_t}g(\theta_t,\xi_t)\|^2\right)MT(\theta_1,v_0) < +\infty.
$$

Because of $\sum_{t=1}^n \epsilon_t^{2+\eta_0} < +\infty$, there is a scalar $\bar{M} > 0$ such that for $\forall n$

$$
\sum_{t=1}^n \frac{\epsilon_t^{2+2\eta_0}}{2}\mathbb{E}\left(\|\nabla_{\theta_t}g(\theta_t) - \nabla_{\theta_t}g(\theta_t,\xi_t)\|^2\right) < MT(\theta_1,v_0)\sum_{t=1}^n \frac{\epsilon_t^{2+\eta_0}}{2} < \bar{M} < +\infty.
$$

Then it follows from (33) that

$$
\frac{1}{2}\sum_{t=1}^n (1 - \alpha^2 - c\epsilon_t)\,\mathbb{E}\left(\epsilon_t^{2\eta_0}\|v_t\|^2\right)
$$
$$
\leq \bar{M} + \epsilon_1^{2\eta_0}g(\theta_1) - \epsilon_{n+1}^{1+2\eta_0}\mathbb{E}\left(g(\theta_{n+1})\right) + \frac{\alpha^2}{2}\mathbb{E}\left(\epsilon_0^{1+2\eta_0}\|v_0\|^2 - \epsilon_n^{1+2\eta_0}\|v_n\|^2\right)
$$
$$
- \sum_{t=1}^n \frac{\epsilon_t^{2+2\eta_0}}{2}\mathbb{E}\left(\|\nabla_{\theta_t}g(\theta_t)\|^2\right) < K,
$$

where $K$ is a positive scalar. Since $\epsilon_n \to 0$ when $n$ is large enough, it holds that $\frac{1}{5}(1 - \alpha^2) < 1 - \alpha^2 - c\epsilon_n$. Without loss of generality, assume $\frac{1}{5}(1 - \alpha^2) < 1 - \alpha^2 - c\epsilon_n^2$ for $n \geq 0$, so $\sum_{t=1}^n \mathbb{E}\left(\epsilon_t^{2\eta_0}\|v_t\|^2\right) < \frac{10K}{1-\alpha^2} < +\infty$. By Lemma B.5, we obtain $\sum_{t=1}^n \epsilon_t^{2\eta_0}\|v_t\|^2 < +\infty$.

## B.6 Proof of Lemma B.10

Through Taylor expansion, we derive that

$$
\epsilon_{t+1}^{2\eta_0}g(\theta_{t+1}) - \epsilon_t^{2\eta_0}g(\theta_t)
$$
$$
= \epsilon_t^{2\eta_0}\nabla_{\theta_{\xi_t}}g(\theta_{\xi_t})^T(\theta_{t+1} - \theta_t) = -\epsilon_t^{2\eta_0}\nabla_{\theta_t}g(\theta_t)^T v_t + \epsilon_t^{2\eta_0}\left(\nabla_{\theta_{\xi_t}}g(\theta_{\xi_t}) - \nabla_{\theta_t}g(\theta_t)\right)^T(\theta_{t+1} - \theta_t).
$$

(34)

We first focus on the term $\epsilon_t^{2\eta_0}\nabla_{\theta_t}g(\theta_t)^T v_t$. It holds that

$$
\epsilon_t^{2\eta_0}\nabla_{\theta_t}g(\theta_t)^T v_t = \epsilon_t^{2\eta_0}(\nabla_{\theta_t}g(\theta_t))^\mathsf{T}(\alpha v_{t-1} + \epsilon_t\nabla_{\theta_n}g(\theta_t,\xi_t))
$$
$$
= \alpha\epsilon_t^{2\eta_0}(\nabla_{\theta_{t-1}}g(\theta_{t-1}) + \nabla_{\theta_t}g(\theta_t) - \nabla_{\theta_{t-1}}g(\theta_{t-1}))^\mathsf{T} v_{t-1} + \epsilon_t^{1+2\eta_0}\nabla_{\theta_t}g(\theta_t)^\mathsf{T}\nabla_{\theta_t}g(\theta_t,\xi_t)
$$
$$
= \alpha\epsilon_{t-1}^{2\eta_0}\nabla_{\theta_{t-1}}g(\theta_{t-1})^\mathsf{T} v_{t-1} + \alpha\epsilon_{t-1}^{2\eta_0}(\nabla_{\theta_t}g(\theta_t) - \nabla_{\theta_{t-1}}g(\theta_{t-1}))^\mathsf{T} v_{t-1} + \epsilon_t^{1+2\eta_0}\nabla_{\theta_t}g(\theta_t)^\mathsf{T}\nabla_{\theta_t}g(\theta_t,\xi_t)
$$
$$
+ (\epsilon_t^{2\eta_0} - \epsilon_{t-1}^{2\eta_0})\nabla_{\theta_{t-1}}g(\theta_{t-1})^\mathsf{T} v_{t-1}.
$$

By substituting the above equation into (11) and noting $-(\nabla_{\theta_i} g(\theta_i) - \nabla_{\theta_{i-1}} g(\theta_{i-1}))^\mathsf{T} v_{i-1} \le \big\|\nabla_{\theta_i} g(\theta_i) - \nabla_{\theta_{i-1}} g(\theta_{i-1})\big\| \|v_{i-1}\| \le c\|v_{t-1}\|^2$, we obtain

$$\epsilon_{t+1}^{2\eta_0} g(\theta_{t+1}) - \epsilon_t^{2\eta_0} g(\theta_t)$$

$$\le -\alpha^{t-1}\epsilon_1^{2\eta_0}\nabla_{\theta_1} g(\theta_1)^\mathsf{T} v_1 - \sum_{i=2}^{t}\alpha^{t-i}\epsilon_i^{1+2\eta_0}\nabla_{\theta_i} g(\theta_i)^\mathsf{T}\nabla_{\theta_i} g(\theta_i,\xi_i) + \frac{c}{2}\epsilon_t^{2\eta_0}\|v_t\|^2$$

$$-\sum_{i=1}^{t-1}\alpha^{t-i}\epsilon_t^{2\eta_0}(\nabla_{\theta_t} g(\theta_t) - \nabla_{\theta_{t-1}} g(\theta_{t-1}))^T v_{t-1}$$

$$+\sum_{i=2}^{t}\alpha^{t-i}(\epsilon_t^{2\eta_0} - \epsilon_{t-1}^{2\eta_0})\nabla_{\theta_{t-1}} g(\theta_{t-1})^\mathsf{T} v_{t-1} \tag{35}$$

$$< -\alpha^{t-1}\epsilon_1^{2\eta_0}\nabla_{\theta_1} g(\theta_1)^\mathsf{T} v_1 - \sum_{i=2}^{t}\alpha^{t-i}\epsilon_i^{1+2\eta_0}\nabla_{\theta_i} g(\theta_i)^\mathsf{T}\nabla_{\theta_i} g(\theta_i,\xi_i) + c'\sum_{i=1}^{t}\alpha^{t-i}\epsilon_i^{2\eta_0}\|v_i\|^2$$

$$+\sum_{i=2}^{t}\alpha^{t-i}(\epsilon_t^{2\eta_0} - \epsilon_{t-1}^{2\eta_0})^2\|\nabla_{\theta_{t-1}} g(\theta_{t-1})\|^2,$$

where $c'$ is a constant which does not affect the result. It follows that

$$\epsilon_{n+1}^{2\eta_0} g(\theta_{n+1})$$

$$= \epsilon_1^{2\eta_0} g(\theta_1) + \sum_{t=1}^{n}\left(\epsilon_{t+1}^{2\eta_0} g(\theta_{t+1}) - \epsilon_t^{2\eta_0} g(\theta_t)\right)$$

$$= \epsilon_1^{2\eta_0} g(\theta_1) - \frac{1-\alpha^n}{1-\alpha}\epsilon_1^{2\eta_0}\nabla_{\theta_1} g(\theta_1)^\mathsf{T} v_1 + \frac{1-\alpha^n}{1-\alpha}\epsilon_1^{1+2\eta_0}\nabla_{\theta_1} g(\theta_1)\nabla_{\theta_1} g(\theta_1,\zeta_1)$$

$$-\sum_{t=1}^{n}\frac{1-\alpha^{n-t+1}}{1-\alpha}\epsilon_t^{1+2\eta_0}\nabla_{\theta_t} g(\theta_t)\nabla_{\theta_t} g(\theta_t,\xi_t) + c''\sum_{i=1}^{t}\epsilon_i^{2\eta_0}\|v_i\|^2 \tag{36}$$

$$+ c''\sum_{i=2}^{t}(\epsilon_t^{2\eta_0} - \epsilon_{t-1}^{2\eta_0})^2\|\nabla_{\theta_{t-1}} g(\theta_{t-1})\|^2,$$

where $c''$ is a constant which does not affect the result. Take the mathematical expectation of (36), and notice Assumption 3.1 1), then we have

$$\epsilon_{n+1}^{2\eta_0}\mathbb{E}\left(g(\theta_{n+1})\right) \le \epsilon_1^{2\eta_0}\mathbb{E}\left(g(\theta_1)\right) + \frac{\alpha\epsilon_1^{2\eta_0}}{1-\alpha}\left|\nabla_{\theta_1} g(\theta_1)^\mathsf{T} v_1\right| + \frac{1}{1-\alpha}\epsilon_1^{1+2\eta_0}\mathbb{E}\left(\|\nabla_{\theta_1} g(\theta_1)\|^2\right)$$

$$- c'''\sum_{t=1}^{n}(\epsilon_t^{1+2\eta_0} - (\epsilon_t^{2\eta_0} - \epsilon_{t-1}^{2\eta_0})^2)\,\mathbb{E}\left(\|\nabla_{\theta_t} g(\theta_t)\|^2\right) + c''\sum_{i=1}^{t}\epsilon_i^{2\eta_0}\mathbb{E}(\|v_i\|^2),$$

where $c'''$ is a constant which does not affect the result. From Lemma B.9, it follows that for some positive constant $Q$,

$$c''\sum_{t=1}^{n}\epsilon_i^{2\eta_0}\mathbb{E}\left(\|v_t\|^2\right) < Q.$$

Since $(\epsilon_t^{1+2\eta_0} - (\epsilon_t^{2\eta_0} - \epsilon_{t-1}^{2\eta_0})^2)$ can be controlled by $k_0\epsilon_t^{1+2\eta_0}$, it holds that

$$\sum_{t=1}^{n}\epsilon_t^{1+2\eta_0}\mathbb{E}\left(\|\nabla_{\theta_t} g(\theta_t)\|^2\right) < \frac{Q'}{k_0} < +\infty, \tag{37}$$

where $Q'$ is a constant. From Lemma B.5, we have $\sum_{t=1}^{n}\epsilon_t^{1+2\eta_0}\|\nabla_{\theta_t} g(\theta_t)\|^2 < +\infty\ a.s..$

## B.7  Proof of Lemma B.11

It follows from (11) and (12) that

$$g(\theta_{t+1}) - g(\theta_t)$$

$$\le \alpha^{t-1}\nabla_{\theta_1} g(\theta_1)^\mathsf{T} v_1 + c\sum_{i=1}^{t}\alpha^{t-i}\|v_i\|^2 - \sum_{i=2}^{t}\alpha^{t-i}\epsilon_i\big\|\nabla_{\theta_i} g(\theta_i)\big\|^2$$

$$+\sum_{i=2}^{t}\alpha^{t-i}\epsilon_i\nabla_{\theta_i} g(\theta_i)^T(\nabla_{\theta_i} g(\theta_i) - \nabla_{\theta_i} g(\theta_i,\xi_i)).$$

Then in light of ([16](#)), we get that

$$\sum_{i=1}^{t}\alpha^{t-i}\|v_i\|^2$$

$$\leq \sum_{i=1}^{t}\alpha^{t-i}\alpha^{(1+\delta)i}\|v_0\|^2 + \sigma_0\sum_{i=1}^{t}\alpha^{(1+\delta)(t-i)}\epsilon_i^2\|\nabla_{\theta_i}g(\theta_i) - \nabla_{\theta_i}g(\theta_i,\xi_i)\|^2$$

$$- \sigma_1\sum_{i=1}^{t}\alpha^{(1+\delta)(t-i)}\big(\epsilon_{k+1}g(\theta_{k+1}) - \epsilon_k g(\theta_k)\big),$$

where $\sigma_0$ and $\sigma_1$ are two constants not affecting the analysis. Then we make some transformations to obtain

$$\sigma_0\sum_{i=1}^{t}\alpha^{(1+\delta)(t-i)}\epsilon_i^2\|\nabla_{\theta_i}g(\theta_i) - \nabla_{\theta_i}g(\theta_i,\xi_i)\|^2$$

$$= \sigma_0\sum_{i=1}^{t}\alpha^{(1+\delta)(t-i)}\epsilon_i^2\,\mathbb{E}\left(\|\nabla_{\theta_i}g(\theta_i) - \nabla_{\theta_i}g(\theta_i,\xi_i)\|^2\Big|\mathcal{F}_n\right) + \sigma_0\sum_{i=1}^{t}\alpha^{(1+\delta)(t-i)}\epsilon_i^2 M_i,$$

where

$$M_i = \mathbb{E}\left(\|\nabla_{\theta_i}g(\theta_i) - \nabla_{\theta_i}g(\theta_i,\xi_i)\|^2\Big|\mathcal{F}_n\right) - \|\nabla_{\theta_i}g(\theta_i) - \nabla_{\theta_i}g(\theta_i,\xi_i)\|^2.$$

From Assumption [3.1](#) 3), it follows that

$$\sigma_0\sum_{i=1}^{t}\alpha^{(1+\delta)(t-i)}\epsilon_i^2\,\mathbb{E}\left(\|\nabla_{\theta_i}g(\theta_i) - \nabla_{\theta_i}g(\theta_i,\xi_i)\|^2\Big|\mathcal{F}_n\right) \leq \sigma_0 M_0\sum_{i=1}^{t}\alpha^{(1+\delta)(t-i)}\epsilon_i^2\|\nabla_{\theta_i}g(\theta_i)\|^2$$

$$+ a'\sum_{i=1}^{t}\alpha^{(1+\delta)(t-i)}\epsilon_i^2.$$

Then we derive that

$$\sum_{i=1}^{t}\alpha^{t-i}\|v_i\|^2$$

$$\leq \sum_{i=1}^{t}\alpha^{t-i}\alpha^{(1+\delta)i}\|v_0\|^2 + \sigma_0 M_0\sum_{i=1}^{t}\alpha^{(1+\delta)(t-i)}\epsilon_i^2\|\nabla_{\theta_i}g(\theta_i)\|^2$$

$$+ a'\sum_{i=1}^{t}\alpha^{(1+\delta)(t-i)}\epsilon_i^2 - \sigma_1\sum_{i=1}^{t}\alpha^{(1+\delta)(t-i)}\big(\epsilon_{k+1}g(\theta_{k+1}) - \epsilon_k g(\theta_k)\big).$$

It holds that

$$g(\theta_{t+1}) - g(\theta_t)$$

$$\leq k_0\alpha^{t-1} + k_1\alpha^t + \sigma_0 M_0\sum_{i=1}^{t}\alpha^{(1+\delta)(t-i)}\epsilon_i^2\|\nabla_{\theta_i}g(\theta_i)\|^2$$

$$+ a'\sum_{i=1}^{t}\alpha^{(1+\delta)(t-i)}\epsilon_i^2 - \sigma_1\sum_{i=1}^{t}\alpha^{(1+\delta)(t-i)}\big(\epsilon_{k+1}g(\theta_{k+1}) - \epsilon_k g(\theta_k)\big) - \sum_{i=2}^{t}\alpha^{t-i}\epsilon_i\big\|\nabla_{\theta_i}g(\theta_i)\big\|^2$$

$$+ \sum_{i=2}^{t}\alpha^{t-i}\epsilon_i\nabla_{\theta_i}g(\theta_i)^T(\nabla_{\theta_i}g(\theta_i) - \nabla_{\theta_i}g(\theta_i,\xi_i)).$$

It follows from some transformations that

$$g(\theta_{t+1}) - g(\theta_t) \leq \frac{a'}{1-\alpha^{1+\delta}}\epsilon_t^{1+2\eta_0} + Q_t, \tag{38}$$

where

$$Q_t = a'\left(\sum_{i=1}^{t}\alpha^{(1+\delta)(t-i)}\epsilon_i^2 - \frac{1}{1-\alpha^{1+\delta}}\epsilon_t^2\right) + k_0\alpha^{t-1} + k_1\alpha^t$$

$$+ \sigma_0 M_0\sum_{i=1}^{t}\alpha^{(1+\delta)(t-i)}\epsilon_i\|\nabla_{\theta_i}g(\theta_i)\|^2 - \sigma_1\sum_{i=1}^{t}\alpha^{(1+\delta)(t-i)}\big(\epsilon_{k+1}g(\theta_{k+1}) - \epsilon_k g(\theta_k)\big)$$

$$+ \sum_{i=2}^{t}\alpha^{t-i}\epsilon_i\nabla_{\theta_i}g(\theta_i)^T(\nabla_{\theta_i}g(\theta_i) - \nabla_{\theta_i}g(\theta_i,\xi_i)).$$

Next we will prove that $\sum_{t=1}^{+\infty} Q_t$ exists and is finite $a.s.$. First through Lemma B.7, we get

$$\sum_{t=1}^{+\infty} a' \left( \sum_{i=1}^{t} \alpha^{(1+\delta)(t-i)} \epsilon_i^2 - \frac{1}{1-\alpha^{1+\delta}} \epsilon_t^2 \right) = 0.$$

Obviously

$$\sum_{t=1}^{+\infty} \left( k_0 \alpha^{t-1} + k_1 \alpha^t \right) < +\infty.$$

For the term $\sigma_0 M_0 \sum_{i=1}^{t} \alpha^{(1+\delta)(t-i)} \epsilon_i^2 \|\nabla_{\theta_i} g(\theta_i)\|^2$, from (B.10) and $0 < \eta_0 < 1/2$, we derive

$$\sum_{t=1}^{+\infty} \sigma_0 M_0 \sum_{i=1}^{t} \alpha^{(1+\delta)(t-i)} \epsilon_i^2 \|\nabla_{\theta_i} g(\theta_i)\|^2 = \frac{\sigma_0 M_0}{1-\alpha^{1+\delta}} \sum_{t=1}^{+\infty} \epsilon_t^2 \|\nabla_{\theta_t} g(\theta_t)\|^2$$

$$< \frac{\sigma_0 M_0}{1-\alpha^{1+\delta}} \sum_{t=1}^{+\infty} \epsilon_t^{1+2\eta_0} \|\nabla_{\theta_t} g(\theta_t)\|^2 < +\infty \ \ a.s..$$

For the term $-\sigma_1 \sum_{i=1}^{t} \alpha^{(1+\delta)(t-i)} \left( \epsilon_{k+1} g(\theta_{k+1}) - \epsilon_k g(\theta_k) \right)$, from (36), it follows that $\epsilon_n^{2\eta_0} g(\theta_n) < \zeta' < +\infty$ ($\forall n > 0$). Furthermore, it holds that $\epsilon_n g(\theta_n) = \epsilon_n^{1-2\eta_0} \epsilon_n^{2\eta_0} g(\theta_n) \to 0 \ \ a.s.$ and

$$\sum_{t=1}^{+\infty} -\sigma_1 \sum_{i=1}^{t} \alpha^{(1+\delta)(t-i)} \left( \epsilon_{k+1} g(\theta_{k+1}) - \epsilon_k g(\theta_k) \right) = \frac{\sigma_0 \epsilon_1 g(\theta_1)}{1-\alpha^{1+\delta}}.$$

For the term $\sum_{i=2}^{t} \alpha^{t-i} \epsilon_i \nabla_{\theta_i} g(\theta_i)^T (\nabla_{\theta_i} g(\theta_i) - \nabla_{\theta_i} g(\theta_i, \xi_i))$, we first consider $\sum_{t=1}^{+\infty} \epsilon_t \nabla_{\theta_t} g(\theta_t)^T (\nabla_{\theta_t} g(\theta_t) - \nabla_{\theta_t} g(\theta_t, \xi_t))$. From Lemma B.10, it follows that

$$\sum_{t=1}^{+\infty} \mathbb{E} \left( \epsilon_t^2 \left( \nabla_{\theta_t} g(\theta_t)^T \left( \nabla_{\theta_t} g(\theta_t) - \nabla_{\theta_t} g(\theta_t, \xi_t) \right) \right)^2 \big| \mathcal{F}_n \right)$$

$$\leq a \sum_{t=1}^{+\infty} \mathbb{E} \left( \epsilon_t^2 \|\nabla_{\theta_t} g(\theta_t)\|^2 \big| \mathcal{F}_n \right) + \hat{M} \sum_{t=1}^{+\infty} \mathbb{E} \left( \epsilon_t^2 \|\nabla_{\theta_t} g(\theta_t)\|^4 \big| \mathcal{F}_n \right) < +\infty \ \ a.s.$$

This means $\sum_{t=1}^{+\infty} \epsilon_t \nabla_{\theta_t} g(\theta_t)^T (\nabla_{\theta_t} g(\theta_t) - \nabla_{\theta_t} g(\theta_t, \xi_t))$ exists and is finite a.s.. From Lemma B.7, it holds that $\sum_{i=2}^{t} \alpha^{t-i} \epsilon_i \nabla_{\theta_i} g(\theta_i)^T (\nabla_{\theta_i} g(\theta_i) - \nabla_{\theta_i} g(\theta_i, \xi_i))$ exists and is finite a.s.. Thus, $\sum_{t=1}^{+\infty} Q_t$ exists and is finite a.s..

# C  Proofs of Theorems 3.1–3.4 and Proposition 3.1

## C.1  Proof of Theorem 3.1

We prove this theorem under two situations, namely, $\sum_{n=1}^{+\infty} \epsilon_n^{1+2\eta_0} < +\infty$ and $\sum_{n=1}^{+\infty} \epsilon_n^{1+2\eta_0} = +\infty$. If $\sum_{n=1}^{+\infty} \epsilon_n^{1+2\eta_0} < +\infty$, we can conclude that $\sum_{n=1}^{+\infty} \epsilon_n^2 < +\infty$. Then our condition degenerates to the classical condition $\sum_{i=1}^{+\infty} \epsilon_n = +\infty$, $\sum_{i=1}^{+\infty} \epsilon_n^2 < +\infty$. The results then follow from Theorem 1 in [7]. Thus, the following proof is all based on the condition that $\sum_{n=1}^{+\infty} \epsilon_n^{1+2\eta_0} = +\infty$. First we will prove that there exists a subsequence $\{\theta_{k_n}\}$, such that $\|\nabla_{\theta_{k_n}} g(\theta_{k_n})\| \to 0 \ \ a.s.$. We prove it by contradiction. We assume $\exists N(\zeta) > 0$, when $\theta_n > N(\zeta)$, it holds that $\|\nabla_{\theta_t} g(\theta_t)\|^2 > \delta_0^2 > 0$. Then we have

$$\sum_{t=N(\zeta)}^{+\infty} \epsilon_t^{1+2\eta_0} \|\nabla_{\theta_t} g(\theta_t)\|^2 > \delta_0^2 \sum_{t=N(\zeta)}^{+\infty} \epsilon_t^{1+2\eta_0} = +\infty, \tag{39}$$

which contradicts with (37). Thus, there exists a subsequence $\{\theta_{k_n}\}$ satisfying $\|\nabla_{\theta_{k_n}} g(\theta_{k_n})\| \to 0 \ \ a.s.$. It is easy to find that $J_i$ is a bounded closed set. So $\forall \epsilon > 0$, we can construct an open cover $H_\epsilon^{(i)} = \{U(\theta, \epsilon)\}$ ($\theta \in J_i$) of $J_i$. Through the $Heine\breve{~}Borel\ theorem$, we can get a finite open subcover $\{U(\theta_k, \epsilon)$ ($k = 0, 1, ..., n$) from $H_\epsilon^{(i)}$. Let $U_\epsilon^{(i)} = \bigcup_{k=0}^{n} U(\theta_k, \epsilon)$, then $U_\epsilon^{(i)}$ is a open set. Under Assumption 3.1, $J = \{\theta | \nabla_\theta g(\theta)\}$ has only finite connected components $J_1, J_2, ..., J_m$. So $\inf_{i \neq j} d(J_i, J_j) = \min_{i \neq j} d(J_i, J_j)$. Let $\delta_0 = \min_{i \neq j} d(J_i, J_j)$. It follows from Lemma B.3 that $\exists \epsilon_0 > 0$, such that when $d(\theta, J_i) < \epsilon_0$, it holds that

$$\|\nabla_\theta g(\theta)\|^2 \leq 2c|g(\theta) - g_i|, \tag{40}$$

where $g_i$ denotes $g(\theta)$ ($\theta \in J_i$). Let $c = min\{\epsilon_0, \delta_0/4\}$ and construct $U_c^{(1)}, U_c^{(2)}, ..., U_c^{(m)}$. It is obvious that $\forall U_c^{(i)}, U_c^{(j)}$ ($i \neq j$), $d(U_c^{(i)}, U^{(j)}) > \delta_0/2$, and $\|\nabla_\theta g(\theta)\|^2 \leq 2c|g(\theta) - g_i|$ ($\theta \in U_c^{(i)}$). Since $J$ is a bounded

set, $\exists N > 0$, such that $J \subset K$ ($K$ is the closure of $U(0, N)$). Then we construct a set $M = K / \bigcup_{i=1}^{m} U_c^{(i)}$. Since $U_c^{(i)}$ is an open set and $K$ is a closed set, we conclude $M$ is a closed set. Since $\|\nabla_\theta g(\theta)\|$ is a continuous function, $\exists \theta_0 \in M$, $\|\nabla_{\theta_0} g(\theta_0)\| = \min_{\theta \in M} \|\nabla_\theta g(\theta)\|$. Let $r = \|\nabla_{\theta_0} g(\theta_0)\| > 0$.

Then we prove that $\forall u > 0, \theta \in K, \exists \delta > 0$, if $\|\nabla_\theta g(\theta)\| < \delta$, then $d(\theta, J) < u$ holds. We prove it by contradiction. Assume $\exists u_0 > 0, \forall \delta_1 > 0, \exists \theta_{\delta_1}$, such that $\|\nabla_{\theta_{\delta_1}} g(\theta_{\delta_1})\| < \delta_1$ and $d(\theta, J) \geq u_0$. We choose $\delta_1 = 1, 1/2, 1/3...$, and construct a sequence $\{\theta_{1/n}\}$. It is obvious that $\|\nabla_{\theta_{1/n}} g(\theta_{1/n}) \to 0\|$. Since $\{\theta_{1/n}\}$ is bounded, through the *Accumulation point theorem*, there exists a convergent subsequence $\{\theta_{1/k_n}\} \subset \{\theta_{1/n}\}$. Let $\theta^{(0)} = \lim_{n \to +\infty} \theta_{1/k_n}$. From the continuity of $d(\theta, J)$ and $\|\nabla_\theta g(\theta)\|$, we get $d(\theta^{(0)}) \geq u_0$ and $\|\nabla_{\theta^{(0)}} g(\theta^{(0)})\| = 0$. It contradicts with the definition of $J$. So $\forall u > 0, \theta \in K, \exists \delta > 0$, $\|\nabla_\theta g(\theta)\| < \delta$, such that $d(\theta, J) < u$. Furthermore, due to the continuity of $g(\theta)$, we can get $\forall \epsilon_1 > 0$, $\exists \delta' > 0$, if $d(\theta, J_i) < \delta'$, it holds that $|g(\theta) - g_i| < \epsilon_1$. Combine these two consequences, then we can prove $\forall \epsilon_1 > 0, \exists b > 0$, if $\theta \in U_c^{(i)}$ and $\|\nabla_\theta g(\theta)\| < b$, it holds that $|g(\theta) - g_i| < \epsilon_1$. Through (39)

$$\lim_{n \to +\infty} \|\nabla_{\theta_{k_n}} g(\theta_{k_n})\|^2 = 0 \quad a.s.. \tag{41}$$

Next we aim to prove $\lim_{n \to +\infty} \|\nabla_{\theta_n} g(\theta_n)\|^2 = 0$. It is equivalent to prove that $\{\|\nabla_n g(\theta_n)\|^2\}$ has no positive accumulation points, that is to say, $\forall e_0 > 0$, there are only finite values of $\|\nabla_{\theta_n} g(\theta)\|$ larger than $e_0$. Obviously, we just need to prove $\forall 0 < e_0 < r$, there are only finite values of $\|\nabla_{\theta_n} g(\theta)\|$ larger than $e$. We prove this by contradiction. We suppose $\exists 0 < e < a$, such that the set $S = \{\|\nabla_{\theta_n} g(\theta_n)\| > e\}$ is an infinite set. Then we let $\epsilon_1 = e/8c$ and $o = min\{b, e/4\}$. Due to (41), there exists a subsequence $\{\theta_{p_n}\}$ of $\{\theta_n\}$ which satisfies $\|\nabla_{\theta_{p_n}} g(\theta_{p_n})\| < o$. We rank $S$ as a subsequence $\{\|\nabla_{m_n} g(\theta_{m_n})\|^2\}$ of $\{\|\nabla_n g(\theta_n)\|^2\}$. Then there is an infinite subsequence $\{\|\nabla_{m_{i_n}} g(\theta_{m_{i_n}})\|^2\}$ of $\{\|\nabla_{m_n} g(\theta_{m_n})\|^2\}$ such that $\forall n \in \mathbb{N}_+, \exists l, n_{p_n} \in (m_{i_l}, m_{i_{l+1}})$. For convenience, we abbreviate $\{m_{i_n}\}$ as $\{i_n\}$. We construct another infinite sequence $\{q_n\}$ as follows

$$q_1 = \max \left\{ n : p_1 < n < \min\{m_{i_l : m_{i_l} > p_1}\}, \|\nabla_{\theta_n} g(\theta_n)\| \leq o \right\},$$

$$q_2 = \min \left\{ n : n > q_1, \|\nabla_{\theta_n} g(\theta_n)\| > e \right\},$$

$$q_{2n-1} = \max \left\{ n : \min\{m_{i_l} : m_{i_l} > q_{2n-3}\} < n < \min\{m_l : m_l > \min\{m_{i_l} : m_{i_l} > q_{2n-3}\}, \right.$$
$$\left. \|\nabla_{\theta_n} g(\theta_n)\| \leq o \right\},$$

$$q_{2n} = \min \left\{ n : n > q_{2n-1}, \|\nabla_{\theta_n} g(\theta_n)\| > e \right\}.$$

Now we prove that $\exists N_0$, when $q_{2n} > N_0$, it holds that $e < \|\nabla_{\theta_{q_{2n}}} g(\theta_{q_{2n}})\| < r$. The left-hand side is obvious (the definition of $q_{2n}$). For the right-hand side, it holds that $\|\nabla_{\theta_{q_{2n}-1}} g(\theta_{q_{2n}-1})\| \leq e$. From (36) and Lemma B.7, it follows that

$$\|\theta_{n+1} - \theta_n\|^2 = \|\alpha v_{n-1} + \epsilon_n \nabla_{\theta_n} g(\theta_n, \xi_n)\|^2 = \left\| \sum_{i=1}^{n} \alpha^{n-i} \epsilon_i \nabla_{\theta_i} g(\theta_i, \xi_i) \right\|^2$$

$$\leq \frac{1}{1-\alpha} \sum_{i=1}^{n} \alpha^{n-i} \epsilon_i^2 \|\nabla_{\theta_i} g(\theta_i, \xi_i)\|^2 \leq \frac{M'}{1-\alpha} \sum_{i=1}^{n} \alpha^{n-i} \epsilon_i^2 \|\nabla_{\theta_i} g(\theta_i)\|^2 + \frac{a}{1-\alpha} \sum_{i=1}^{n} \alpha^{n-i} \epsilon_i^2$$

$$+ \frac{1}{1-\alpha} \sum_{i=1}^{n} \alpha^{n-i} \epsilon_i^2 \left( \|\nabla_{\theta_i} g(\theta_i, \xi_i)\|^2 - \mathbb{E}\left( \|\nabla_{\theta_i} g(\theta_i, \xi_i)\|^2 \big| \mathcal{F}_i \right) \right) \to 0 \quad a.s..$$

From Assumption 3.1 2), it holds that $\left| \|\nabla_{\theta_{n+1}} g(\theta_{n+1})\|^2 - \|\nabla_{\theta_n} g(\theta_n)\|^2 \right| \leq \left| \|\nabla_{\theta_{n+1}} g(\theta_{n+1})\| - \|\nabla_{\theta_n} g(\theta_n)\| \right|^2 \leq \|\nabla_{\theta_{n+1}} g(\theta_{n+1}) - \nabla_{\theta_n} g(\theta_n)\|^2 \leq c\|\theta_{n+1} - \theta_n\| \to 0 \quad a.s.$, So $\exists N_0$, when $n > N_0$, it holds that $\left| \|\nabla_{\theta_{n+1}} g(\theta_{n+1})\|^2 - \|\nabla_{\theta_n} g(\theta_n)\| \right| < r - e$. Then we can get that when $q_{2n} > N_0 + 1$, it holds that $\|\nabla_{\theta_{q_{2n}}} g(\theta_{q_n})\| \leq \|\nabla_{\theta_{q_{2n}-1}} g(\theta_{q_{2n}-1})\| + \left| \|\nabla_{\theta_{q_{2n}}} g(\theta_{q_{2n}})\| - \|\nabla_{\theta_{q_{2n}-1}} g(\theta_{q_{2n}-1})\| \right| \leq e + r - e = r$. This means that $\theta_{q_{2n}} \in \bigcup_{i=1}^{m} U_\tau^{(i)}$. Then $\exists i_0$, such that $\theta_{q_{2n}} \in U_\tau^{(i_0)}$. Due to $\|\nabla_{\theta_n} g(\theta_n)\| \leq e < r$ ($n \in [q_{2n-1}, q_{2n})$), it holds that $\forall k \in [q_{2n-1}, q_{2n}), \exists i_k$, such that $\theta_n \in U_\tau^{(i_k)}$. Since $\|\theta_{n+1} - \theta_n\| \to 0 a.s.$, it holds that $i_0 = i_k$ ($\forall j \in [q_{2n-1}, q_{2n})$). For convenience, we let $i_0 = i_{q_{2n-1}} = ... = i_{q_{2n}-1} = i_{q_{2n}}$. Then we conclude that

$$\|\nabla_\theta g(\theta_n)\|^2 \leq 2c|g(\theta_n) - g_{i_{q_{2n}}}| \quad (n \in [q_{2n-1}, q_{2n}]).$$

Due to the locally sign-preserving property, it follows that

$$\|\nabla_\theta g(\theta_n)\|^2 \leq 2c(g(\theta_n) - g_{i_{q_{2n}}}) \quad (g(\theta_n) \geq g_{i_{q_{2n}}}) \quad or$$

$$\|\nabla_\theta g(\theta_n)\|^2 \leq -2c(g(\theta_n) - g_{i_{q_{2n}}}) \quad (g(\theta_n) \leq g_{i_{q_{2n}}}) \quad (n \in [q_{2n-1}, q_{2n}]).$$

Since it is the same to study the two cases, we just show how to prove the first case. We derive that

$$e - o < \|\nabla_{\theta_{q_{2n}}} g(\theta_{q_{2n}})\|^2 - \|\nabla_{\theta_{q_{2n-1}}} g(\theta_{q_{2n-1}})\|^2 < 2c(g(\theta_{q_{2n}}) - g_{i_{q_{2n}}}) - \|\nabla_{\theta_{q_{2n-1}}} g(\theta_{q_{2n-1}})\|^2$$

$$= \left( 2c \sum_{i=0}^{q_{2n} - q_{2n-1} - 1} g(\theta_{q_{2n-1}+i+1}) - g(\theta_{q_{2n-1}+i}) \right) + 2c\left( g(\theta_{q_{2n-1}}) - g_{i_{q_{2n}}} \right) - \|\nabla_{\theta_{q_{2n-1}}} g(\theta_{q_{2n-1}})\|^2.$$

From (38), we obtain

$$g(\theta_{q_{2n-1}+i+1}) - g(\theta_{q_{2n-1}+i}) \leq \frac{a'}{1-\alpha^{1+\delta}}\epsilon^2_{q_{2n-1}+i} + Q_{q_{2n-1}+i}$$

$$< \frac{a'}{1-\alpha^{1+\delta}}\epsilon^{1+2\eta_0}_{q_{2n-1}+i} + Q_{q_{2n-1}+i}.$$

So it holds that

$$e - o < \sum_{i=0}^{q_{2n}-q_{2n-1}-1} L\epsilon^{1+2\eta_0}_{q_{2n-1}+i} + \sum_{i=0}^{q_{2n}-q_{2n-1}-1} Q_{q_{2n-1}+i} \tag{42}$$
$$+ 2c\Big(g(\theta_{q_{2n-1}}) - g_{i_{2n-1}}\Big) - \big\|\nabla_{\theta_{q_{2n-1}}}g(\theta_{q_{2n-1}})\big\|^2,$$

where

$$L = \frac{a'}{1-\alpha^{1+\delta}}.$$

Due to $\big\|\nabla_{\theta_{q_{2n-1}}}g(\theta_{q_{2n-1}})\big\|^2 < o < b$, it follows that $g(\theta_{q_{2n-1}}) - g_{i_{2n-1}} < e/8c$. Substitute it into (43), then we get

$$\sum_{i=0}^{q_{2n}-q_{2n-1}-1} \epsilon^{1+2\eta_0}_{q_{2n-1}+i} > \frac{e}{2L} - \sum_{i=0}^{q_{2n}-q_{2n-1}-1} Q_{q_{2n-1}+i}. \tag{43}$$

From Lemma B.11, it follows that $\sum_{n=1}^{+\infty} Q_n$ is finite almost surely. Thus, we attain

$$\sum_{i=0}^{q_{2n}-q_{2n-1}-1} Q_{q_{2n-1}+i} \to 0 \ a.s.$$

by the $Cauchy's\ test\ for\ convergence$. From $\epsilon^{1+2\eta_0}_{q_{2n-1}+i} \to 0\ a.s.$, it follows that

$$\sum_{i=1}^{q_{2n}-q_{2n-1}-1} \epsilon^{1+2\eta_0}_{q_{2n-1}+i} > \frac{e}{2L} - \epsilon^{1+2\eta_0}_{q_{2n-1}+i} - \sum_{i=0}^{q_{2n}-q_{2n-1}-1} Q_{q_{2n-1}+i} \to \frac{e}{2L}\ a.s.. \tag{44}$$

Thus, it holds that

$$\sum_{n=1}^{+\infty}\left(\sum_{i=1}^{q_{2n}-q_{2n-1}-1} \epsilon^{1+2\eta_0}_{q_{2n-1}+i}\right) = +\infty\ a.s.. \tag{45}$$

On the other hand, it holds that $\big\|\nabla_{\theta_{q_{2n-1}+i}}g(\theta_{q_{2n-1}+i})\big\| > o\ (i > 0)$. Together with (37), we get

$$\sum_{n=1}^{+\infty}\left(\sum_{i=1}^{q_{2n}-q_{2n-1}-1} \epsilon^{1+2\eta_0}_{q_{2n-1}+i}\right) < \frac{1}{o}\sum_{n=1}^{+\infty}\left(\sum_{i=1}^{q_{2n}-q_{2n-1}-1} \big\|\nabla_{\theta_{q_{2n-1}+i}}g(\theta_{q_{2n-1}+i})\big\|^2 \epsilon^{1+2\eta_0}_{q_{2n-1}+i}\right)$$
$$< \frac{1}{o}\sum_{n=3}^{n} \big\|\nabla_{\theta_n}g(\theta_n)\big\|^2 \epsilon^{1+2\eta_0}_{q_{2n-1}+i} < +\infty\ a.s.. \tag{46}$$

It contradicts with (45), so we get that $\big\|\nabla_{\theta_n}g(\theta_n)\big\| \to 0\ a.s..$ Under Assumption 3.1 1), it is safe to conclude that there exists a connected component $J^*$ of $J$ such that $\lim_{n\to\infty} d(\theta_n, J^*) = 0$.

## C.2 Proof of Theorem 3.2

First we construct an event

$$\hat{B}_n = \{there\ exists\ at\ least\ one\ point\ n_0 \in [0,n],\ satisfing\ \|\nabla_{\theta_{n_0}}g(\theta_{n_0})\|^2 \leq k_0\epsilon^2_{n_0}\},$$

where $k_0$ is an undetermined coefficient Then we define another events

$$B_i = \{\|\nabla_{\theta_i}g(\theta_i)\|^2 < k_0\epsilon^2_i\},$$

$$B_{i,j} = \{\|\nabla_{\theta_i}g(\theta_i)\|^2 \leq k_0\epsilon^2_i\ and\ for\ all\ n_0 \in (i,j]\ \|\nabla_{\theta_n}g(\theta_n)\|^2 > k_0\epsilon^2_n\}.$$

Specially, we denote $B_{n,n} = B_n$ Absolutely, there is

$$\hat{B}_n = \bigcup_{i=1}^{n} B_{i,n}.$$

Let the characteristic function of $B_n$ be $\hat{I}_n$, $B_{i,j}$ be $\hat{I}_{i,j}$. Through Theorem 3.1, we know $\theta_n$ must converge to some stationary point, so $g(\theta_n) \to \sum_{k=1}^m I_k g_k$ $a.s.$ and $g_k$ $(k = 1, 2, ..., m)$ must be a local minimum. We denote $\hat{g} = g - \sum_{k=1}^m I_k g_k$. Then we get $\hat{g} \to 0$ $a.s.$ and $\liminf_{n \to +\infty} \text{sgn}(\hat{g}(\theta_n)) > 0$ $a.s.$. There is

$$\hat{I}_{i,n+1}\hat{g}(\theta_{n+1}) - \hat{I}_{i,n}\hat{g}(\theta_n) \le \mu\alpha^{n-i}\hat{I}_{i,i+1}\|\nabla_{\theta_{i+1}}g(\theta_{i+1})\|^2 - \sum_{j=i+1}^n \alpha^{n-j}\epsilon_j\hat{I}_{i,j}\nabla_{\theta_j}g(\theta_j)^T\nabla_{\theta_j}g(\theta_j, \xi_j)$$

$$+ c'' \sum_{j=i+1}^n \alpha^{n-j}\epsilon_j^2\hat{I}_{i,j}\|\nabla_{\theta_j}g(\theta_j) - \nabla_{\theta_j}g(\theta_j, \xi_j)\|^2.$$

where $\mu > 0$ is a constant. We define $\hat{S} = \{g(\theta) < \hat{T}\}$, $\hat{I}_{i,j}^{(a)} = \{some\ t \in (i, j)\ satisfies\ \|\nabla_{\theta_t}g(\theta_t)\|^2 > q_t\epsilon_t^2\ and\ \|\nabla_{\theta_j}g(\theta_j)\|^2 > a\}$. We denote $k_0 = 2ac''$. Then exists $a > 0$, after adjusting $\{q_t\}$, making $\mathbb{E}(\hat{I}_{i,j}\|\nabla_{\theta_j}g(\theta_j)\|^2) \ge \mathbb{E}(\hat{I}_{i,j}^{(a)}\|\nabla_{\theta_j}g(\theta_j)\|^2)$ and $1/2\,\mathbb{E}(\hat{I}_{i,j}) \le \mathbb{E}(\hat{I}_{i,j}^{(a)})$. Then through Assumption 3.1 (3), i.e., $E_{\xi_n}\left(\|\nabla_\theta g(\theta, \xi_n)\|^2\right) \le M'\|\nabla_\theta g(\theta)\|^2 + a'$. We take the mathematical expectation on above equation to obtain

$$\mathbb{E}\left(\hat{I}_{i,n+1}\hat{g}(\theta_{n+1})\right) - \mathbb{E}\left(\hat{I}_{i,n}\hat{g}(\theta_n)\right) \le \mu\alpha^{n-i}\mathbb{E}\left(\hat{I}_{i,i+1}\|\nabla_{\theta_{i+1}}g(\theta_{i+1})\|^2\right)$$

$$- \sum_{j=i+1}^n \alpha^{n-j}\epsilon_j\,\mathbb{E}\left(\hat{I}_{i,j}\|\nabla_{\theta_j}g(\theta_j)\|^2\right)$$

$$+ c'' \sum_{j=i+1}^n \alpha^{n-j}\epsilon_j^2\left(M' + \frac{a'}{2k_0 a}\right)\mathbb{E}\left(\hat{I}_{i,j}\|\nabla_{\theta_j}g(\theta_j)\|^2\right)$$

$$= \mu\alpha^{n-i}\mathbb{E}\left(\hat{I}_{i,i+1}\|\nabla_{\theta_{i+1}}g(\theta_{i+1})\|^2\right)$$

$$- \sum_{j=i+1}^n \alpha^{n-j}\left(\epsilon_j - c''\epsilon_j^2\left(M' + \frac{a'}{2k_0 a}\right)\right)\mathbb{E}\left(\hat{I}_{i,j}\|\nabla_{\theta_j}g(\theta_j)\|^2\right).$$

We construct a variable $F_{i,n+1}$ as follows

$$F_{i,n+1} = \sum_{t=i+1}^{n+1}\left(\frac{1}{2-\alpha}\right)^{n+1-t}\mathbb{E}\left(\hat{I}_{i,t}\hat{g}(\theta_t)\right).$$

So we can get

$F_{i,n+1} - F_{i,n}$

$$\le \sigma_0\left(\frac{1}{2-\alpha}\right)^{n-i}\mathbb{E}\left(\hat{I}_{i,i+1}\|\nabla_{\theta_{i+1}}g(\theta_{i+1})\|^2\right) - \frac{2}{(1-\alpha)^2}\sum_{j=i+1}^n\left(1 - \frac{a'c''}{2k_0 a}\right)\left(\frac{1}{2-\alpha}\right)^{n-j}\epsilon_j\,\mathbb{E}\left(\hat{I}_{i,j}\|g(\theta_j)\|^2\right).$$

Through Assumption 3.3, we know when $\theta \in S = \{\|\nabla_\theta g(\theta)\|^2/(g(\theta) - g^*) < \delta_0\}$, there is $g(\theta) < \hat{T}$. We denote $k_0 = a'c''/a$. Then when $\theta_j \in \hat{S}$, there is $\mathbb{E}_{\theta_j \in \hat{S}}\left(\hat{I}_{i,j}\|\nabla_{\theta_{i,j}}g(\theta_{i,j})\|^2\right) = \mathbb{E}_{\theta_j \in \hat{S}}\left(\hat{I}_{i,j}^{(a)}\|\nabla_{\theta_{i,j}}g(\theta_{i,j})\|^2\right) > (a/2\hat{T})\mathbb{E}_{\theta_j \in \hat{S}}\left(\hat{I}_{i,j}g(\theta_j)\right)$, and when $\theta_j \in \mathbb{R}^N/S$, there is $\mathbb{E}_{\theta_j \in \mathbb{R}^N/S}\left(\hat{I}_{i,j}\|\nabla_{\theta_j}g(\theta_j)\|^2\right) > 2\delta_0\,\mathbb{E}_{\theta_j \in \mathbb{R}^N/S}\left(\hat{I}_{i,j}g(\theta_j)\right)$. We assign $\hat{c} = \min\{2\delta_0, a/2\hat{T}\}$. Then we can get that

$$F_{i,n} \le K_0 F_{i,i+1} e^{-\frac{2\hat{c}}{(1-\alpha)^2}\sum_{k=i+1}^n \epsilon_k},$$

where $K_0$ is a constant. Regarding the value of $F_{i,i+1}$, it holds that

$$F_{i,i+1} \le \mathbb{E}\left(\hat{I}_{i,i+1}|\hat{g}(\theta_{i+1})|\right) \le \mathbb{E}(\hat{I}_i|\hat{g}(\theta_{i+1})|),$$

$$\hat{I}_i|\hat{g}(\theta_{i+1}) - \hat{g}(\theta_i)| = \hat{I}_i|\nabla_i g(\theta_i)^T v_i| = \hat{I}_i|\nabla_i g(\theta_i)^T(\alpha v_{i-1} + \epsilon_i\nabla_i g(\theta_i, \xi_i))| \le O(\epsilon_i^2).$$

As a result,

$$F_{i,n} = O\left(\epsilon_i^2 e^{-\frac{2\hat{c}}{(1-\alpha)^2}\sum_{k=i+1}^n \epsilon_k}\right).$$

Then we get

$$\sum_{i=1}^n \mathbb{E}(\hat{I}_{i,n}\|\nabla_{\theta_n}g(\theta_n)\|^2) = O\left(\sum_{i=1}^n \epsilon_i^2 e^{-\frac{2\hat{c}}{(1-\alpha)^2}\sum_{k=i+1}^n \epsilon_k}\right).$$

For the adverse events of $\hat{B}_n$,

$$\Omega/\hat{B}_n = \{there\ is\ no\ point\ n_0 \in [0, n],\ satisfying\ \|\nabla_{\theta_{n_0}}g(\theta_{n_0})\|^2 \le k_0\epsilon_{n_0}^2\},$$

we use the same function, getting

$$\mathbb{E}\left(I_{\Omega/\hat{B}_n}\|\nabla_{\theta_n}g(\theta_n)\|^2\right) = O\left(e^{-\frac{2\hat{c}}{(1-\alpha)^2}\sum_{k=1}^n \epsilon_k}\right).$$

Finally, we get that

$$\mathbb{E}\left(\|\nabla_{\theta_n} g(\theta_n)\|^2\right) = O\left(\sum_{i=1}^{n} \epsilon_i^2 e^{-\frac{2\hat{c}}{(1-\alpha)^2}\sum_{k=i}^{n}\epsilon_k}\right) = O\left(e^{-\frac{2\hat{c}}{(1-\alpha)^2}\sum_{k=1}^{n}\epsilon_k}\sum_{i=1}^{n}\epsilon_i^2 e^{\frac{2\hat{c}}{(1-\alpha)^2}\sum_{k=1}^{i}\epsilon_k}\right).$$

## C.3 Proof of Theorem 3.3

*Proof.* First we have

$$\begin{aligned} v_n &= \alpha v_{n-1} + \epsilon_n \nabla_{\theta_n} g(\theta_n, \xi_n) \\ \theta_{n+1} &= \theta_n - v_n. \end{aligned} \tag{47}$$

Then we calculate that

$$g(\theta_{n+1}) \leq g(\theta_n) - \nabla_{\theta_n} g(\theta_n)^T v_n + \frac{c}{2}\|v_n\|^2.$$

First, we focus on the term $\nabla_{\theta_n} g(\theta_n)^T v_n$, which satisfies

$$\begin{aligned} -\nabla_{\theta_n} g(\theta_n)^T v_n &= -\nabla_{\theta_n} g(\theta_n)^T (\alpha v_{n-1} + \epsilon_n \nabla_{\theta_n} g(\theta_n, \xi_n)) \\ &\leq -\alpha \nabla_{\theta_{n-1}} g(\theta_{n-1})^T v_{n-1} + c\alpha\|v_{n-1}\|^2 - \epsilon_n \nabla_{\theta_n} g(\theta_n)^T \nabla_{\theta_n} g(\theta_n, \xi_n)). \end{aligned} \tag{48}$$

We multiple $I_n^{(\hat{a})}$ on both sides of (48), leading to

$$\begin{aligned} &-I_n^{(\hat{a})} \nabla_{\theta_n} g(\theta_n)^T v_n \\ &\leq -\alpha I_{n-1}^{(\hat{a})} \nabla_{\theta_{n-1}} g(\theta_{n-1})^T v_{n-1} + \alpha\big(I_{n-1}^{(\hat{a})} - I_n^{(\hat{a})}\big) \nabla_{\theta_{n-1}} g(\theta_{n-1})^T v_{n-1} + c\alpha I_{n-1}^{(\hat{a})}\|v_{n-1}\|^2 \\ &\quad - \epsilon_n I_n^{(\hat{a})} \nabla_{\theta_n} g(\theta_n)^T \nabla_{\theta_n} g(\theta_n, \xi_n)). \end{aligned} \tag{49}$$

For convenience, we denote $\hat{u}_{n-1} = \big(I_{n-1}^{(\hat{a})} - I_n^{(\hat{a})}\big)\nabla_{\theta_{n-1}} g(\theta_{n-1})^T v_{n-1}$, and then we iterate (49), getting

$$\begin{aligned} &-I_n^{(\hat{a})} \nabla_{\theta_n} g(\theta_n)^T v_n \\ &\leq -\alpha^{n-1} I_1^{(\hat{a})} \nabla_{\theta_1} g(\theta_1)^T v_1 + \sum_{t=1}^{n-1} \alpha^{n-t} c_t + \sum_{t=1}^{n-1} c\alpha^{n-t} I_t^{(\hat{a})}\|v_t\|^2 \\ &\quad - \sum_{t=1}^{n-1} \epsilon_{t+1} \alpha^{n-t-1} I_{t+1}^{(a)} \nabla_{\theta_{t+1}} g(\theta_{t+1})^T \nabla_{\theta_{t+1}} g(\theta_{t+1}, \xi_{t+1}). \end{aligned} \tag{50}$$

We multiple $I_n^{(\hat{a})}$ on both sides of (48) and note $I_n^{(\hat{a})} \geq I_n^{(\hat{a})}$, then it holds that

$$I_{n+1}^{(\hat{a})} g(\theta_{n+1}) \leq I_n^{(\hat{a})} g(\theta_n) - I_n^{(a\hat{a})} \nabla_{\theta_n} g(\theta_n)^T v_n + \frac{c}{2} I_n^{(\hat{a})}\|v_n\|^2. \tag{51}$$

Then we substitute (50) into (51) to obtain

$$\begin{aligned} I_{n+1}^{(\hat{a})} g(\theta_{n+1}) &\leq I_n^{(\hat{a})} g(\theta_n) - \alpha^{n-1} I_1^{(a)} \nabla_{\theta_1} g(\theta_1)^T v_1 + \sum_{t=1}^{n-1} \alpha^{n-t} c_t + \sum_{t=1}^{n-1} c\alpha^{n-t} I_t^{(\hat{a})}\|v_t\|^2 \\ &\quad - \sum_{t=1}^{n-1} \epsilon_{t+1} \alpha^{n-t-1} I_{t+1}^{(\hat{a})} \nabla_{\theta_{t+1}} g(\theta_{t+1})^T \nabla_{\theta_{t+1}} g(\theta_{t+1}, \xi_{t+1}) + \frac{c}{2} I_n^{(\hat{a})}\|v_n\|^2. \end{aligned} \tag{52}$$

Then we use the $am - gm$ *inequality* on the term $\sum_{t=1}^{n-1} \alpha^{n-t} c_{t-1}$ to attain

$$\begin{aligned} \sum_{t=1}^{n-1} \alpha^{n-t} c_t &= \sum_{t=1}^{n-1} \alpha^{n-t}\big(I_t^{(\hat{a})} - I_{t+1}^{(\hat{a})}\big) \nabla_{\theta_t} g(\theta_t)^T v_t \leq \sum_{t=1}^{n-1} \frac{\alpha^{n-t}}{2}\big(I_t^{(\hat{a})} - I_{t+1}^{(\hat{a})}\big) I_t^{(\hat{a})}\|\nabla_{\theta_t} g(\theta_t)\|^2 \\ &\quad + \sum_{t=1}^{n-1} \frac{\alpha^{n-t}}{2}\big(I_t^{(\hat{a})} - I_{t+1}^{(\hat{a})}\big) I_t^{(\hat{a})}\|v_t\|^2. \end{aligned} \tag{53}$$

We substitute (53) into (52), note $I_n^{(\hat{a})} - I_{n+1}^{(\hat{a})} = o(\epsilon_n)$ *a.e.* (due to $\sum_{n=1}^{+\infty} I_n^{(\hat{a})} - I_{n+1}^{(\hat{a})} < +\infty$, $\sum_{n=1}^{+\infty} \epsilon_n = +\infty$), and make the mathematical expectation, then we derive

$$\begin{aligned} \mathbb{E}(I_{n+1}^{(\hat{a})} g(\theta_{n+1})) &\leq \mathbb{E}(I_n^{(\hat{a})} g(\theta_n)) - \alpha^{n-1}\big|I_1^{(\hat{a})} \nabla_{\theta_1} g(\theta_1)^T v_1\big| + 2\sum_{t=1}^{n} c\alpha^{n-t} \mathbb{E}(I_t^{(\hat{a})}\|v_t\|^2) \\ &\quad - \sum_{t=1}^{n} (\epsilon_t - o(\epsilon_t))\alpha^{n-t} \mathbb{E}(I_t^{(\hat{a})}\|\nabla_{\theta_t} g(\theta_t)\|^2). \end{aligned} \tag{54}$$

On the other hand, we have

$$
\mathbb{E}\left(\epsilon_{n+1} I_{n+1}^{(\hat{a})} g(\theta_{n+1})\right) - \mathbb{E}\left(\epsilon_n I_n^{(\hat{a})} g(\theta_n)\right)
$$

$$
\leq -\frac{1}{2}\left(\mathbb{E}\left(I_n^{(\hat{a})}\|v_n\|^2\right) - \alpha^2\,\mathbb{E}\left(I_{n-1}^{(\hat{a})}\|v_{n-1}\|^2\right)\right) - \frac{\epsilon_n^2}{2}\,\mathbb{E}\left(I_n^{(\hat{a})}\|\nabla_{\theta_n} g(\theta_n)\|^2\right) + \frac{c}{2}\epsilon_n\,\mathbb{E}\left(I_n^{(\hat{a})}\|v_n\|^2\right)
$$

$$
+ \frac{\epsilon_n^2}{2}\,\mathbb{E}\left(I_n^{(\hat{a})}\|\nabla_{\theta_n} g(\theta_n) - \nabla_{\theta_n} g(\theta_n, \xi_n)\|^2\right).
$$

$$(55)$$

Substitute (55) into (54) yields

$$
\mathbb{E}(I_{n+1}^{(\hat{a})} g(\theta_{n+1})) - \mathbb{E}(I_n^{(\hat{a})} g(\theta_n)) \leq \mu\alpha^{n-1}\|\nabla_{\theta_1} g(\theta_1)\|^2 - \sum_{i=1}^{n}\alpha^{n-i}(\epsilon_i - o(\epsilon_i))\,\mathbb{E}\left(I_i^{(\hat{a})}\|\nabla_{\theta_i} g(\theta_i)\|^2\right)
$$

$$
+ c''\sum_{i=1}^{n}\alpha^{n-i}\epsilon_i^2\,\mathbb{E}\left(I_i^{(\hat{a})}\|\nabla_{\theta_i} g(\theta_i) - \nabla_{\theta_i} g(\theta_i, \xi_i)\|^2\right),.
$$

$$(56)$$

where $c''$ and $\mu$ are two constants which do not affect results Then we note when $\|\nabla_{\theta_n} g(\theta_n)\|^2 > \hat{a}$, there is

$$
\mathbb{E}(\|\nabla_{\theta_n} g(\theta_n) - \nabla_{\theta_n} g(\theta_n, \xi_n)\|^2) \leq \left(M' - 1 + \frac{a'}{\hat{a}}\right)\mathbb{E}\left(\|\nabla_{\theta_n} g(\theta_n)\|^2\right) \leq \hat{M}\,\mathbb{E}\left(\|\nabla_{\theta_n} g(\theta_n)\|^2\right),
$$

where $\hat{M}$ is a constant which does not affect results. So we get

$$
\sum_{i=1}^{n}\alpha^{n-i}\epsilon_i^2\,\mathbb{E}\left(I_i^{(\hat{a})}\|\nabla_{\theta_i} g(\theta_i) - \nabla_{\theta_i} g(\theta_i, \xi_i)\|^2\right) \leq \hat{M}\sum_{i=1}^{n}\alpha^{n-i}\epsilon_i^2\,\mathbb{E}\left(I_i^{(\hat{a})}\|\nabla_{\theta_i} g(\theta_i)\|^2\right). \tag{57}
$$

Substitute (57) into (56) and note $c''\hat{M}\epsilon_n^2 = o(\epsilon_n)$, then it follows that

$$
\mathbb{E}(I_{n+1}^{(\hat{a})} g(\theta_{n+1})) - \mathbb{E}(I_n^{(\hat{a})} g(\theta_n)) \leq \mu\alpha^{n-i}\|\nabla_{\theta_1} g(\theta_1)\|^2 - \sum_{i=1}^{n}\alpha^{n-i}(\epsilon_i - o(\epsilon_i))\,\mathbb{E}\left(I_i^{(\hat{a})}\|\nabla_{\theta_i} g(\theta_i)\|^2\right).
$$

$$(58)$$

We denote

$$
\hat{F}_n^{(\hat{a})} = \sum_{i=1}^{n}\left(\frac{1}{2-\alpha}\right)^{n-i}\mathbb{E}\left(I_n^{(\hat{a})} g(\theta_n)\right) \tag{59}
$$

For convenience, we let

$$
\hat{G}_n^{(\hat{a})} = \frac{2}{(1-\alpha)^2}\sum_{i=1}^{n}\left(\frac{1}{2-\alpha}\right)^{n-i}(\epsilon_i + o(\epsilon_i))\,\mathbb{E}\left(I_i^{(a)}\|\nabla_{\theta_n} g(\theta_n)\|^2\right).
$$

Through (54), we get

$$
\hat{F}_{n+1}^{(\hat{a})} - \hat{F}_n^{(\hat{a})} \leq \hat{\mu}\left(\frac{1}{2-\alpha}\right)^n - \hat{G}_n^{(\hat{a})},
$$

so there is

$$
\hat{F}_{n+1}^{(\hat{a})} \leq \hat{F}_n^{(\hat{a})}\left(1 - \frac{\hat{G}_n^{(\hat{a})}}{\hat{F}_n^{(\hat{a})}}\right) + \hat{\mu}\left(\frac{1}{2-\alpha}\right)^n
$$

$$
\leq \hat{F}_1^{(\hat{a})}\prod_{i=1}^{n}\left(1 - \frac{\hat{G}_i^{(\hat{a})}}{\hat{F}_i^{(\hat{a})}}\right) + \sum_{t=1}^{n}\hat{\mu}\left(\frac{1}{2-\alpha}\right)^n\prod_{i=n-t}^{n}\left(1 - \frac{\hat{G}_i^{(a)}}{\hat{F}_i^{(\hat{a})}}\right) \leq \hat{q}_0\prod_{i=1}^{n}\left(1 - \frac{\hat{G}_i^{(\hat{a})}}{\hat{F}_i^{(a)}}\right),
$$

$$(60)$$

where $\hat{q}_0$ is a constant. We focus on $\frac{\hat{G}_i^{(\hat{a})}}{\hat{F}_i^{(\hat{a})}}$. Using $O'stolz\ theorem$ yields

$$
\liminf_{i\to+\infty}\frac{\hat{G}_i^{(\hat{a})}}{\epsilon_i \hat{F}_i^{(\hat{a})}} = \liminf_{i\to+\infty}\frac{2}{(1-\alpha)^2}\frac{\sum_{t=1}^{i}(\frac{1}{2-\alpha})^{i-t}\mathbb{E}(I_t^{(\hat{a})}\|\nabla_{\theta_t} g(\theta_t)\|^2)}{\sum_{t=1}^{i}(\frac{1}{2-\alpha})^{i-t}\mathbb{E}(I_t^{(\hat{a})} g(\theta_t))}
$$

$$
\geq \liminf_{i\to+\infty}\frac{2}{(1-\alpha)^2}\frac{\mathbb{E}(I_i^{(\hat{a})}\|\nabla_{\theta_i} g(\theta_i)\|^2)}{\mathbb{E}(I_t^{(\hat{a})} g(\theta_i))}.
$$

Through Assumption 3.3, we know when $\theta \in S = \{\|\nabla_\theta g(\theta)\|^2 < \delta g(\theta)\}$, there is $g(\theta) < \hat{T}$. We define $\hat{S} = \{g(\theta) < 2\hat{T}\}$. Then when $\theta_i \in \hat{S}$, there is $I_i^{(\hat{a})}\|\nabla_{\theta_i} g(\theta_i)\|^2 > (\hat{a}/2\hat{T})I_i^{(\hat{a})} g(\theta_i)$, and when $\theta_i \in \mathbb{R}^N/S$, there is $I_i^{(\hat{a})}\|\nabla_{\theta_i} g(\theta_i)\|^2 > 2\delta_0 I_i^{(\hat{a})} g(\theta_i)$. We assign $\hat{c} = \min\{2\delta_0, \hat{a}/2\hat{T}\}$, then it holds that

$$
\liminf_{i\to+\infty}\frac{2}{(1-\alpha)^2}\frac{\mathbb{E}(I_i^{(\hat{a})}\|\nabla_{\theta_i} g(\theta_i)\|^2)}{\mathbb{E}(I_t^{(\hat{a})} g(\theta_i))} \geq \frac{2\hat{c}}{(1-\alpha)^2}. \tag{61}
$$

Thus, it follows that

$$\mathbb{E}(I_n^{(\hat{a})} g(\theta_n)) \leq \hat{F}_{n+1}^{(\hat{a})} = O\left(e^{-\frac{2\hat{c}}{(1-\alpha)^2} \sum_{i=1}^{n} \epsilon_n}\right). \tag{62}$$

$\square$

## C.4 Proof of Theorem 3.4.

From (31) with $\epsilon_n \equiv \epsilon$, it follows that

$$\mathbb{E}\left(\epsilon g(\theta_{n+1})\right) - \mathbb{E}\left(\epsilon g(\theta_n)\right)$$
$$\leq -\frac{1}{2}\left(\mathbb{E}\left(\|v_n\|^2\right) - \alpha^2 \mathbb{E}\left(\|v_{n-1}\|^2\right)\right) - \frac{\epsilon^2}{2}\mathbb{E}\left(\|\nabla_{\theta_n} g(\theta_n)\|^2\right) + \frac{c}{2}\epsilon \mathbb{E}\left(\|v_n\|^2\right) \tag{63}$$
$$+ \frac{\epsilon^2}{2}\mathbb{E}\left(\|\nabla_{\theta_n} g(\theta_n) - \nabla_{\theta_n} g(\theta_n, \xi_n)\|^2\right).$$

Let the event $A_n^{(\varphi)} = \left\{\|\nabla_{\theta_1} g(\theta_1)\|^2 > \varphi, \ \|\nabla_{\theta_2} g(\theta_2)\|^2 > \varphi, ..., \ \|\nabla_{\theta_n} g(\theta_n)\|^2 > \varphi\right\}$ and the characteristic function be $I_n^{(\varphi)}$, then we attain

$$\sum_{i=1}^{n} \mathbb{E}(I_i^{(\varphi)} \|v_i\|^2) \leq \hat{c} + \hat{\sigma}_0 \sum_{i=1}^{n} \epsilon^2 \mathbb{E}(I_i^{(\varphi)} \|\nabla_{\theta_i} g(\theta_i) - \nabla_{\theta_i} g(\theta_i, \xi_i)\|^2). \tag{64}$$

Using Taylor expansion, we derive that

$$I_{t+1}^{(\varphi)} g(\theta_{t+1}) - I_t^{(\varphi)} g(\theta_t)$$
$$= I_t^{(\varphi)} \nabla_{\theta_{\xi_t}} g(\theta_{\xi_t})^T (\theta_{t+1} - \theta_t) = -I_t^{(\varphi)} \nabla_{\theta_t} g(\theta_t)^T v_t \tag{65}$$
$$+ I_t^{(\varphi)} \left(\nabla_{\theta_{\xi_t}} g(\theta_{\xi_t}) - \nabla_{\theta_t} g(\theta_t)\right)^T (\theta_{t+1} - \theta_t).$$

Considering the term $\nabla_{\theta_t} g(\theta_t)^T v_t$, it holds that

$$I_t^{(\varphi)} \nabla_{\theta_t} g(\theta_t)^T v_t$$
$$= I_{t-1}^{(\varphi)} \alpha \nabla_{\theta_{t-1}} g(\theta_{t-1})^{\mathsf{T}} v_{t-1} + \alpha I_{t-1}^{(\varphi)} (\nabla_{\theta_t} g(\theta_t) - \nabla_{\theta_{t-1}} g(\theta_{t-1}))^T v_{t-1}$$
$$+ \epsilon I_{t-1}^{(\varphi)} \nabla_{\theta_t} g(\theta_t)^{\mathsf{T}} \nabla_{\theta_t} g(\theta_t, \xi_t) + (I_t^{(\varphi)} - I_{t-1}^{(\varphi)}) \nabla_{\theta_t} g(\theta_t)^T v_t.$$

Note $(I_t^{(\varphi)} - I_{t-1}^{(\varphi)}) I_{t-1}^{(\varphi)} = I_t^{(\varphi)} - I_{t-1}^{(\varphi)}$ and $I_t^{(\varphi)} - I_{t-1}^{(\varphi)} \to 0$ $a.s.$. Then by substituting the above equation into (11) and noting $-(\nabla_{\theta_{(i)}} g(\theta_{(i)}) - \nabla_{\theta_{i-1}} g(\theta_{i-1}))^{\mathsf{T}} v_{i-1} \leq \left\|\nabla_{\theta_{(i)}} g(\theta_{(i)}) - \nabla_{\theta_{i-1}} g(\theta_{i-1})\right\| \|v_{i-1}\| \leq c\|v_{t-1}\|^2$, we obtain

$$I_{t+1}^{(\varphi)} g(\theta_{t+1}) - I_t^{(\varphi)} g(\theta_t)$$
$$\leq -\alpha^{t-1} I_1^{(\varphi)} \nabla_{\theta_1} g(\theta_1)^{\mathsf{T}} v_1 - \sum_{i=2}^{t} \alpha^{t-i} \epsilon I_{i-1}^{(\varphi)} \nabla_{\theta_i} g(\theta_i)^{\mathsf{T}} \nabla_{\theta_i} g(\theta_i, \xi_i) + \frac{c}{2}\|v_t\|^2 \tag{66}$$
$$- \sum_{i=1}^{t-1} \alpha^{t-i} I_{i-1}^{(\varphi)} (\nabla_{\theta_i} g(\theta_i) - \nabla_{\theta_{i-1}} g(\theta_{i-1}))^T v_{i-1},$$

where $c > 0$ is a constant. It follows that

$$I_{t+1}^{(\varphi)} g(\theta_{n+1})$$
$$= I_1^{(\varphi)} g(\theta_1) + \sum_{t=1}^{n} \left(I_{t+1}^{(\varphi)} g(\theta_{t+1}) - I_t^{(\varphi)} g(\theta_t)\right)$$
$$= I_1^{(\varphi)} g(\theta_1) - \frac{1-\alpha^n}{1-\alpha} I_1^{(\varphi)} \nabla_{\theta_1} g(\theta_1)^{\mathsf{T}} v_1 + \frac{1-\alpha^n}{1-\alpha} \epsilon I_1^{(\varphi)} \nabla_{\theta_1} g(\theta_1) \nabla_{\theta_1} g(\theta_1, \xi_1) \tag{67}$$
$$- \sum_{t=1}^{n} \frac{1-\alpha^{n-t+1}}{1-\alpha} \epsilon I_{t-1}^{(\varphi)} \nabla_{\theta_t} g(\theta_t) \nabla_{\theta_t} g(\theta_t, \xi_t) + c'' \sum_{i=1}^{t} \|v_i\|^2,$$

where $c''$ is a constant which does not affect the result. Substituting (64) into (67) yields

$$I_{t+1}^{(\varphi)} g(\theta_{n+1})$$
$$\leq I_1^{(\varphi)} g(\theta_1) - \frac{1-\alpha^n}{1-\alpha} I_1^{(\varphi)} \nabla_{\theta_1} g(\theta_1)^{\mathsf{T}} v_1 + \frac{1-\alpha^n}{1-\alpha} \epsilon I_1^{(\varphi)} \nabla_{\theta_1} g(\theta_1) \nabla_{\theta_1} g(\theta_1, \xi_1)$$
$$- \sum_{t=1}^{n} \frac{1-\alpha^{n-t+1}}{1-\alpha} \epsilon I_{t-1}^{(\varphi)} \nabla_{\theta_t} g(\theta_t) \nabla_{\theta_t} g(\theta_t, \xi_t) + c''\hat{c} + c''\hat{\sigma}_0 \sum_{t=1}^{n} \epsilon^2 I_t^{(\varphi)} \|\nabla_{\theta_t} g(\theta_t) - \nabla_{\theta_t} g(\theta_t, \xi_t)\|^2. \tag{68}$$

Then we multiply $I_n^{(\varphi)}$ on the both sides of (68) and notice $I_n^{(\varphi)} \geq I_{n+1}^{(\varphi)}$ to derive that

$$
\begin{aligned}
&I_{n+1}^{(\varphi)} g(\theta_{n+1}) \\
\leq &I_1^{(\varphi)} g(\theta_1) - \frac{1-\alpha^n}{1-\alpha} I_1^{(\varphi)} \nabla_{\theta_1} g(\theta_1)^\mathsf{T} v_1 + \frac{1-\alpha^n}{1-\alpha} \epsilon I_1^{(\varphi)} \nabla_{\theta_1} g(\theta_1) \nabla_{\theta_1} g(\theta_1, \xi_1) \\
&- \sum_{t=1}^n \frac{1-\alpha^{n-t+1}}{1-\alpha} \epsilon I_{t-1}^{(\varphi)} \nabla_{\theta_t} g(\theta_t) \nabla_{\theta_t} g(\theta_t, \xi_t) + c'' \hat{c} + c'' \hat{\sigma}_0 \sum_{t=1}^n \epsilon^2 I_t^{(\varphi)} \| \nabla_{\theta_t} g(\theta_t) - \nabla_{\theta_t} g(\theta_t, \xi_t) \|^2.
\end{aligned}
\tag{69}
$$

Then we take the expectation on (68) and obtain

$$
\begin{aligned}
&\mathbb{E}\left( I_{n+1}^{(\varphi)} g(\theta_{n+1}) \right) \\
\leq &\mathbb{E}\left( I_1^{(\varphi)} g(\theta_1) \right) - \frac{1-\alpha^n}{1-\alpha} \mathbb{E}\left( I_1^{(\varphi)} \nabla_{\theta_1} g(\theta_1)^\mathsf{T} v_1 \right) + \frac{1-\alpha^n}{1-\alpha} \epsilon \, \mathbb{E}\left( \nabla_{\theta_1} g(\theta_1) \nabla_{\theta_1} g(\theta_1, \xi_1) \right) \\
&- \sum_{t=1}^n \frac{1-\alpha^{n-t+1}}{1-\alpha} \epsilon \, \mathbb{E}\left( I_{t-1}^{(\varphi)} \nabla_{\theta_t} g(\theta_t) \nabla_{\theta_t} g(\theta_t, \xi_t) \right) + c'' \hat{c} + c'' \hat{\sigma}_0 \sum_{t=1}^n \epsilon^2 \, \mathbb{E}\left( I_t^{(\varphi)} \| \nabla_{\theta_t} g(\theta_t) - \nabla_{\theta_t} g(\theta_t, \xi_t) \|^2 \right).
\end{aligned}
$$

Under Assumption 3.1 2), it holds that

$$
\begin{aligned}
c'' \hat{\sigma}_0 \sum_{t=1}^n \epsilon^2 \, \mathbb{E}\left( I_t^{(\varphi)} \| \nabla_{\theta_t} g(\theta_t) - \nabla_{\theta_t} g(\theta_t, \xi_t) \|^2 \right) &= c'' \hat{\sigma}_0 \sum_{t=1}^n \epsilon^2 \, \mathbb{E}\left( \mathbb{E}\left( I_t^{(\varphi)} \| \nabla_{\theta_t} g(\theta_t) - \nabla_{\theta_t} g(\theta_t, \xi_t) \|^2 \big| \mathcal{F}_t \right) \right) \\
&\leq c'' \hat{\sigma}_0 \epsilon^2 \left( M_0 + \frac{a}{\varphi} \right) \sum_{t=1}^n \mathbb{E}\left( I_t^{(\varphi)} \| \nabla_{\theta_t} g(\theta_t) \|^2 \right).
\end{aligned}
$$

It follows that

$$
\mathbb{E}\left( I_{n+1}^{(\varphi)} g(\theta_{n+1}) \right) \leq \hat{K} - \sum_{t=1}^n \left( \epsilon - c'' \hat{\sigma}_0 \epsilon^2 \left( M_0 + \frac{a}{\varphi} \right) \right) \mathbb{E}\left( I_t^{(\varphi)} \| \nabla_{\theta_t} g(\theta_t) \|^2 \right).
$$

Let $\mu_0 = 1/c'' \hat{\sigma}_0 \left( M_0 + \frac{a}{\varphi} \right)$. Then when $\epsilon < \mu_0$, it holds that $\left( \epsilon - c'' \hat{\sigma}_0 \epsilon^2 \left( M_0 + \frac{a}{\varphi} \right) \right) > 0$. We derive that

$$
\sum_{t=1}^n \varphi \, \mathbb{E}\left( I_t^{(\varphi)} \right) < \sum_{t=1}^n \mathbb{E}\left( I_t^{(\varphi)} \| \nabla_{\theta_t} g(\theta_t) \|^2 \right) < \frac{\hat{K}}{\epsilon - c'' \hat{\sigma}_0 \epsilon^2 \left( M_0 + \frac{a}{\varphi} \right)} < +\infty.
\tag{70}
$$

In addition, we have

$$
\lim_{n \to +\infty} P(A_n^{(\varphi)}) = \lim_{n \to +\infty} \mathbb{E}\left( I_t^{(\varphi)} \right) = 0.
$$

Due to $A_1^{(\varphi)} \supset A_2^{(\varphi)} \supset ... \supset A_n^{(\varphi)}$, we get

$$
P\left( \lim_{n \to +\infty} A_n^{(\varphi)} \right) = \lim_{n \to +\infty} P(A_n^{(\varphi)}) = 0.
$$

Through the arbitrariness of $\phi$, we attain the conclusion.

## C.5   Proof of Proposition 3.1.

From (70), it follows that $\mathbb{E}\left( I_n^{(\varphi)} \| \nabla_{\theta_n} g(\theta_n) \|^2 \right) \to 0$. So for any $\varphi > 0$, there exists $n_0$, when $n > n_0$, it holds that

$$
\mathbb{E}\left( I_n^{(\varphi)} \| \nabla_{\theta_n} g(\theta_n) \|^2 \right) < \varphi.
\tag{71}
$$

Then we define another event First we construct an event

$$
\hat{B}_n = \{ there\ exists\ at\ least\ one\ point\ n_0 \in [0, n],\ satisfying\ g(\theta_{n_0}) - g^* \leq \varphi \},
$$

where $k_0$ is an undetermined coefficient Then we define another events

$$
B_i = \{ g(\theta_i) - g^* < \varphi \},
$$

$$
B_{i,j} = \{ g(\theta_i) - g^* \leq \varphi\ and\ for\ all\ n_0 \in (i, j]\ g(\theta_n) - g^* > \varphi \}.
$$

Specially, we denote $B_{n,n} = B_n$. There is

$$
\hat{B}_n = \bigcup_{i=1}^n B_{i,n}.
$$

Let the characteristic function of $B_n$ be $\hat{I}_n$, $B_{i,j}$ be $\hat{I}_{i,j}$. It is obvious that

$$
\mathbb{E}\left( \hat{I}_{n,n}(g(\theta_n) - g^*) \right) \leq \varphi \, \mathbb{E}(\hat{I}_{n,n}) \leq \varphi.
\tag{72}
$$

Then from (68) we can get that

$$\hat{I}_{i,n+1}\hat{g}(\theta_{n+1}) - \hat{I}_{i,n}\hat{g}(\theta_n) \leq \mu\alpha^{n-i}\hat{I}_{i,i+1}\|\nabla_{\theta_{i+1}}g(\theta_{i+1})\|^2 - \sum_{j=i+1}^{n}\alpha^{n-j}\epsilon\hat{I}_{i,j}\nabla_{\theta_j}g(\theta_j)^T\nabla_{\theta_j}g(\theta_j,\xi_j)$$

$$+ c''\sum_{j=i+1}^{n}\alpha^{n-j}\epsilon^2\hat{I}_{i,j}\|\nabla_{\theta_j}g(\theta_j) - \nabla_{\theta_j}g(\theta_j,\xi_j)\|^2,$$

(73)

where $\mu > 0$ is a constant. Then we denote $\hat{g} = g - \varphi$. We take the mathematical expectation on (73) to obtain

$$\mathbb{E}\left(\hat{I}_{i,n+1}\hat{g}(\theta_{n+1})\right) - \mathbb{E}\left(\hat{I}_{i,n}\hat{g}(\theta_n)\right) \leq \mu\alpha^{n-i}\mathbb{E}\left(\|\nabla_{\theta_{i+1}}g(\theta_{i+1})\|^2\right)$$

$$- \sum_{j=i+1}^{n}\alpha^{n-j}\epsilon\mathbb{E}\left(\hat{I}_{i,j}\|\nabla_{\theta_j}g(\theta_j)\|^2\right)$$

$$+ c''\sum_{j=i+1}^{n}\alpha^{n-j}\epsilon^2\left(M' + \frac{a}{\varphi}\right)\mathbb{E}\left(\hat{I}_{i,j}\|\nabla_{\theta_j}g(\theta_j)\|^2\right)$$

$$= \mu\alpha^{n-i}\mathbb{E}\left(\hat{I}_{i,i+1}\|\nabla_{\theta_{i+1}}g(\theta_{i+1})\|^2\right)$$

$$- \left(\epsilon - c''\epsilon^2\left(M' + \frac{a}{\varphi}\right)\right)\sum_{j=i+1}^{n}\alpha^{n-j}\mathbb{E}\left(\hat{I}_{i,j}\|\nabla_{\theta_j}g(\theta_j)\|^2\right).$$

We construct a variable $F_{i,n+1}$ as follows

$$F_{i,n+1} = \sum_{t=i+1}^{n+1}\left(\frac{1}{2-\alpha}\right)^{n+1-t}\mathbb{E}\left(\hat{I}_{i,t}\hat{g}(\theta_t)\right).$$

So we can get

$$F_{i,n+1} - F_{i,n} \leq \sigma_0\left(\frac{1}{2-\alpha}\right)^{n-i}\mathbb{E}\left(\hat{I}_{i,i+1}\|\nabla_{\theta_{i+1}}g(\theta_{i+1})\|^2\right) - F_0\sum_{t=i+1}^{n}\left(\frac{1}{2-\alpha}\right)^{n-t}\epsilon\mathbb{E}\left(\hat{I}_{i,t}\|\hat{g}(\theta_t)\|^2\right),$$

where $\sigma_0$ and $F_0$ are two constants not dependent on $\epsilon$ or $\varphi$. We let

$$Q_{i,n} := F_0\sum_{t=i+1}^{n}\left(\frac{1}{2-\alpha}\right)^{n-t}\epsilon\mathbb{E}\left(\hat{I}_{i,t}\|\hat{g}(\theta_t)\|^2\right).$$

Then we get

$$F_{i,n+1} \leq F_{i,n}\left(1 - \frac{Q_{i,n}}{F_{i,n}}\right) + \sigma_0\left(\frac{1}{2-\alpha}\right)^{n-i}\mathbb{E}\left(\hat{I}_{i,i+1}\|\nabla_{\theta_{i+1}}g(\theta_{i+1})\|^2\right).$$

Then we get $Q_{i,n}/F_{i,n} > \epsilon F_0/\varphi$. So we can attain

$$F_{i,n+1} \leq F_{i,i+1}(1 - \epsilon\varphi F_0)^{n-i-1} + y_0^{n-i-1+}\sigma_0\mathbb{E}\left(\hat{I}_{i,i+1}\|\nabla_{\theta_{i+1}}g(\theta_{i+1})\|^2\right),$$

(74)

where $y_0 = \max\{1 - \epsilon\varphi F_0, 1/2 - \alpha\}$. Then we first prove $\mathbb{E}(g(\theta_n))$ is uniformly bounded. We have

$$\mathbb{E}\left(\hat{g}(\theta_{i+1})\right) - \mathbb{E}\left(\hat{g}(\theta_i)\right) = -\mathbb{E}\left(\nabla_{\theta_i}g(\theta_i)^T v_i\right) = -\mathbb{E}\left(\nabla_{\theta_i}g(\theta_i)^T(\alpha v_{i-1} + \epsilon\nabla_{\theta_i}g(\theta_i,\xi_i))\right)$$

$$\leq \mathbb{E}\left(\|\nabla_{\theta_i}g(\theta_i)\|\|v_{i-1}\|\right) + \epsilon\mathbb{E}\left(\|\nabla_{\theta_i}g(\theta_i))\|^2\right) \leq \sqrt{\varphi}\|v_{i-1}\| + \epsilon\varphi < \hat{m},$$

where $\hat{m}$ is a constant. Then we get

$$\mathbb{E}\left(g(\theta_n)\right) \leq \varphi + \mathbb{E}\left(\hat{g}(\theta_n)\right) \leq \varphi + \hat{m}\sum_{i=0}^{+\infty}(1 - \epsilon\varphi F_0)^i + \hat{m}\sum_{i=0}^{+\infty}y_0^i < \frac{\hat{K}}{\epsilon\varphi} < +\infty.$$

(75)

Now we have proved $\mathbb{E}\left(g(\theta_n)\right)$ uniformly bounded. Then we aim to prove $\mathbb{E}\left(g(\theta_n)\right)$ can become arbitrarily small if $n$ sufficient large and $\epsilon$ sufficient small. First We have

$$\mathbb{E}\|v_n\|^2 = \mathbb{E}\|\theta_{n+1} - \theta_n\|^2 = \left\|\alpha v_{n-1} + \epsilon\nabla_{\theta_n}g(\theta_n,\xi_n)\right\|^2 = \mathbb{E}\left\|\sum_{i=1}^{n}\alpha^{n-i}\epsilon\nabla_{\theta_i}g(\theta_i,\xi_i)\right\|^2$$

$$\leq \frac{1}{1-\alpha}\sum_{i=1}^{n}\alpha^{n-i}\epsilon^2\mathbb{E}\left\|\nabla_{\theta_i}g(\theta_i,\xi_i)\right\|^2 \leq \frac{M'}{1-\alpha}\sum_{i=1}^{n}\alpha^{n-i}\epsilon^2\mathbb{E}\left\|\nabla_{\theta_i}g(\theta_i)\right\|^2 + \frac{a}{1-\alpha}\sum_{i=1}^{n}\alpha^{n-i}\epsilon^2 \leq \hat{p}_0\frac{\epsilon}{\varphi},$$

where $\hat{p}_0$ is a constant. Then we get that

$$\mathbb{E}\left(\hat{g}(\theta_{i+1})\right) - \mathbb{E}\left(\hat{g}(\theta_i)\right) = -\mathbb{E}\left(\nabla_{\theta_i}g(\theta_i)^T v_i\right) = -\mathbb{E}\left(\nabla_{\theta_i}g(\theta_i)^T(\alpha v_{i-1} + \epsilon\nabla_{\theta_i}g(\theta_i,\xi_i))\right)$$

$$\leq \sum_{i=1}^{n}\alpha^{n-i}\|v_i\|^2 \leq \hat{p}_1\frac{\epsilon}{\varphi}.$$

Since $\mathbb{E}\left(\hat{g}(\theta_i)\right) \leq 0$, we get $\mathbb{E}\left(\hat{g}(\theta_{i+1})\right) \leq \hat{p}_1\epsilon/\varphi$. Finally we substituted it into (74), getting

$$\mathbb{E}\left\|\nabla_{\theta_n}g(\theta_n)\right\|^2 \leq \mathbb{E}\left(g(\theta_n)\right) \leq \varphi + \mathbb{E}\left(\hat{g}(\theta_n)\right) \leq \varphi + \hat{p}_1\frac{\epsilon}{\varphi}\sum_{i=0}^{+\infty}(1-\epsilon\varphi F_0)^i + \hat{p}_1\frac{\epsilon}{\varphi}\sum_{i=0}^{+\infty}y_0^i < \frac{\hat{p}_2}{\varphi},$$

where $\hat{p}_2$ is a constant. Then we can get

$$\mathbb{E}\|v_n\|^2 = \mathbb{E}\|\theta_{n+1} - \theta_n\|^2 = \left\|\alpha v_{n-1} + \epsilon\nabla_{\theta_n}g(\theta_n,\xi_n)\right\|^2 = \mathbb{E}\left\|\sum_{i=1}^{n}\alpha^{n-i}\epsilon\nabla_{\theta_i}g(\theta_i,\xi_i)\right\|^2$$

$$\leq \frac{1}{1-\alpha}\sum_{i=1}^{n}\alpha^{n-i}\epsilon^2\,\mathbb{E}\left\|\nabla_{\theta_i}g(\theta_i,\xi_i)\right\|^2 \leq \frac{M'}{1-\alpha}\sum_{i=1}^{n}\alpha^{n-i}\epsilon^2\,\mathbb{E}\left\|\nabla_{\theta_i}g(\theta_i)\right\|^2 + \frac{a}{1-\alpha}\sum_{i=1}^{n}\alpha^{n-i}\epsilon^2 \leq \hat{p}_0'\frac{\epsilon^2}{\varphi},$$

where $\hat{p}_0'$ is a constant. Then we get that

$$\mathbb{E}\left(\hat{g}(\theta_{i+1})\right) - \mathbb{E}\left(\hat{g}(\theta_i)\right) = -\mathbb{E}\left(\nabla_{\theta_i}g(\theta_i)^T v_i\right) = -\mathbb{E}\left(\nabla_{\theta_i}g(\theta_i)^T(\alpha v_{i-1} + \epsilon\nabla_{\theta_i}g(\theta_i,\xi_i))\right)$$

$$\leq \sum_{i=1}^{n}\alpha^{n-i}\|v_i\|^2 \leq \hat{p}_1'\frac{\epsilon^2}{\varphi},$$

where $\hat{p}_1'$ is a constant. Then we can get

$$\mathbb{E}\left\|\nabla_{\theta_n}g(\theta_n)\right\|^2 \leq \mathbb{E}\left(g(\theta_n)\right) \leq \varphi + \mathbb{E}\left(\hat{g}(\theta_n)\right) \leq \varphi + \hat{p}_1'\frac{\epsilon^2}{\varphi}\sum_{i=0}^{+\infty}(1-\epsilon\varphi F_0)^i + \hat{p}_1\frac{\epsilon}{\varphi}\sum_{i=0}^{+\infty}y_0^i < \frac{\hat{p}_2'\epsilon}{\varphi},$$

where $\hat{p}_2'$ is a constant. Now we can find that provided $\epsilon$ small enough, we can get for any small bound $\varphi_0$, $\mathbb{E}\left\|\nabla_{\theta_n}g(\theta_n)\right\|^2 < \varphi_0$ (when $n$ is large enough).