# OpenReview forum: "Revisit last-iterate convergence of mSGD under milder requirement on step size"
_NeurIPS.cc/2022/Conference — NeurIPS 2022 Accept_

### Official Review · Reviewer_Kgxh · 2022-07-10

**Rating:** 5
**Confidence:** 4
**Soundness:** 3 good
**Presentation:** 3 good
**Contribution:** 2 fair

**Summary:**

This work considers mSGD and SGD under more weaker conditions on the step size as considered before. This can be important since in practice, picking a large step size initially is shown to improve convergence. Therefore, a theory of convergence with large step sizes can be useful. It is standard to consider step sizes such that $\sum_t \epsilon_t = \infty$ and $\sum_t \epsilon_t^2 < \infty$ whereas this work improves this to $\sum_t \epsilon_t^{2+\eta_0} < \infty$. Under these conditions, along with standard assumptions like smoothness, asymptotic convergence to a stationary point is established (Theorem 3.1).

Under additional growth assumptions of the function, the Theorem 3.2 establishes rates of convergence as a function of the step size sequence, which shows the dependence of the momentum parameter alpha. However, since this does not establish how picking larger step sizes can be effective, the authors show Theorem 3.3 where the upper bounds are such that picking a larger step size can help the algorithm reach a point better than the initial point more reliably.

The proof techniques and assumptions mostly seem to be modifications of [7].

[7] R. Jin, Y. Xing, and X. He, “On the convergence of mSGD and AdaGrad for stochastic optimization,” in
312 International Conference on Learning Representations, 2022

**Questions:**

1. Given the observations above, how do we justify using larger step sizes based on the theoretical results provided?

2. The authors should to carefully compare the techniques used in [7] to the current work since a lot of the main techniques and assumptions seem very similar. They currently mainly compare the results obtained in [7].

3. Below Theorem 3.2, the authors discuss that [7] uses the condition $\mathbb{E}\|\nabla g(\theta,\zeta)\|^2 \leq M \|\nabla g(\theta)\|^2$. However, when I checked the ICLR 2022 paper, this work did not contain this assumption in Theorem 2 (See (4) in Assumption 2). However, I do agree Theorem 2 in [7] cannot be established without such an assumption and the original work [7] could have a typo. The authors should elaborate more on this.

**Limitations:**

The authors have been forthright about several of the weaknesses in the analysis in their discussions (as pointed above), however several of the limitations need to be discussed more as pointed out in the rest of the review.

**Strengths And Weaknesses:**

*Strengths*:
The work establishes convergence for widely used stochastic gradient methods in regimes which have not been considered in the literature but nevertheless seem very important.

*Weaknesses*:
The work does not justify why picking larger step sizes is useful. Theorem 3.2, like the authors discuss, does not show that picking a step size sequence such that $\sum_t \epsilon_t^2 = \infty$ is helpful in terms of convergence rates.

Theorem 3.3, on the other hand, considers a non-standard stopping time $\tau^{(a)}$. This seems to be quite a weak metric to compare algorithms since this is the first time when the algorithm's output is better than the initial point. Also, arguing about how certain step sizes are better based on upper bounds is fallacious since we do not know if the upper bounds are sharp.

---

> ### Author Response · Authors · 2022-08-02
> **Response**
>
> Response: We are grateful for the reviewer's recommendation. We appreciate your valuable comments and suggestions, which have helped us improve the presentation of the paper substantially. The revised expressions of the paper are marked in blue.
>
> Regarding how to justify using larger step sizes, we have made some improvement: For the first-time reaching probability, we have calculated a lower bound, which is $P({\tau}^{(a)}\ge n)\ge ke^{-\frac{4c}{(1-\alpha)^{2}}\sum_{k=1}^{n}\epsilon_{k}}.$ Through this bound,  we can rigorously prove that when $n$ is sufficiently large, the iterates under larger step size can reach an arbitrarily small neighborhood defined with $a$ within $n$ steps with a larger probability. This essentially means that the algorithm becomes stronger in  pulling back the abnormal values. This is because that  as iteration index goes on, if the gradient norm jumps out of the neighborhood at some step, we can reinitialize the algorithm with this value. Then larger step size can make the gradient norm back to the neighborhood again with fewer steps. According to Theorem 3.1, the algorithm is also convergent a.s. under the larger step size, so for the cases where the initial value is far from the optimal one, larger step size can make the convergence to the neighborhood faster and accomplish asymptotic convergence eventually.
>
>  Regarding the comparison with [7], our analysis method for  convergence contains more techniques. In [7], the authors used $\sum_{n=1}^{+\infty}\epsilon_{n}^{2}<+\infty$ to obtain $\sum_{n=1}^{+\infty}\epsilon_{n}\\|\nabla_{\theta_{n}}g(\theta_{n})\\|^{2}<+\infty\ \ a.s.$ and $g(\theta_{n})$ convergence a.s.. Since we study the problem under a relaxed condition  $\sum_{n=1}^{+\infty}\epsilon_{n}^{2+\eta_{0}}<+\infty$,
> using the method as [7], we can only obtain $\sum_{n=1}^{+\infty}\epsilon_{n}^{1+\eta_{0}}\\|\nabla_{\theta_{n}}g(\theta_{n})\\|^{2}<+\infty$，$\sum_{n=1}^{+\infty}\epsilon_{n}^{1+2\eta_{0}}\\|\nabla_{\theta_{n}}g(\theta_{n})\\|^{2}<+\infty$ and $\epsilon_{n}^{\eta_{0}}g(\theta_{n})$ convergence a.s., and $\epsilon_{n}^{\eta_{0}}g(\theta_{n})$ convergence a.s. can not conclude $g(\theta_{n})$ convergence directly.  Thus, we turn to  another more technical method as follows. First, we prove that （Lemma A.11）
> $$g(\theta_{n+1})-g(\theta_{n})\le k\epsilon_{n}^{1+2\eta_{0}}+Q_{t},$$where $\sum_{n=1}^{+\infty}Q_{t}<+\infty\ \ a.s.$. In addition, by using $\sum_{n=1}^{+\infty}\epsilon_{n}^{1+2\eta_{0}}\\|\nabla_{\theta_{n}}g(\theta_{n})\\|^{2}<+\infty$, we show that 0 is a stationary point of  $\{\\|\nabla_{\theta_{n}}g(\theta_{n})\\|^{2}\}$. After further analysis and proof, we attain the result of the paper. Furthermore, on convergence rate, since [7] used the strong growth condition,  i.e. $E_{\xi_{n}}\\|\nabla_{\theta}g(\theta,\xi_{n})\\|^{2}<M\\|\nabla_{\theta}g(\theta)\\|^{2}$ (discussed in the following response) while we do not, there are several major differences between our method and [7].   In the revised paper, we have carefully compared the techniques used in [7] to our work.
>
> Regarding the condition below Theorem 3.2, this condition is not mentioned in the main body of [7], but employed in the appendix on page 30, which is quoted that `Then we make some transformations, take absolute values, take the mathematical expectation, and
> use same techniques in Theorem 1. Since the sampling noise follows a uniform distribution, there is $E_{\xi_{n}}\\|\nabla_{\theta}g(\theta,\xi_{n})\\|^{2}\le M\\|\nabla_{\theta}g(\theta)\\|^{2}$.'

---

### Official Review · Reviewer_947T · 2022-07-10

**Rating:** 7
**Confidence:** 3
**Soundness:** 4 excellent
**Presentation:** 4 excellent
**Contribution:** 3 good

**Summary:**

The authors improve the step size requirement for mSGD and prove the convergence result. Specifically, they have extra assumption condition 2 in Assumption 3.1 then they allow the step size to decay more slowly. Additionally, they prove the subsequence convergence with constant step size. They implement the experiments to verify their theoretical results.

**Questions:**

1) I am not totally sure if Condition 2 in Assumption 3.1is a reasonable one. Is the set of stationary points bounded? For overparameterized models, I believe it's not. Is it critical in the proof? Can you clarify it a bit more?

2) The equation in Assumption 3.3 is very close to PL condition. In my understanding, you want the loss to decrease exponentially out of this set? I suggest the authors have more discussion about the relation to PL condition.

3) Can you clarify why we care about the convergence result for subsequence i.e., Theorem 3.4?


**Limitations:**

yes

**Strengths And Weaknesses:**

Strengths:

The problem studied in the paper is very interesting and important for the community. The authors improve the results which are verified by the experiments. The neural network example to support Assumption 3.3 is nice.

Weaknesses:

I don't see any major weakness. Maybe the authors can point out the intuition of the extra assumption (Condition 2 in Assumption 3.1) and how to use it to prove the results.

---

> ### Author Response · Authors · 2022-08-02
> **Response**
>
> Response: We are grateful for the reviewer's recommendation. We appreciate your valuable comments and suggestions, which have helped us improve the presentation of the paper substantially. The revised expressions of the paper are marked in blue.
>
> Regarding condition 2 in Assumption 3.3, it can be met in many cases, since it allows the loss function to have multiple stationary points and to be non-convex. However, it is worth mentioning that this boundedness condition can be replaced with a relaxed one: The set of the stationary points is not empty and the distance between  any two connected components of stationary points has a lower bound,  i.e., $\inf_{i,j}d(J_{i},J_{j})>z_{0}>0.$ With this condition, overparameterized models can be covered. However, this condition can make the proof very  tedious without making extra difficulties.  We use the current condition for the brevity.
>
> Regarding the connection between  Assumption 3.3 and PL condition, we note that if PL holds, then Ass 3.3 holds, but not vice versa. This is because if PL condition with parameter $\mu>0$ holds, we can let $\delta_0=\mu$, so that  $S=\{\theta|\\|\nabla_{\theta}g(\theta)\\|^{2}/(g(\theta)-g^{*})<\delta_{0}\}=\emptyset$. As we know that empty set is a subset of any set, Assumption 3.3 holds. On the other hand, if PL condition does not hold, Assumption 3.3 may be still met under the two general situations: 1) There are multiple stationary points holding different  values of loss function;  2) The loss function is bounded.  Thus, Assumption 3.3 is much milder than PL condition.
> In the revised paper, we have added more discussion on the relationship between our assumptions and the ones from existing works.
>
>  Regarding the importance of subsequence convergence in  Theorem 3.4,  subsequence convergence ensures that  the minimum value within the first $n$ steps remain in a neighborhood of zero, i.e.,  $ \limsup_{n\rightarrow+\infty}\min_{0\le k \le n}\\|\nabla_{\theta_{k}}g(\theta_{k})\\|^{2}\le \varphi\ a.s..$ As known that   minimum value is meaningful in practice,  one can employ the iterates $\theta_{n}^{'}=\arg\min_{0\le k\le n}\\|\nabla_{\theta_{k}}g(\theta_{k})\\|^{2}$.

---

### Official Review · Reviewer_61dG · 2022-07-11

**Rating:** 7
**Confidence:** 4
**Soundness:** 3 good
**Presentation:** 2 fair
**Contribution:** 3 good

**Summary:**

The paper proves last-iterate convergence for SGD with and without momentum for larger step sizes and weaker assumptions. In particular, the paper provides asymptotic convergence rates for mSGD when the learning rate decay is slower than what was needed by previous works, hence showing theoretically why mSGD can work with larger learning rates than shown by previous theory. The paper also provides non-asymptotic rates under a more general assumption than strong convexity. The paper has theorems 3.3 and 3.2 which show that even though larger step size may lead to one iterate being very close to a stationary point, the expected error at a fixed iteration number might not necessarily get smaller with larger step size.

**Questions:**

1. Although Theorem 3.2 hints that larger step sizes might not necessarily lead to faster convergence, to actually prove that it is indeed the case, a lower bound needs to be shown. Is there any lower bound existing in literature that matches or closely follows your upper bound from Theorem 3.2? If not, then a discussion should be added in the paper about this.

2. It will be nice to have some experiments that show that large step sizes that satisfy assumption 3.2 can still be slower for convergence than smaller step sizes, to support the discussion for Theorem 3.2 (lines 190-192).

**Limitations:**

Most of the results in the paper are asymptotic in nature and only prove convergence in the limit (that is, they do not have asymptotic convergence rates). Hence they do not provide much information about the actual rates of convergence.

**Strengths And Weaknesses:**

Strengths:
1. The paper proves asymptotic convergence for mSGD for learning rates that decay slower than what was proved by previous theory.
2. The paper proves non-asymptotic convergence rates for mSGD in a milder condition than strong convexity.
3. The paper proves that for larger step sizes, indeed convergence is faster in the sense that there will be one iterate that is closer to the optimizer (Theorem 3.3)



Weaknesses:
1. Most of the results in the paper are asymptotic in nature and only prove convergence in the limit (that is, they do not have asymptotic convergence rates). Hence they do not provide much information about the actual rates of convergence.


Suggestions:
1. The paper should include proof sketches or ideas for the main theorems in the main paper. Since the part where Assumption 3.3 is demonstrated for shallow neural networks is not rigorous, maybe that can be moved to the appendix to make space for proof sketches.


NOTE: I have not verified the proofs in the appendix.

---

> ### Author Response · Authors · 2022-08-02
> **Response**
>
> Response: Thanks a lot for the reviewer's recommendation. We appreciate your valuable comments and suggestions, which have helped us improve the presentation of the paper substantially. The revised expressions of the paper are marked in blue.
>
>
> Regarding the weakness mentioned by the reviewer, we note that Theorem 3.2 provides an asymptotic mean-square convergence rate, which shows how the important parameters affect the convergence rate.
>
> In the revised version, we have added the proof sketches for the main results and moved the neural network case to the appendix.
>
>  Regarding question 1, it is usually to work on the upper bound of convergence rate, since for any algorithm, if the initial value is very special, the algorithm can converge in finite steps, which means 0 is the  lower bound   making the analysis   meaningless.  Our upper bound of the convergence rate is   tight under some conditions. For instance, for a simple case $g=x^{2}$ under i.i.d  Gaussian noise, i.e., $g'(x,\xi_{n})=2x+N_{0}$,   we see that the convergence rate is equal to $$k_{0}\Big(e^{-\frac{4}{(1-\alpha)^{2}}\sum_{i=1}^{n}\epsilon_{i}}\Big)\Big(\sum_{i=1}^{n}\epsilon_{i}^{2}e^{\frac{4}{(1-\alpha)^{2}}\sum_{k=1}^{i}\epsilon_{k}}\Big)$$.
>
>  Regarding question 2, there have been many experiment results,  showing that a larger step size works well in the early stage but not in the later stage. We refer the reviewer to  [22], which used   a constant step size that is a kind of large step size.

---

### Official Review · Reviewer_hkDq · 2022-07-11

**Rating:** 7
**Confidence:** 2
**Soundness:** 3 good
**Presentation:** 3 good
**Contribution:** 3 good

**Summary:**

This paper establishes the convergence result of the last iterate of momentum algorithms. In contrast to the existing results that require the step size to be squared summable, this paper allows a larger spectrum of step size rules and even constant step size. The proof technique is different from the standard martingale-based argument.

**Questions:**

1. I didn't check all proofs, but I think it would be helpful to sketch the proof of Theorem 3.1, in addition to mentioning the main challenges. In particular, how does the proof technique differ from those in [7]?

2.  I am wondering if the rate of $2+\eta_0$, $\eta\in[0,1/2)$, is the best achievable in the setting of Section 3.1. This can be answered either theoretically, or numerically.

3. For Proposition 3.1, the result improves the previous one [22] that imposes strong convexity. But is there any substantial difference in terms of the proof compared with [22]?

4. Minor: I cannot find the definition of $\alpha^{(n)}$ in line 121.

**Limitations:**

This is a theoretical work so I think there's not much potential negative societal impact. The authors do not discuss much about the limitations of their results.

**Strengths And Weaknesses:**

Strength:

1. The presentation is clear, and the authors make efforts to explain the meaning and use of each assumption, condition in their results, and compare their results to relevant literature.

2. From the practical point of view, it establishes provable convergence for larger step sizes, so the results look nice to me, although I am not an expert in this field.

3. The additional numerical results certify the theoretical analysis.

Weakness:

1. I am not sure how strong Assumption 3.3 is, even though it is weaker than the P-L condition. The paper illustrates that this assumption holds for a two layer ReLU network, but I doubt it works for more layers.

2. Some illustration of the proof seems important. See my questions below.

---

> ### Author Response · Authors · 2022-08-02
> **Response**
>
>  We are grateful for the reviewer's recommendation. We appreciate the valuable comments and suggestions, which have helped us improve the presentation of the paper substantially. The revised expressions of the paper are marked in blue.
>
> We have added  proof sketches for our main results including Theorem 3.1.
>
> Regarding the comparison with [7], our analysis method for  convergence contains more techniques. In [7], the authors used $\sum_{n=1}^{+\infty}\epsilon_{n}^{2}<+\infty$ to obtain $\sum_{n=1}^{+\infty}\epsilon_{n}\\|\nabla_{\theta_{n}}g(\theta_{n})\\|^{2}<+\infty\ \ a.s.$ and $g(\theta_{n})$ convergence a.s.. Since we study the problem under a relaxed condition  $\sum_{n=1}^{+\infty}\epsilon_{n}^{2+\eta_{0}}<+\infty$,
> using the method as [7], we can only obtain $\sum_{n=1}^{+\infty}\epsilon_{n}^{1+\eta_{0}}\\|\nabla_{\theta_{n}}g(\theta_{n})\\|^{2}<+\infty$，$\sum_{n=1}^{+\infty}\epsilon_{n}^{1+2\eta_{0}}\\|\nabla_{\theta_{n}}g(\theta_{n})\\|^{2}<+\infty$ and $\epsilon_{n}^{\eta_{0}}g(\theta_{n})$ convergence a.s., and $\epsilon_{n}^{\eta_{0}}g(\theta_{n})$ convergence a.s. can not conclude $g(\theta_{n})$ convergence directly. Thus, we turn to  another more technical method as follows. First, we prove that （Lemma B.11）
> $$g(\theta_{n+1})-g(\theta_{n})\le k\epsilon_{n}^{1+2\eta_{0}}+Q_{t},$$where $\sum_{n=1}^{+\infty}Q_{t}<+\infty\ \ a.s.$ In addition, by using $\sum_{n=1}^{+\infty}\epsilon_{n}^{1+2\eta_{0}}\\|\nabla_{\theta_{n}}g(\theta_{n})\\|^{2}<+\infty$, we show that 0 is a stationary point of  $\{\\|\nabla_{\theta_{n}}g(\theta_{n})\\|^{2}\}$. After further analysis and proof (omitted here for brevity), we attain the result of the paper.
>
> Regarding the  rate of step size, we note that this setting condition is a sufficient condition and it remains challenging to find the best step size setting. We would like to work on it as a future work.
>
> Regarding the difference with [22] using strong convexity, our method contains more techniques. We take SGD for example. Under the strong convexity or PL condition,  one can derive
> $$E (g(\theta_{n+1}))-E(g(\theta_{n}))\le -\epsilon E\\|\nabla_{\theta_{n}}g(\theta_{n})\\|^{2}+\frac{c}{2}\epsilon^{2}\le -l_{0}\epsilon E(g(\theta_{n}))+\frac{c}{2}\epsilon^{2}.$$
> Then the result is obtained  by solving the above recursion. However, it would be much more difficult to analyze, if  strong convexity or PL condition is not met. That is why we use much space to derive the results. Please check our proof for more detail and involved techniques.
>
> The notion $\alpha^{(n)}$ stands for the term of $\alpha$, since in some literature this term is time-varying .

---

### Official Review · Reviewer_aWD4 · 2022-07-12

**Rating:** 6
**Confidence:** 2
**Soundness:** 3 good
**Presentation:** 3 good
**Contribution:** 3 good

**Summary:**

The paper considers SGD with momentum (and classical SGD if momentum coefficient is equal to zero) and provides last-iterate convergence rates for such an algorithm with (i) allowing for the bigger stepsizes, and (ii) by relaxing the PL-condition.


**Questions:**

7. What would be the optimal stepsize to give the fastest convergence?

8. Can you show some advantages in the theory of using momentum ?

9. how “a” is defined in theorem 3.2. Should it be a’ ? if not, what is the dependance on a’ in the convergence rate ?


**Limitations:**

-

**Strengths And Weaknesses:**

1. Overall, understanding the theoretical convergence of SGD and momentum SGD is an important problem in machine learning. I think it is an interesting and solid result that both SGD and momentum SGD allow for the larger learning rates (even without improving the convergence rate).

2. The paper misses the discussion on the tightness of the obtained result: is the obtained convergence rate tighter than existing ones (also the ones that use PL-condition or strong-convexity)? Also, does it recover known convergence rates of momentum SGD for the average iterates? Are obtained results recover best-known last iterate convergence rates of SGD when momentum is set to zero?

3. I found Theorem 3.1 to be confusing. I quickly checked the proof and it seems that you basically show there that ||\nabla g(theta_n) || -> 0 as n -> \infty. I think it would be less confusing to state the theorem in terms of the norm of the gradient. I also did not find that you prove that \theta_n stays bounded, thus it can happen that ||\theta_n|| -> \infty, e.g. if the function behaves similarly to e^{-theta} for \theta -> + \infty. Thus the algorithm might not converge to any connected component of stationary points.

4. Regarding presentation, assumptions 3.1, 2) and 3.3. are not common in the literature. It would be useful for the reader to spend more time to explain the connections to the common assumptions in the literature: e.g. for assumption 3.3. explaining how does delta_0 relates to the strong convexity parameter, or \mu in PL condition (I think delta_0 <= \mu). It would be also helpful to provide a table with existing last-iterate convergence results under

5. Overall the proofs are hard to read.

6. Less important, I found the notation in the paper to be confusing, usually the stochastic noise is denoted as \sigma, and lipschitz smoothness as L.

---

> ### Author Response · Authors · 2022-08-02
> **Response**
>
> Summary of responses to  Reviewer aWD4:
>
> Response: Many thanks for the reviewer's recommendation. We appreciate the valuable comments and suggestions, which have helped us improve the presentation of the paper substantially. The revised expressions of the paper are marked in blue.
>
>  Regarding the comparison with the existing ones on convergence rate, we should point out that there are few works on the convergence rate of last iterate, where references
> [7] and [20] cited in the paper are  typical. As we discussed in the paper, some assumptions we make are quite from different from [7] and [20]. For instance,  [7] required a condition on strong growth, i.e.,  $E_{\xi_{n}}\\|\nabla_{\xi_{n}}g(\theta,\xi_{n})\\|^{2}\le M_{0}\\|\nabla_{\theta}g(\theta)\\|^{2}$, which however makes it close to the deterministic case. In contrast, our assumption allows more randomness of data sampling, which  inevitably  sacrifices some convergence rate in certain cases, just like the convergence rate gap between gradient descent (GD) and stochastic GD.  The results in [20] were established for a different version of mSGD under the assumption that the loss function is bounded. Thus, it may not be appropriate to make a comparison with it.
> There are quite a few results on the average iterates in the literature, such as  [31,32,13].  In [31,32,13], the convergence rate $O(\frac{1}{\sqrt{n}})$ of mSGD  is established with the step size $\frac{1}{\sqrt{n}}$. According to our result, let $\epsilon_n=\frac{1}{\sqrt{n}},$  if holds that
> $$E(\\|\nabla_{\theta_{n}}g(\theta_{n})\\|^{2})=O\Big(e^{-\sqrt{n}}\sum_{k=1}^{n}\frac{e^{\sqrt{k}}}{k}\Big)=O\Big(\frac{1}{\sqrt{n}}\Big).$$ It can be proved that the convergence rate of average iterates is also $O\Big(\frac{1}{\sqrt{n}}\Big).$ Based on the above discussion, we have added more comparisons with the existing results in the revised version.
>
> Regarding Theorem 3.1, we totally agree with the reviewer that using the norm of the gradient is better to show the result. In the revised version, we have updated the statement. 	Regarding $\theta_{n}$, we have the following remarks:
>
> 1) If the loss function has a positive lower bound at infinity, i.e., $\liminf_{\theta\rightarrow \infty}\\|\nabla_{\theta}g(\theta)\\|>0,$ there is a connected set $J*$ include to $J$ such that the {iterate} $\theta_n$ is convergent to the set $J*$ almost surely, i.e.,  $$\lim_{n\rightarrow\infty} d(\theta_{n},J^{*})=0$$;
>
> 2) If the loss function has  $0$ as lower bound at infinity, i.e., $\liminf_{\theta\rightarrow \infty}\|\nabla_{\theta}g(\theta)\|=0,$ there is a connected set $J*$ include to $J$ such that the {iterate} $\theta_n$ is convergent to the set $J*$ or $\theta_{n}$ tend to infinity almost surely, i.e.,  $\lim_{n\rightarrow\infty} d(\theta_{n},J*)=0\ \ \ or\ \lim_{n\rightarrow+\infty}\theta_{n}=\infty
> \quad \text{a.s..}$
>
>  We have to notice that the situation where $\theta_{n}\rightarrow \infty\ a.s. $ is not meaningless. In [1'], the authors proved that using logistic regression and GD on separable data can lead to   $\theta_{n}\rightarrow\infty$ and make its direction to a finite value, i.e., $\theta_{n}/\|\theta_{n}\|\rightarrow l_{0}.$
>
> [1']: Soudry, D., Hoffer, E., Nacson, M. S., Gunasekar, S., & Srebro, N. (2018). The implicit bias of gradient descent on separable data. The Journal of Machine Learning Research, 19(1), 2822-2878.
>
> Regarding Assumptions 3.1 2), actually it is  quite general, since it allows the loss function to have multiple stationary points and to be non-convex.   Regarding the connection between  Assumption 3.3 and PL condition, we note that if PL holds, then Ass 3.3 holds, but not vice versa. This is because if PL condition with parameter $\mu>0$ holds, we can let $\delta_0=\mu$, so that  $S=\{\theta|\\|\nabla_{\theta}g(\theta)\\|^{2}/(g(\theta)-g^{*})<\delta_{0}\}=\emptyset$. As we know that empty set is a subset of any set, Assumption 3.3 holds. On the other hand, if PL condition does not hold, Assumption 3.3 may be still met under the two general situations:

---

> > ### Author Response · Authors · 2022-08-02
> > **Response**
> >
> > 1) There are multiple stationary points holding different  values of loss function;  2) The loss function is bounded.  Thus, Assumption 3.3 is much milder than PL condition.
> > In the revised paper, we have added more discussion on the relationship between our assumptions and the ones from existing works.
> >
> >  We have added the proof sketches for the reading convenience.
> >
> >  Regarding the optimal step, we think this problem is tricky and needs more investigation. We would like to work on it as a future work.
> >
> >  Regarding question 8 on the advantages in the theory of using momentum, Theorem 3.2 shows that for mSGD under the same step size, the convergence rate will not decrease as momentum parameter $\alpha$ increases. In addition, according to Theorem 3.3, as the increase of $\alpha$, the gradient norm can converge to an arbitrarily small neighborhood defined with $a$ in the early stage.
> >
> > Regarding the question on 'a' and 'a'',  we note that $a$ is  different from $a'$, since $a$ is  an unknown but existing value for theoretical analysis. The influence of $a'$ exists in the constant term of $O(\cdot)$.

---

> > > ### Comment · Reviewer_aWD4 · 2022-08-08
> > > **Response**
> > >
> > > Thank you for your clarifications.

---

### Author Response · Authors · 2022-08-05
**Looking forward to the feedback of the reviewers**

Dear Reviewers,

      First of all, we would like to thank you for your recommendation as well as your valuable and constructive comments which have helped improve our work substantially.

     In the revised paper,  we have addressed your comments and concerns.   We would be happy to communicate with you if you have further questions or suggestions.  If you think the paper has been improved, we would appreciate it if you can re-evalutate the work.


     Thanks a lot for your time and consideration!

    Best regards,

Authors of Paper 9028

---

### Meta-Review · Area_Chair_ToQQ · 2022-08-29

**Recommendation:** Accept
**Confidence:** Certain

**Metareview:**

The paper studies mSGD and SGD methods and provide convergence under weaker requirement on stepsizes.
The reviewers in general agree that the result is novel, has interesting techniques, and is above bar for Neurips publication.
However, the paper should tone down it's claim about advantage of large step size as it is mostly based on comparing upper bounds.

**Award:**

No

---

### Decision · Program_Chairs · 2022-09-14

Accept